# The ion channel Anoctamin 10/TMEM16K coordinates organ morphogenesis across scales in the urochordate notochord

**Zonglai Liang, Daniel Christiaan Dondorp, Marios Chatzigeorgiou** *

Michael Sars Centre, University of Bergen, Bergen, Norway

* Marios.Chatzigeorgiou@uib.no

**Data Availability Statement:** All individual quantitative observations that underlie the data summarized in the figures and results of our paper can be found in Zenodo: 10.5281/zenodo.

## Abstract

During embryonic development, tissues and organs are gradually shaped into their functional morphologies through a series of spatiotemporally tightly orchestrated cell behaviors. A highly conserved organ shape across metazoans is the epithelial tube. Tube morphogenesis is a complex multistep process of carefully choreographed cell behaviors such as convergent extension, cell elongation, and lumen formation. The identity of the signaling molecules that coordinate these intricate morphogenetic steps remains elusive. The notochord is an essential tubular organ present in the embryonic midline region of all members of the chordate phylum. Here, using genome editing, pharmacology and quantitative imaging in the early chordate *Ciona intestinalis* we show that Ano10/Tmem16k, a member of the evolutionarily ancient family of transmembrane proteins called Anoctamin/TMEM16 is essential for convergent extension, lumen expansion, and connection during notochord morphogenesis. We find that Ano10/Tmem16k works in concert with the plasma membrane (PM) localized $Na^+/Ca^{2+}$ exchanger (NCX) and the endoplasmic reticulum (ER) residing SERCA, RyR, and IP3R proteins to establish developmental stage specific $Ca^{2+}$ signaling molecular modules that regulate notochord morphogenesis and $Ca^{2+}$ dynamics. In addition, we find that the highly conserved $Ca^{2+}$ sensors calmodulin (CaM) and $Ca^{2+}$/calmodulin-dependent protein kinase (CaMK) show an Ano10/Tmem16k-dependent subcellular localization. Their pharmacological inhibition leads to convergent extension, tubulogenesis defects, and deranged $Ca^{2+}$ dynamics, suggesting that Ano10/Tmem16k is involved in both the "encoding" and "decoding" of developmental $Ca^{2+}$ signals. Furthermore, Ano10/Tmem16k mediates cytoskeletal reorganization during notochord morphogenesis, likely by altering the localization of 2 important cytoskeletal regulators, the small GTPase Ras homolog family member A (RhoA) and the actin binding protein Cofilin. Finally, we use electrophysiological recordings and a scramblase assay in tissue culture to demonstrate that Ano10/Tmem16k likely acts as an ion channel but not as a phospholipid scramblase. Our results establish Ano10/Tmem16k as a novel player in the prevertebrate molecular toolkit that controls organ morphogenesis across scales.

12506448. Raw image of gel shown in S4A Fig is also located in 10.5281/zenodo.12506448. Code used to process confocal data and quantify the distance to skeleton metric has been deposited in Zenodo: 10.5281/zenodo.5539439. The Mesmerize GitHub repo with the code used to perform Ca2+ imaging analysis is deposited in Zenodo: 10.5281/zenodo.5539439.

**Funding:** M.C. received funding from the Norges Forskningsråd (https://www.forskningsradet.no/) Grant number 234817. The funders had no role in study design, data collection and analysis, decision to publish, or preparation of the manuscript.

**Competing interests:** The authors have declared that no competing interests exist.

**Abbreviations:** ASW, artificial seawater; AUC, area under the curve; BSA, bovine serum albumin; CaCC, Ca2+-activated Cl- channel; ER, endoplasmic reticulum; GECI, Genetically Encoded Calcium Indicator; ISH, in situ hybridization; MET, mesenchymal-to-epithelial transition; MACS, magnetic-activated cell sorting; ML, Maximum likelihood; NCX, Na+/Ca2+ exchanger; NES, nuclear export sequence; PFA, paraformaldehyde; PM, plasma membrane; PS, phospholipid scramblase; SERCA, Sarcoendoplasmic Reticulum Calcium ATPase.

## Introduction

Organ morphogenesis is a complex biological process that involves the coordinated behavior of cells in space and time to give rise to functional biological form. Elucidating the fundamental mechanisms that coordinate organ morphogenesis across multiple scales, from the molecular to the supracellular levels, will have profound implications for a large range of biological problems in health and disease.

Biological tubes play an essential role in embryogenesis, organogenesis, and postembryonic physiology [1,2]. Indeed, tubular structures are widespread across vertebrates and invertebrates, including salivary glands, renal tubules, vasculature, intestinal tract, and bronchial tubules. Reliable formation of these diverse tubular organs depends on the highly coordinated interplay between collective cell behaviors and complex signaling processes [1,3–5]. Errors in tube formation during development or malfunctions in adult tubular structures can result in severe pathologies [2,6].

The notochord, a defining feature of all chordates, is a tubular organ located along the embryonic midline region of embryos [7,8]. Formation of this flexible rod, tapered at both ends [9], is fundamental to providing structural support to the developing chordate embryo. In the case of vertebrates, the notochord further serves as a signaling center that secretes factors to pattern surrounding tissues [10]. Ascidians, which belong to the sister group to vertebrates the tunicates, have a notochord composed of only 40 cells [11–13]. The small cell number makes it an ideally tractable model for dissecting the rudimentary cellular and molecular mechanisms underlying the multistep process of notochord formation. For example, in *Ciona intestinalis* (formerly known as *C. intestinalis type B*), notochord development begins with cell intercalation-driven convergent extension [14–16]. Next, the notochord elongates to form a cylindrical rod via actomyosin network dependent-cell shape changes [17,18]. Fundamental to tube formation is the establishment of lumens. Here, the notochord cells of *Ciona* undergo a mesenchymal-to-epithelial transition (MET) that leads to the emergence of apical domains at the opposite ends of each cell [11,19]. Extracellular lumens thereby appear and then expand between neighboring cells of the developing tube [17,19–21]. Finally, the notochord cells crawl bidirectionally, exhibiting an endothelial-like cell morphology, which leads to merging of the lumens [11,19].

Tubulogenesis has been intensively studied across a diversity of models, such as the salivary gland of *Drosophila*, the excretory cells of *Caenorhabditis elegans*, the lungs, mammary glands, and neural tube of mice and the notochord of zebrafish and *C. intestinalis* [3,22–24]. Transmembrane proteins have since emerged as an important regulator of signal transduction during the process [22,25–29]. More generally transmembrane proteins such as ion channels and transporters are important in signaling during development [29–37].

To further focus on how bioelectrical signaling shapes the developing tubular organ, we turned to transmembrane pumps, transporters, and ion channels. Of these, the role of the widely conserved Anoctamin (Ano/Tmem16) family proteins [38,39] is essentially unexplored in developmental tube morphogenesis even though several of the Anoctamins have previously been shown to be expressed in mammalian tubular structures including pancreatic acinar cells, proximal renal tubules, and several glands (e.g., submandibular glands, Leydig cells) [40–46].

Anoctamins generally function as Ca2+-activated Cl- channels (CaCCs) and phospholipid scramblases (PS); the latter capable of translocating phospholipids between the 2 monolayers of a membrane [40,41,47]. They have been reported to take part in diverse cellular functions, including for example signal transduction, cell migration, Cl- secretion, and volume regulation [41,43,48,49]. While some Anoctamins, such as ANO1/TMEM16A, act as a CaCC [40,50,51],

and ANO6/TMEM16F as a PS [47]; for other members of the family, including ANO10/ TMEM16K, their functions remain under fruitful debate [41,52,53]. In particular, the subcellular localization and function of several Anoctamins including ANO10/TMEM16K has been debated with some studies reporting, for example, an exclusively intracellular localization for ANO10/TMEM16K with no evidence of activity as a channel [54–56], while others have reported both detectable plasma membrane (PM) localization (albeit with higher portion of localization in intracellular compartments) and channel activity [57–59].

Here, we leverage the simplicity and genetic accessibility of the *C. intestinalis* notochord to study the role of *C. intestinalis* Ano10/Tmem16k (gene model: KH.C3.109/KY21.Chr3.1036 from here on referred to as Ano10) in tube morphogenesis. We employ tissue-specific CRISPR/Cas9 knockout and rescue experiments to reveal that Ano10 is required for convergent extension, lumen expansion, and lumen connection. By combining genome editing and translational fusions, we show that Ano10 is required for the subcellular localization of CaM and CaMK. Using volumetric in vivo functional imaging and the $Ca^{2+}$ integrator CAMPARI, we show that LOF of Ano10 perturbs $Ca^{2+}$ signaling during notochord morphogenesis. Exploiting genetically encoded calcium sponges, genome editing, and pharmacological perturbations, we show that $Ca^{2+}$ signaling machinery in the ER and PM orchestrated by Ano10 contributes to convergent extension and tubulogenesis.

In addition, using time-lapse imaging and translational fusions we show that LOF of Ano10 alters the localization of RhoA and cofilin, interfering with cytoskeletal organization and affecting cell motility and shape. We further demonstrate that Ano10 acts not as a phospholipid scramblase but instead as an ion channel, where its role in establishing and/or maintaining an appropriate electrochemical balance across cell membranes is likely essential for notochord morphogenesis.

## Results

### Ano10 is expressed in the notochord during embryonic development and localizes primarily to the ER

In addition to the previously identified 3 Anoctamins in *C. intestinalis* and *C. savignyi* [39], this work has yielded an additional Anoctamin for *C. intestinalis*. To explore the homology of *C. intestinalis* Anoctamins, we constructed a phylogenetic tree with 89 known Anoctamins from various organisms. Of these, we show that Ciinte.CG.KH.C3.109/ KY21.Chr3.1036 clusters together with vertebrate ANO10/TMEM16K proteins (S1A Fig). Protein alignment analysis additionally revealed that the 10 transmembrane domains and the putative calcium-binding sites of Ano10 share strong conservation with vertebrate ANO10/TMEM16K (S1B Fig).

To determine the expression pattern of *Ano10*, we employed in situ hybridization (ISH) and a transcriptional reporter. Using the first approach, we found that in neurula and initial tailbud stages, *Ano10* mRNA expression was strongest in the mesenchyme and the notochord (S2A–S2D Fig). At later tailbud stages, the expression became localized to the notochord (S2E–S2K Fig). Second, we generated a transcriptional fusion containing a 2-kb region upstream of the *Ano10* start site that drove expression of GFP in the notochord during mid and late tailbud stages (S2L and S2M Fig). Thus, both approaches suggest that *Ano10* is expressed in the developing notochord.

Next, to determine the subcellular localization of Ano10, we co-expressed Ano10::GFP and KDEL::BFP in a notochord-specific manner under the Brachyury promoter, which revealed that at the end of cell elongation (Stage IV according to Dong and colleagues [19]) Ano10 is primarily localized to the endoplasmic reticulum (ER), showing limited localization to the PM

(S2N and S2O Fig). However, in embryos imaged during lumen connection (Stage VII according to Dong and colleagues [19]), Ano10 localized to the PM in addition to the ER (S2O, S2P, and S2Q Fig). This observation was supported by the significantly higher colocalization of the *plasma* membrane marker hCD4::GFP and Ano10::mKate2 at Stage VII compared to Stage III and IV cells (S2P–S2X Fig and S1 Table). In addition, we noticed that during Stage VII, Ano10 was present at the protruding anterior or posterior leading edges [19] of bidirectionally crawling notochord cells (S2P, S2Q white arrowheads). Furthermore, we found that in human cells, Ano10::GFP colocalizes with mCherry-Sec61β, an established ER marker [60] (S3A–S3C Fig). A small amount of Ano10 was found additionally on the PM (S3D–S3G Fig). Combined, our whole-animal and tissue culture data indicate that Ano10 is primarily localized to the ER, with only a small proportion of the protein reaching cell surface.

## *Ano10* is required for notochord convergent extension

Having established that *Ano10* is expressed in the notochord during embryogenesis, we hypothesized that it may contribute to notochord development. To test this, we conducted tissue-specific loss-of-function analysis using the CRISPR/Cas9 system to target *Ano10* exclusively in the notochord [61]. We used a genomic cleavage assay on samples expressing *Cii. Brachyury>nls::Cas9::nls*; *Cii.Brachyury>hCD4GFP* and *U6>Ano10gRNA* or *U6>ControlgRNA* that had been enriched for transfected cells using magnetic-activated cell sorting (MACS). The cleavage assay indicated a 39% in hCD4(+)-sorted cells, whereas cleavage bands from hCD4(-) flow-through or a negative control gRNA were barely visible or absent (S4A Fig). To verify that CRISPR/Cas9-targeted mutagenesis of the *Ano10* locus results in mutant *Ano10* transcripts, we isolated RNA from MACS-selected hCD4(+) cells and sequenced partial Ano10 cDNA fragments that were amplified by RT-PCR and TOPO-cloned. We found that 4 out of 10 sequenced clones contained deletions in the target sequence (S4B Fig).

Confocal microscopy imaging using transgenes labeling the PM and nuclei of notochord cells showed that in negative control embryos notochord development appeared normal both during the process of convergent extension (Fig 1A) as well as during the elongation period resulting in a single column of 40 stacked cells and full tail extension (Fig 1D). In contrast, *Ano10*^CRISPR embryos were characterized by irregular convergent extension at initial tailbud stages (Fig 1B), while at late tailbud stages the embryos exhibited defective Anterior-Posterior (A/P) elongation and abnormal notochord morphology with local thickenings, or multiple columns containing groups of notochord cells that failed to properly intercalate (Fig 1E). We were able to restore notochord morphology by tissue-specific rescue, where we co-electroporated together with the *Ano10* targeting gRNA and the Cas9 mix, a notochord-specific rescue plasmid encoding *Ano10* cDNA with 5 nucleotides replaced (without changing the amino acids encoded) to avoid being targeted by the gRNA (Fig 1C and 1F).

To obtain a quantitative measure of the intercalation defects in *Ano10*^CRISPR embryos, we measured the fraction of notochord cells that have undergone mediolateral intercalation (ML) during notochord development. We find that *Ano10*^CRISPR embryos exhibited a significantly lower fraction of successfully intercalated cells over time compared to negative control and rescue embryos (Fig 1G and S2 Table). At the end of ML intercalation in negative control and rescue embryos almost all cells had successfully intercalated, in contrast to *Ano10*^CRISPR embryos where on average less than 50% of the notochord cells completed intercalation (Fig 1G and S2 Table). A second approach we took to quantify defects in notochord morphology of *Ano10*-^CRISPR embryos due to erroneous intercalation was to measure the distance from the notochord cell center to the midline skeleton. In control embryos where the notochord had

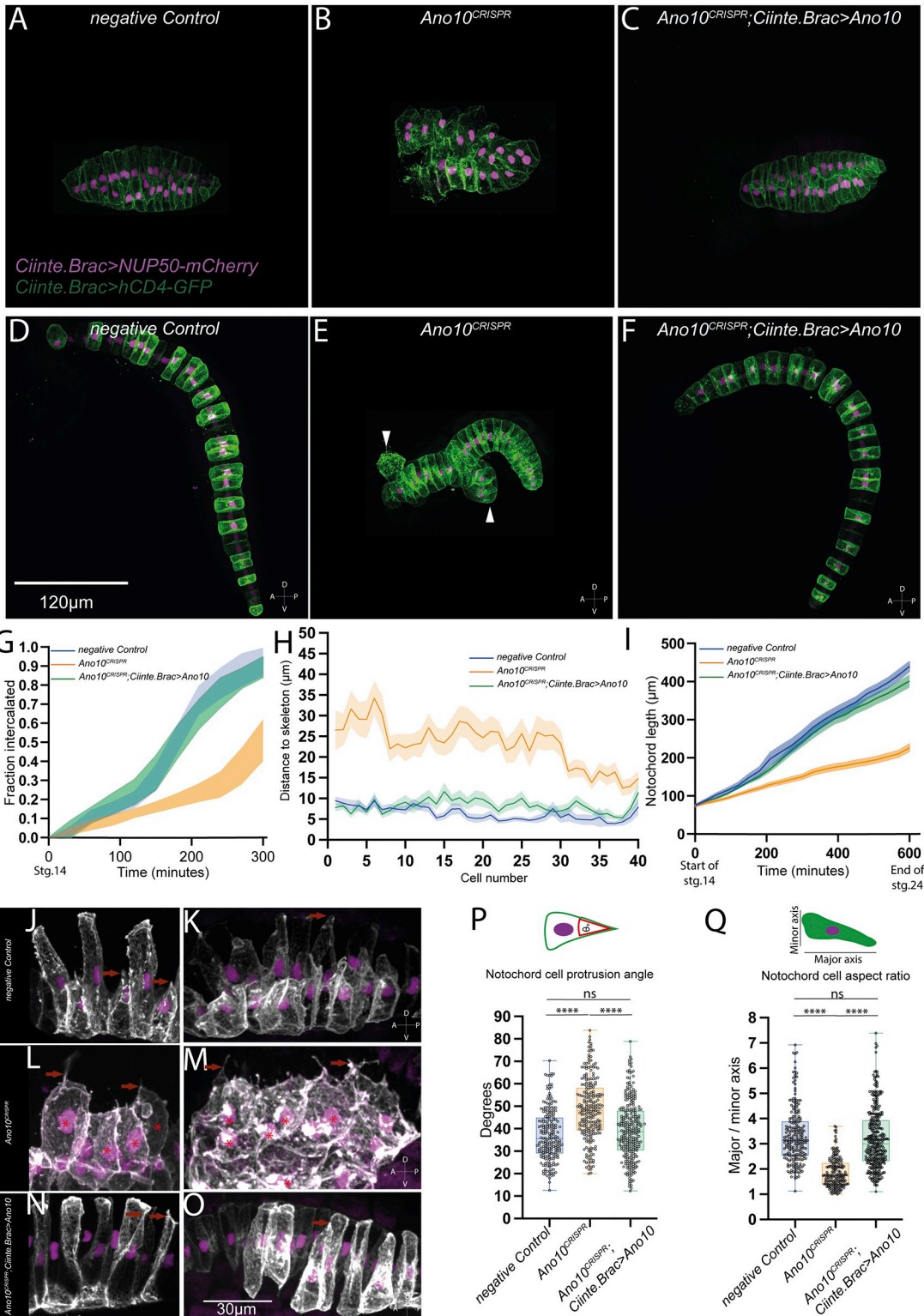

**Fig 1. *Ano10* is required for notochord convergent extension.** (A–C) Maximal projections of confocal stacks showing stage 17 embryos where the notochord cells undergoing intercalation are labeled using *Cinnte.Brac>hCD4GFP* and *Ciinte. Brac>NUP50-mCherry* (A) negative control, (B) *Ano10^CRISPR^*, and (C) rescue *Ano10^CRISPR^; Ciinte.Brac>Ano10*. (D–F) Maximal projections of stage 23 embryos after the completion of notochord cell intercalation. White arrowheads indicate misaligned cells. All embryos are oriented with anterior to the left. (G) Fraction of notochord cells with completed intercalation at a given time point starting

from stage 14 and ending at stage 22. Data presented as mean ± SEM, 30 animals analyzed for each genotype. For statistical analysis, we used a two-way ANOVA, followed by a Tukey's multiple comparisons test (see also S2 Table). (H) Notochord cell misalignment quantification measured as the distance from notochord cell centroid to the embryo midline skeleton in late tailbud embryos as a function of AP position from cell 1 (anterior) to cell 40 (posterior) for the negative control (n = 34 embryos), *Ano10*$^{CRISPR}$ (n = 33 embryos), and *Ano10*$^{CRISPR}$; *Brac>Ano10* rescue embryos (n = 34 embryos). Data presented as mean ± SEM. For statistical analysis, we performed Mann–Whitney test (see also S3 Table). (I) Change in overall notochord length increase during development. Data presented as mean ± SEM, number of animals analyzed: 30≤n≤31. We used a mixed-effects model (REML), followed by Tukey's multiple comparisons test for pairwise comparisons (see also S4 Table). (J–O) Maximal projections of different genetic background embryos with mosaic expression of *Ciinte.Brac>hCD4GFP* and *Ciinte.Brac>NUP50-mCherry* during initial tailbud I stage. Examples of medial protrusions are highlighted with red arrows. Asterisks indicate *Ano10*$^{CRISPR}$ cells, (P, Q). Box plot quantifications of notochord cell medial protrusion angles and cell aspect ratios. We assayed 37≤n≤71 and 189≤n≤355 cells. For statistical analysis, we performed Kruskal–Wallis test, followed by Dunn's multiple comparisons test (see also S5 and S6 Tables for *p*-value details). The numerical data underlying this figure can be found in 10.5281/zenodo.12506448.

successfully completed the intercalation process forming a 40-cell long stack of cells centered along the embryo midline, the distance of each cell center relative to the midline of the skeleton was very short (Fig 1H and S3 Table). In contrast, most notochord cells in the *Ano10*$^{CRISPR}$ embryos were positioned significantly further away from the midline skeleton (Fig 1H and S3 Table). Notochord-specific rescue of the *Ano10*$^{CRISPR}$ defect suggests that *Ano10* acts in a cell autonomous manner to regulate notochord cell intercalation (Fig 1H and S3 Table).

Following the termination of intercalation, notochord cells will elongate along the A/P axis, taking a barrel-like shape [11,19]. As a result of this process, the notochord length progressively increases. We measured the evolution of overall notochord length of the notochord from the point that the sheet of notochord cells invaginates to form a rod of cells until the end of the elongation process. We found that in negative control embryos notochord development proceeded normally resulting in full tail extension (Fig 1I and S4 Table). In contrast, in *Ano10*$^{CRISPR}$ embryos notochord length increased at a slower rate, reaching a shorter final length (Fig 1I) compared to both negative control and rescue embryos (Fig 1I and S4 Table).

We then wondered whether the mediolaterally polarized protrusive activity of notochord cells during cell intercalation was perturbed by the genetic ablation of *Ano10*. In contrast to negative control and rescued notochord cells (Fig 1J, 1K, 1N, and 1O), in *Ano10*$^{CRISPR}$ cells these protrusions appeared shorter and often decorated with multiple filopodia-like structures (Fig 1L and 1M). In addition, we found that the median value of the angle enclosed by the notochord cell protrusion in Ano*10*$^{CRISPR}$ cells was significantly larger compared to negative control and rescued cells (Fig 1P and S5 Table). Conversely, the median aspect ratio of *Ano10*$^{CRISPR}$ cells was significantly smaller (Fig 1Q and S6 Table). Finally, the volume of *Ano10*$^{CRISPR}$ cells was significantly larger compared to negative controls during convergent extension (S4P and S9 Tables). These morphometric parameters indicate that *Ano10* is required for the maintenance of an elongated cell shape and normal cell volume during cell intercalation.

The observed cell intercalation phenotypes could also be potentially attributed to a defect in mediolateral polarization or the loss of the notochord boundary. Visual inspection of confocal stacks indicated that *Ano10*$^{CRISPR}$ notochord cells exhibited protrusions with an overall mediolateral bias, while the nuclei were aligned along the mediolateral axis of the intercalating cells like control embryo cells. Taken together with the observation that cell protrusions had a mediolateral bias, it would suggest that Ano10 is not required for mediolateral polarization. Analysis of the border of *Ano10*$^{CRISPR}$ notochords (S4D Fig) showed reduced regularity and definition compared to negative control notochords (S4C Fig). To quantify this border defect, we calculated the ratio of total/net boundary length (S4E–S4J Fig). In *Ano10*$^{CRISPR}$ embryos this ratio was significantly higher compared to negative controls (S4K Fig and S7 Table).

Rescued embryos had a fully restored total/net boundary length ratio (S4K Fig and S7 Table). Importantly, we detected laminin by immunostaining at the notochord boundary of both negative control and Ano10$^{CRISPR}$ albeit with weaker staining (S4L–S4N Fig), suggesting that the notochord boundary is present, but deformed, in Ano10$^{CRISPR}$. Finally, we note that the defects in intercalation of Ano10$^{CRISPR}$ notochord cells did not emerge from spurious cell proliferation since both negative control and Ano10$^{CRISPR}$ embryos had a median of 40 cells per notochord (S4O Fig and S8 Table).

## Ano10 cooperates with ER and PM Ca$^{2+}$ signaling machinery to mediate convergent extension

Anoctamins interact with the Ca$^{2+}$ signaling machinery of cells to modulate intracellular Ca$^{2+}$ dynamics across a range of tissues [62–67]. However, in the case of *Ciona* notochord development, the importance of Ca$^{2+}$ as a second messenger, as well as the role of specific Ca$^{2+}$ signaling proteins is unclear.

To establish whether Ca$^{2+}$ is required for notochord intercalation, we used the genetically encoded Ca$^{2+}$ scavenger termed SpiCee [68]. This tool allowed us to sequester Ca$^{2+}$ signaling in the notochord. We employed three different versions of SpiCee. The first variant we tested was $^{mut}$SpiCee in which the calcium-binding sites have been mutated. This served as a negative control. To alter local Ca$^{2+}$ signaling at the PM, we used the Lyn-SpiCee variant which contains a tandem of palmitoylation-myristoylation motifs from Lyn Kinase. Finally, to sequester Ca$^{2+}$ signaling in the cytoplasm we employed a nuclear export sequence (NES) tagged SpiCee variant (hereafter termed SpiCee-NES).

$^{mut}$SpiCee embryos that completed convergent extension (Fig 2A) were morphologically indistinguishable from wild-type embryos. In contrast, Lyn-SpiCee and SpiCee-NES embryos were stubby (Fig 2B and 2C). At the end of convergent extension $^{mut}$SpiCee notochords reached a wild-type mean length (410.6 μm) but Lyn-SpiCee (338.4 μm) and SpiCee-NES (255.3 μm) embryos were significantly shorter (Fig 2D and S10 Table). In comparison to $^{mut}$SpiCee (31.37˚) intercalating cells, both Lyn-SpiCee (45.60˚) and SpiCee-NES (52.01˚) cells were characterized by a larger mean value of the angle enclosed by the notochord cell protrusion (Fig 2E and S11 Table). The corresponding Lyn-SpiCee and SpiCee-NES embryos exhibited slower medio-lateral intercalation rates compared to $^{mut}$SpiCee embryos (Fig 2F and S12 Table). Finally, we quantified the defects in notochord morphology at the end of convergent extension due to erroneous intercalation. In $^{mut}$SpiCee embryos where the notochord had successfully completed the intercalation process, the distance of each cell center relative to the midline of the skeleton was very short (Fig 2G and S13 Table). In contrast, most notochord cells in the Lyn-SpiCee and SpiCee-NES embryos were positioned significantly further away from the midline skeleton (Fig 2G and S13 Table). Our analysis suggests that Ca$^{2+}$ is an essential second messenger for robust convergent extension and that local sequestering of Ca$^{2+}$ signaling is sufficient to impact early stages of notochord morphogenesis.

Having established the importance of Ca$^{2+}$ as a second messenger in the notochord, we turned our attention to the molecular toolkit that likely regulates Ca$^{2+}$ signaling during notochord convergent extension. Since cellular Ca$^{2+}$ homeostasis, depends on the uptake and release of Ca$^{2+}$ from the ER and the extracellular milieu, we focused on proteins that are known mediators of these processes [69–72]. By applying pharmacological inhibitors during convergent extension, we blocked the activity of 3 key modulators of intracellular Ca$^{2+}$ stores [69,72], namely the Sarcoendoplasmic Reticulum Calcium ATPase (SERCA), the Ryanodine receptor (RyR), the Inositol 1,4,5-trisphosphate receptor (IP$_3$R), and the plasma membrane Na$^+$/Ca$^{2+}$ exchanger (NCX) which plays a critical role in the maintenance of Ca$^{2+}$ homeostasis

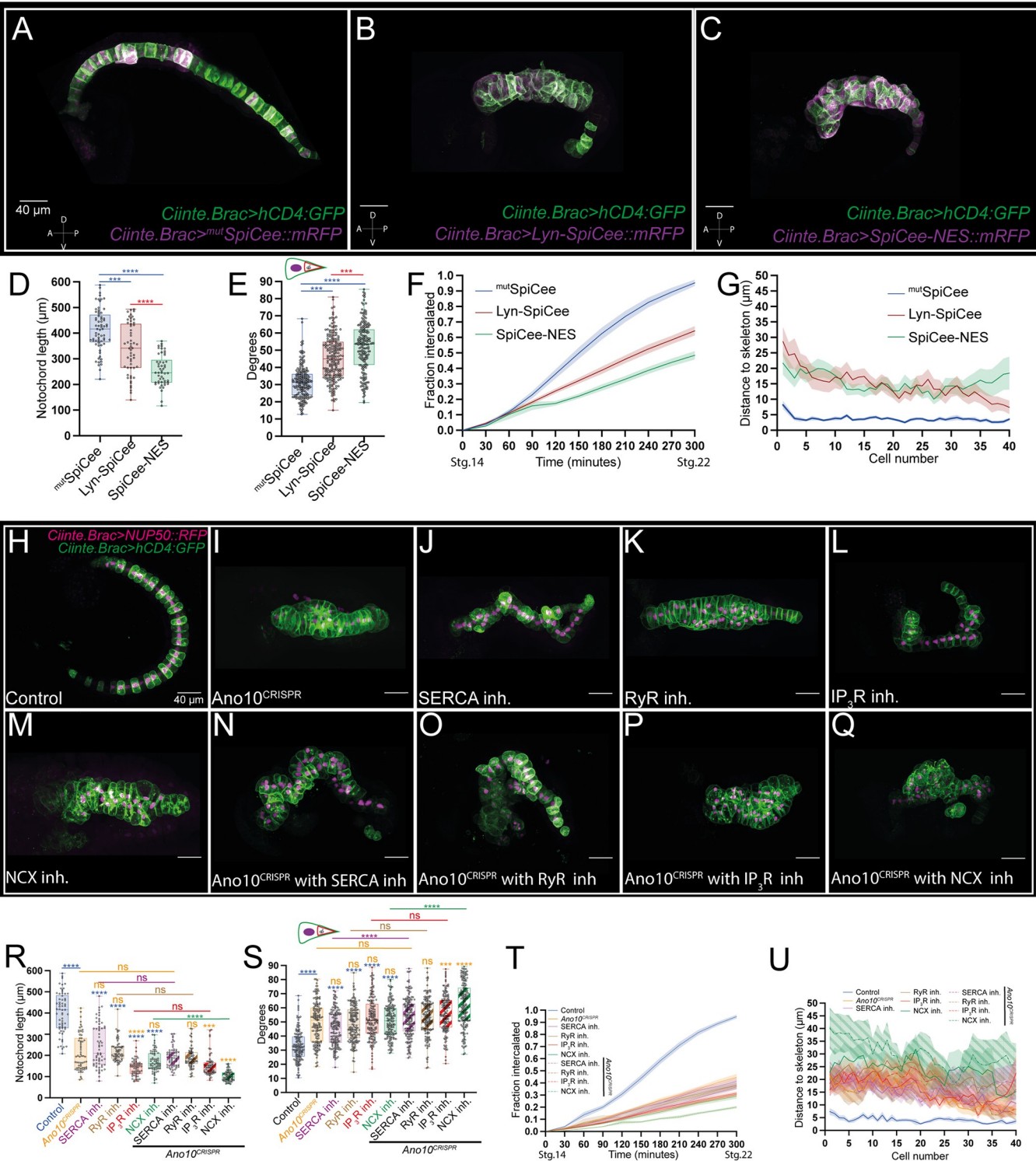

**Fig 2. Ano10 works with ER and PM residing Ca$^{2+}$ machinery to modulate convergent extension.** (A–C) Maximal projections of stage 23 embryos after the completion of notochord cell intercalation. Transgenic embryos are labeled using *Ciinte.Brac>hCD4GFP* in combination with the following variants of the Ca$^{2+}$ scavenger SpiCee (A) *Ciinte.Brac>mtSpiCee::mRFP*, (B) *Ciinte.Brac>Lyn-SpiCee::mRFP*, and (C) *Ciinte.Bra>SpiCee-NES::mRFP*. All embryos are oriented with anterior to the left. Lateral views shown. (D) Notochord length at the end of convergent extension. Data presented as mean ± SEM, 50≤n of animals per genotype. For statistical analysis, we performed a Kruskal–Wallis test followed by a Dunn's multiple comparisons test (see also S10 Table). (E) Box plots quantifying notochord cell medial protrusion angles across the 3 genotypes, and 200 cells from 40 animals were quantified for each genotype. For statistical analysis, we performed a Kruskal–Wallis test followed by a Dunn's multiple comparisons test (see also S11 Table). (F) Fraction of notochord cells with

completed intercalation at a given time point starting from stage 14 and ending at stage 22. Data presented as mean ± SEM, 15 animals analyzed for each genotype. For statistical analysis, we used a mixed-effects model (REML), followed by a Tukey's multiple comparisons test (see also S12 Table). (G) Notochord cell misalignment quantification measured as the distance from notochord cell centroid to the embryo midline skeleton in late tailbud embryos as a function of AP position from cell 1 (anterior) to cell 40 (posterior) for all 3 genotypes. Data presented as mean ± SEM, 42≤n of animals per genotype. For statistical analysis, we used a mixed-effects model (REML), followed by a Tukey's multiple comparisons test (see also S13 Table). (H–Q) Maximal confocal projections of stage 23 embryos after the completion of notochord cell intercalation. Transgenic embryos are labeled using *Ciinte.Brac>hCD4GFP* and *Ciinte.Brac>NUP50::mCherry*. Anterior is to the left. Lateral views shown. (H) Control embryos treated with 5 μm DMSO (I) *Ano10^CRISPR*, (J–M) embryos treated with pharamacological inhibitors against SERCA, RyR, IP3R, and NCX proteins. All inhibitors were tested at 100 μm. (N–Q) *Ano10^CRISPR* embryos treated with the same inhibitors. (R–U) Quantifications of notochord intercalation metrics for control, *Ano10^CRISPR*, inhibitors treated and *Ano10^CRISPR* combined with inhibitors embryos. (R) Quantification of notochord length. For statistical analysis, we performed a Kruskal–Wallis test followed by a Dunn's multiple comparisons test; 62≤n animals per condition (see also S14 Table). The significance notation above each box plot is color-coded according to the experimental condition that it is compared to (e.g., when compared to control asterisks or ns are colored blue). (S) Notochord cell medial protrusion angles. For statistical analysis, we performed a Kruskal–Wallis test followed by a Dunn's multiple comparisons test; 29≤n animals, 145≤n cells per condition (see also S15 Table). (T) Intercalation rates. Data presented as mean ± SEM, 15≤n animals per genotype. For statistical analysis, we used a mixed-effects model (REML), followed by a Tukey's multiple comparisons test (see also S16 Table). (U) Notochord cell misalignment along the A/P midline. Data presented as mean ± SEM, 28≤n animals per genotype. For statistical analysis, we used a mixed-effects model (REML), followed by a Sidak's multiple comparisons test (see also S17 Table). The numerical data underlying this figure can be found in 10.5281/zenodo.12506448. ER, endoplasmic reticulum; NCX, Na+/Ca2+ exchanger; PM, plasma membrane.

by either lowering or raising the intracellular [Ca2+] [69,73] (see Methods for details on the pharmacological inhibitors).

We found that in contrast to controls (incubated in 5 μm DMSO), drug-treated embryos were stubbier exhibiting irregular notochord morphology with local thickenings composed of multiple notochord cells that failed to intercalate, phenotypically resembling Ano10^CRISPR embryos (Fig 2H–2M). The strong effects of pharmacologically inhibiting SERCA, RyR, IP3R, and NCX were confirmed by multiple intercalation metrics where drug-treated embryos were statistically significantly different from controls (Fig 2R–2U and S14–S17 Tables). The effects of the drug treatments were largely indistinguishable when compared to Ano10^CRISPR embryos with respect to the mean value of the angle enclosed by the notochord cell protrusion and the notochord length (except for IP3R inhibition which led to statistically significant shorter notochords) (Fig 2R and 2S and S14 and S15 Tables). However, all drug treatments except for SERCA inhibition resulted in slower intercalation rates and higher rates of erroneous intercalation (quantified as the mean distance from the notochord cell center to the midline skeleton) that were statistically significant (Fig 2T and 2U and S16 and S17 Tables). These results indicate that the regulation of ER Ca$^{2+}$ stores and the transport of Ca$^{2+}$ across the PM are both required for robust convergent extension.

Subsequently, we repeated the pharmacological inhibition experiments in an Ano10^CRISPR background to determine whether the effects of the treatments were enhanced (or not) by the absence of Ano10 in the signaling machinery of the notochord cells. Morphologically, Ano10^CRISPR combined with either the IP3R or the NCX inhibitor showed enhanced notochord irregularities and appeared even stubbier compared to Ano10^CRISPR alone or IP3R and NCX inhibitors alone (Fig 2H–2Q). Ano10^CRISPR embryos treated with the NCX inhibitor were the treatment group to show the most significant differences in notochord intercalation compared to either the Ano10^CRISPR or inhibitor alone groups (Fig 2R–2U and S14–S17 Tables). These results suggest that NCX is essential for convergent extension.

Inhibition of IP3R in Ano10^CRISPR embryos gave a significantly stronger phenotype compared to Ano10^CRISPR alone (but not compared to IP3R inhibition alone), raising the possibility that Ano10 is acting as a downstream effector of IP3R (Fig 2R–2U and S14–S17 Tables).

On the other hand, groups composed of Ano10^CRISPR embryos treated with either the SERCA or the RyR inhibitor did not show a statistically significant enhancement in 3 out of 4 intercalation metrics relative to Ano10^CRISPR or inhibitors alone suggesting that these act in the same pathway as Ano10 (Fig 2R–2U and S14–S17 Tables).

## Ano10 is required for notochord tubulogenesis

We then sought to investigate whether Ano10 contributes to the process of tubulogenesis. While *Ano10*CRISPR embryos generated using the *Brachyury* promoter (driving Cas9) exhibited strong tubulogenesis phenotypes, we were concerned that these defects could be the indirect outcome of erroneous cell intercalation. To overcome this issue, we drove expression of Cas9 using the promoter of carbonic anhydrase [74] (S5A–S5C Fig) which is expressed in the notochord at later developmental stages compared to Brachyury [75,76], thus allowing us to assess the role of Ano10 specifically during tubulogenesis. In particular, we found that the *Ciinte. Cah>GFP* reporter expressed in almost half of the embryos at Hotta stage 20, with the highest frequency of expression in embryos achieved during Hotta stages 22 to 26 [77] (S5A–S5C and S18 Table). To obtain independent validation of the genome editing based findings, we used an acute pharmacological perturbation by applying 3 different drugs that are well-established blockers of calcium-activated chloride channels [78–80] just prior to the onset of lumen initiation (at the end of stage IV [19]). To quantitatively assess notochord cell behavior during lumen formation, we generated and imaged using time-lapse confocal microscopy transgenic embryos co-expressing the transmembrane proteins *hCD4::GFP* and *Ciiinte.Caveolin 1-mCherry*. Caveolin 1 has been shown to be enriched at the apical/luminal domain of *Ciona* notochord cells during tubulogenesis [26] where it contributes to lumen formation by inducing membrane curvature and vesicular trafficking [81]. In this way, we were able to simultaneously visualize the PM and the apical/luminal region of the notochord cells. Following cell elongation in negative control CRISPR embryos (Fig 3A, asterisks) or DMSO-treated embryos (S6A Fig, asterisks), apical domains form in the center of the lateral domains as previously shown [26]. These are also present in *Ano10*CRISPR notochord cells (Fig 3B, asterisks) indicating that Ano10 is not involved in cell polarization and lumen initiation. We also observed apical domains in drug-treated animals (S6B–S6D Fig). In addition, when we measured the longitudinal radius of the lumen at the end of the initiation period, the median value for *Ano10*CRISPR cells was indistinguishable from that of negative controls (Fig 3D and 3E and S19 Table).

However, *Ano10*CRISPR notochord cells displayed a limited ability to expand their lumen (Fig 3B, white arrowheads). We found that the lumen longitudinal radius at the end of the expansion phase was significantly shorter in *Ano10*CRISPR compared to negative control and rescued embryos (Fig 3A–3C and 3F and S20 Table). Additionally, lumen curvature measured from transverse notochord sections was significantly higher in *Ano10*CRISPR cells suggesting that they had a narrower lumen compared to negative control and rescued cells (Fig 3D and 3G and S21 Table). Drug-treated notochord cells exhibited a similar behavior in terms of lumen initiation (S6B–S6E Fig and S23 Table), expansion (S6B–S6D Fig white arrows, S6F Fig and S24 Table), and curvature (S6B–S6D, S6G Fig and S25 Table) as *Ano10*CRISPR notochord cells. In *Ano10*CRISPR notochords, the gap between neighboring expanding lumens was larger compared to that of negative controls, suggesting that the lumens of mutant embryos were not able to extend sufficiently for them to eventually fuse (S6I Fig and S27 Table). Taken together, these findings suggest that Ano10 is not required for lumen initiation, but it is necessary for lumen expansion and lumen geometry. Following lumen expansion negative control notochord cells initiate the process of tilting and connecting the notochord lumen (Fig 3A and 3C, see $ sign). We observed that *Ano10*CRISPR notochord cells were not able to tilt and connect the notochord lumen to the same extent as negative control and rescue cells (Fig 3B). To analyze the dynamics of lumen growth, we measured the cross-sectional area change from lumen initiation until the completion of lumen connection (Fig 3H and S22 Table). We found that *Ano10*CRISPR notochord lumens showed a consistently narrower cross-sectional area compared

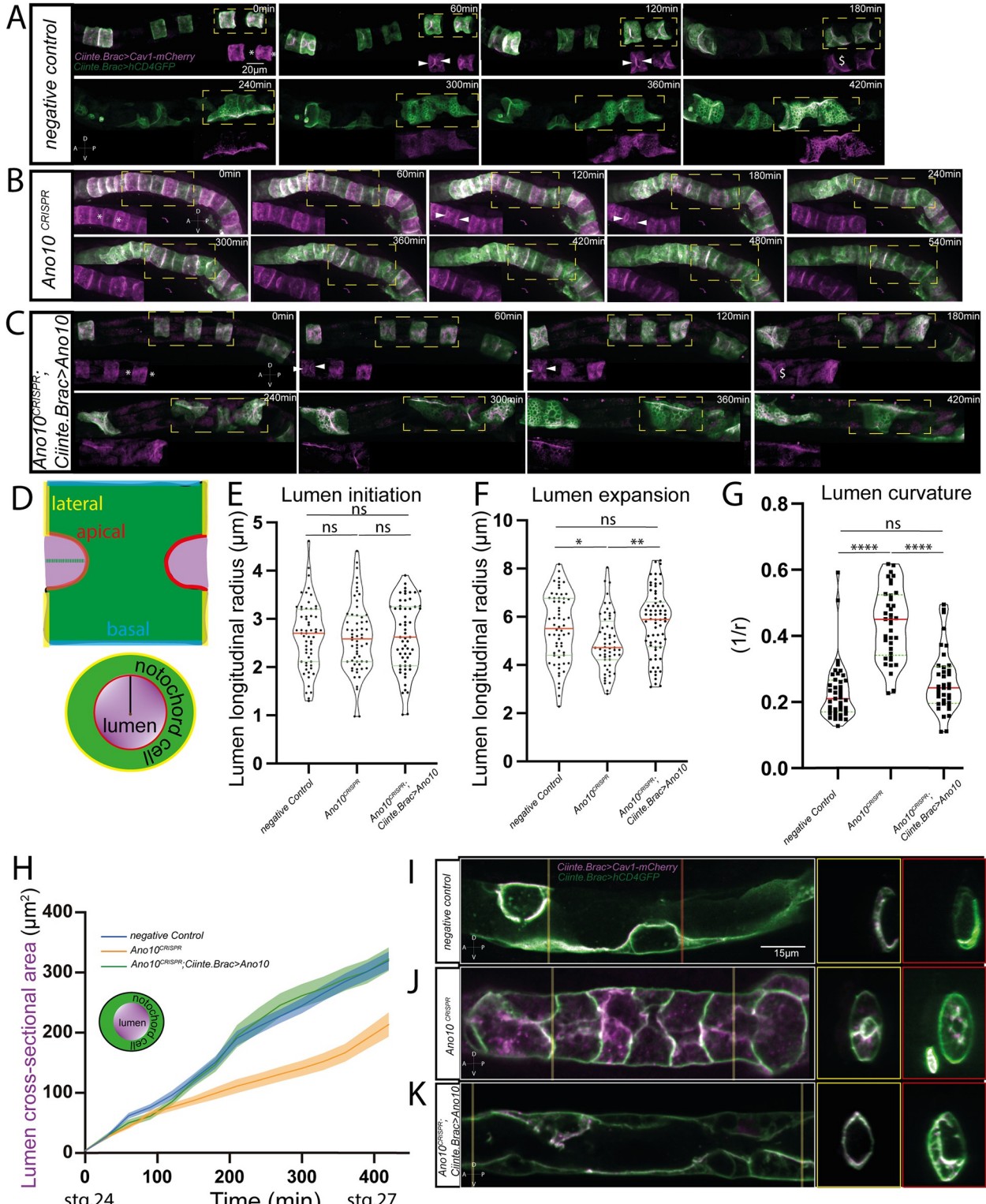

**Fig 3. Ano10 is required for notochord tubulogenesis.** (A–C) Montage of maximum projections showing confocal time lapse imaging of *Ciiinte.Brac>hCD4::GFP; Ciiinte.Caveolin 1-mCherry* covering the period of tubulogenesis. Insets show *Ciiinte.*Caveolin 1-mCherry signal from the dashed orange boxes. Asterisks indicate the emerging apical domains. White arrowheads point to the expanding lumen (see also S1–S3 Movies). All embryos are oriented with anterior to the left. (D) Top schematic illustrates a longitudinal view of the notochord cell during lumen extension. The red lines define the apical domain and the dashed green line the longitudinal radius. The purple area defines the expanding lumen volume. The

yellow and blue lines indicate the lateral and basal domains. Lower schematic illustrates a transverse section through the same cell. Black line indicates radius of curvature and the brown point the center of curvature. (E) Quantification of lumen longitudinal radius during lumen initiation. For statistical analysis, we performed a Kruskal–Wallis test followed by a Dunn's multiple comparisons test; $59 \leq n$ cells and $11 \leq n$ of animals per condition (see also S19 Table). The significance notation above each box plot is color-coded according to the experimental condition that it is compared to (e.g., when compared to control asterisks or ns are colored blue). (F) Quantification of lumen longitudinal radius during lumen expansion. For statistical analysis, we performed a Kruskal–Wallis test followed by a Dunn's multiple comparisons test; $56 \leq n$ cells and $11 \leq n$ of animals per condition (see also S20 Table). (G) Quantification of lumen curvature after lumen connection. For statistical analysis, we performed a Kruskal–Wallis test followed by a Dunn's multiple comparisons test; $37 \leq n$ cells and $7 \leq n$ of animals per condition (see also S21 Table). (H) Evolution of lumen cross-sectional area during tubulogenesis. Schematic shows in purple the lumen cross-section and in green the notochord cell membrane. For statistical analysis, we performed a two-way ANOVA followed by Tukey's multiple comparison test. We used $20 \leq n \leq 25$ of animals per condition (see also S22 Table). (I–K) Longitudinal cross-sections of example notochords from negative control (I), *Ano10*$^{CRISPR}$ (J), and rescue (K) embryos. Transverse sections from 2 points along the notochord. The most anterior is indicated by a yellow bar and the more posterior section is highlighted with a red bar. Notochords are oriented with anterior to the left (lateral views). The numerical data underlying this figure can be found in 10.5281/zenodo.12506448.

to negative controls and that this defect could be rescued in a tissue specific manner (Fig 3H–3K and S22 Table). Drug-treated notochords exhibited a lumen cross-sectional area progression phenotype that was very close to that of *Ano10*$^{CRISPR}$ notochords (S6H Fig and S26 Table). In addition, the tilting angle of the lumen pockets was smaller compared to controls, suggesting that tilting direction is also a contributor to the reduced success with which lumen formation takes place in *Ano10*$^{CRISPR}$ embryos. We note that we were not able to fully rescue this phenotype (S6J Fig and S28 Table). Thus, we believe that Ano10 is required for expanding, tilting, and connecting the notochord lumen.

## Lumen formation is regulated through a synergy of Ano10 with ER and PM Ca$^{2+}$ signaling machinery

Having demonstrated that Ano10 is required for robust tubulogenesis, we went on to ask whether the interactions we identified between Ano10 and the cellular Ca$^{2+}$ signaling machinery throughout intercalation are preserved during lumen formation.

To establish in the first instance whether Ca$^{2+}$ is required during lumen formation, we leveraged the genetically encoded Ca$^{2+}$ scavenger SpiCee [68]. However, for this set of experiments the different SpiCee variants were driven by the *Ciinte.Cah* promoter, which is switched on later in development compared to *Ciinte.Brachyury*. This combination allowed us to evaluate the outcome of sequestering Ca$^{2+}$ signaling in the notochord specifically during tubulogenesis.

In contrast to our negative control embryos ($^{mut}$SpiCee) which successfully completed tubulogenesis forming a wide and continuous lumen, Lyn-SpiCee and SpiCee-NES embryos failed to complete lumen formation featuring narrower lumens that mostly failed to connect (Fig 4A–4F and S29–S31 Tables).

Therefore, our analysis demonstrates that the compartmentalized sequestration of Ca$^{2+}$ in notochord cells during tubulogenesis impedes lumen formation.

Subsequently, we turned our attention to the molecular players that likely regulate Ca$^{2+}$ signaling during tubulogenesis. Using pharmacological inhibitors from the end of the notochord developmental Stage IV up to and include Stage VII, we blocked the activity of SERCA, RyR, IP$_3$R [69,72], and NCX [69,73] throughout tubulogenesis.

Control embryos (incubated in 5 μm DMSO) were able to complete tubulogenesis (Fig 4G). Similarly, SERCA and NCX inhibitor-treated embryos were also able to form lumens which were characterized by curvature, lumen radius and cross-sectional area values comparable to those of control embryos (Fig 4I, 4O, and 4Q–4S and S32–S34 Tables).

In contrast, inhibition of the IP$_3$ receptor and the Ryanodine receptor led to statistically significant effects on lumen formation (Fig 4K, 4M, and 4Q–4S and S32–S34 Tables). Compared

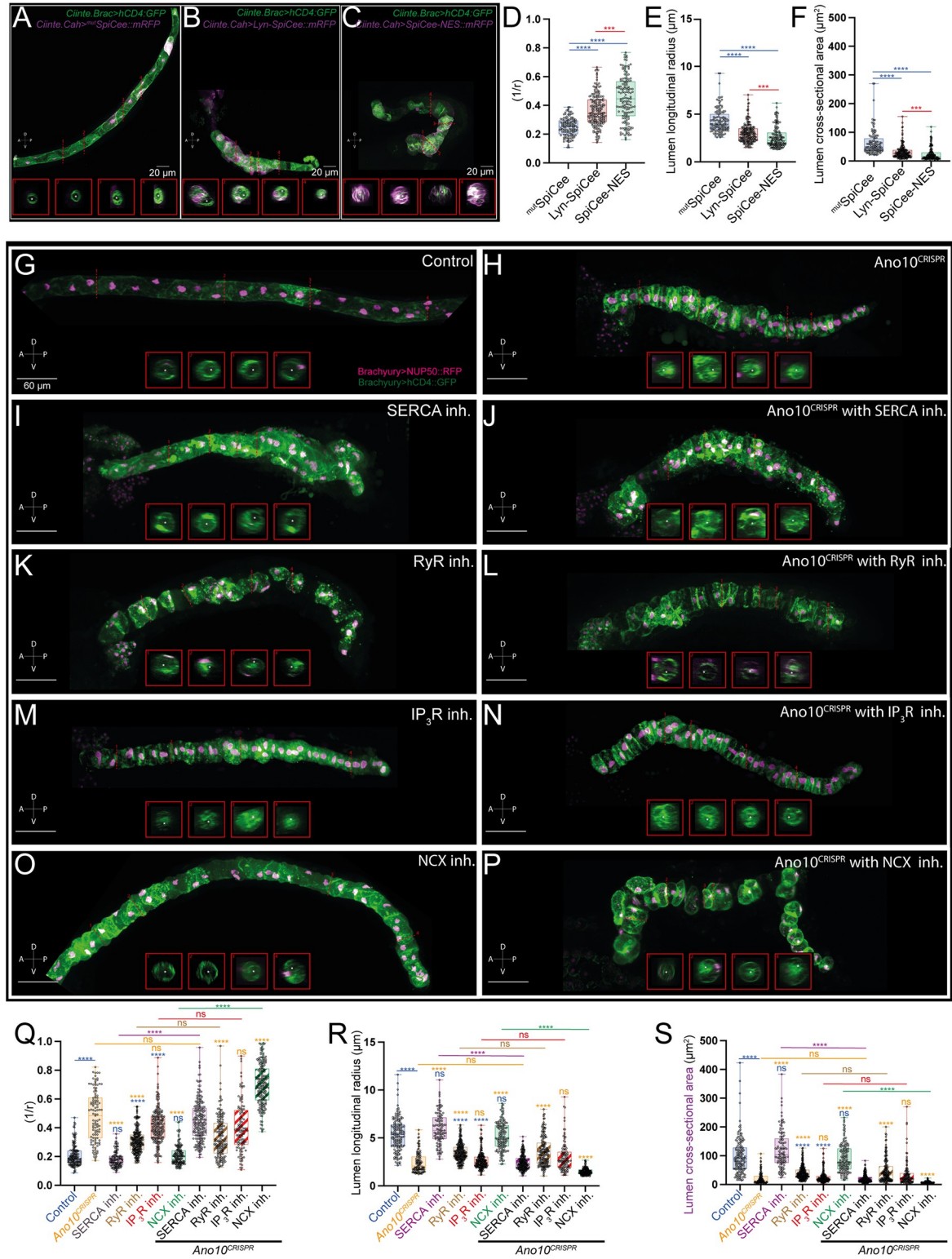

**Fig 4. Ano10 cooperates with ER and PM machinery to modulate lumen formation during tubulogenesis.** (A–C) Maximal projections of stage 26 *Cionas* upon the completion of notochord development stage VII. Transgenic animals are labeled using *Ciinte. Brac>hCD4GFP* in combination with the following variants of the Ca²⁺ scavenger SpiCee (A) *Ciinte.Cah>mtSpiCee::mRFP*, (B) *Ciinte. Cah>Lyn-SpiCee::mRFP*, and (C) *Ciinte.Cah>SpiCee-NES::mRFP*. All animals are oriented with anterior to the left. Lateral views shown. Insets show transverse sections of the notochord corresponding to the locations indicated with the red dashed lines (numbers 1–4).

Asterisks mark the lumen. (D–F) Quantification of lumen metrics across the 3 SpiCee variants. (D) Lumen curvature; (E) lumen longitudinal radius; (F) lumen cross-sectional area. For statistical analysis of data presented in panels D–F, we performed a Kruskal–Wallis test followed by a Dunn's multiple comparisons test. We used 25≤n animals, 126≤n cells per condition (see also S29–S31 Tables). (G–P) Maximal projections of stage 26 *Cionas* upon the completion of notochord development stage VII. Transgenic animals are labeled using *Ciinte.Brac>hCD4GFP* and *Ciinte.Brac>NUP50-mCherry*. Insets show transverse sections at different locations along the notochord marked by the numbered red dashed lines. The lumen is marked with an asterisk. Lateral views of the animals are shown, with the anterior side located on the left. Scale bars = 60 μm. (G) Control embryos treated with 5 μm DMSO (H) $Ano10^{CRISPR}$. (I, K, M, O) Embryos treated with pharamacological inhibitors against SERCA, RyR, IP3R, and NCX proteins. All inhibitors were tested at 100 μm. (J, L, N, P) $Ano10^{CRISPR}$ embryos treated with the same inhibitors. (Q–S) Quantification of lumen metrics across the different genetic backgrounds and pharmacological treatments (Q) lumen curvature; (R) lumen longitudinal radius; (S) lumen cross-sectional area. For statistical analysis of data presented in panels Q–S, we performed a Kruskal–Wallis test followed by a Dunn's multiple comparisons test. We used 21≤n ≤50 animals, 109≤n≤253 cells per condition (see also S32–S34 Tables). The numerical data underlying this figure can be found in 10.5281/zenodo.12506448. ER, endoplasmic reticulum; NCX, Na+/Ca2+ exchanger; PM, plasma membrane; SERCA, Sarcoendoplasmic Reticulum Calcium ATPase.

to $Ano10^{CRISPR}$, the effects of RyR inhibition on tubulogenesis were significantly less severe; however, inhibition of $IP_3R$ led to lumen metrics, which were statistically indistinguishable from those of $Ano10^{CRISPR}$ embryos (Fig 4H, 4K, 4M, and 4Q–4S and S32–S34 Tables).

Additionally, we repeated these pharmacological inhibition experiments, in an Ano10-$^{CRISPR}$ background to establish whether the effects of the pharmacological treatments were altered by the loss of function of Ano10 in notochord cells. Morphologically, we noticed a striking phenotypic difference between genetically wild-type embryos treated with the NCX inhibitor and $Ano10^{CRISPR}$ embryos treated with same inhibitor (Fig 4O and 4P). Notably, NCX inhibitor-treated $Ano10^{CRISPR}$ embryos showed a significantly more severe lumen phenotype compared to that of $Ano10^{CRISPR}$ alone or NCX inhibitor alone embryos (Fig 4O, 4P, and 4Q–4S and S32–S34 Tables). Our data uncover a cryptic interaction between Ano10 and NCX during tubulogenesis, raising the possibility that NCX activity is subject to regulation by an Ano10-dependent mechanism.

Interestingly, $Ano10^{CRISPR}$; SERCA inhibitor-treated embryos had a similar phenotype in terms of lumen metrics to $Ano10^{CRISPR}$ embryos which suggests that SERCA is not involved in tubulogenesis (Fig 4H–4J and 4Q–4S and S32–S34 Tables). However, $Ano10^{CRISPR}$ embryos treated with the RyR inhibitor showed a significantly weaker phenotype as compared to $Ano10^{CRISPR}$ embryos, indicating that the Ryanodine receptor may be acting upstream of Ano10 during tubulogenesis (Fig 4H, 4K, 4L, and 4Q–S and S32–S34 Tables). Finally, Ano10-$^{CRISPR}$ embryos treated with the $IP_3R$ inhibitor showed a phenotype that was statistically indistinguishable from that of either $Ano10^{CRISPR}$ embryos or embryos treated with just the $IP_3R$ inhibitor (Fig 4H, 4M, 4N, and 4Q–4S and S32–S34 Tables).

## The $Ca^{2+}$ sensors CaM and CaMK modulate convergent extension and lumen formation, while their subcellular localization is Ano10 dependent

This work points towards an important role for the second messenger $Ca^{2+}$ in notochord development. The presence of multiple proteins and compartments that may modulate $Ca^{2+}$ signaling in the notochord paints a quite complex $Ca^{2+}$ signaling network that may be involved in a myriad of complex processes taking place during notochord morphogenesis such as cell shape changes, cell movement, lumen expansion, and others. Many of the second messenger effects of $Ca^{2+}$ are mediated through the $Ca^{2+}$ sensing proteins CaM and CaMK [82–85]. Thus, we asked the question of whether CaM and CaMK are required for during the morphogenetic process of convergent extension and tubulogenesis.

First, we explored the role of CaM and CaMK during convergent extension. The mCherry translational fusions of CaM and CaMK both localized to the notochord cell–cell contacts, showing strong positive correlation with a hCD4::GFP plasma membrane marker (Fig 5A 5D,

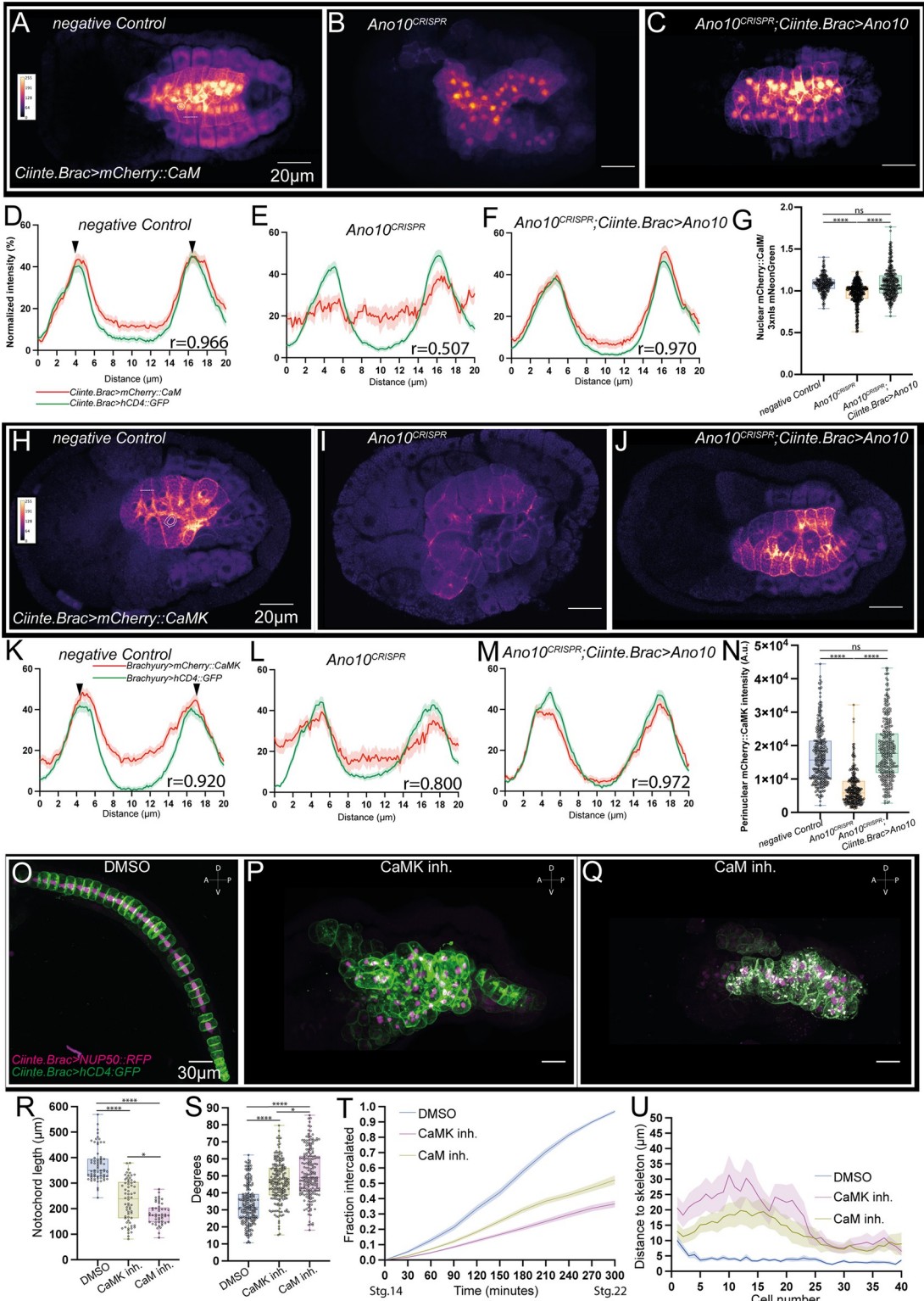

**Fig 5. The Ca²⁺ sensors CaM and CaMK modulate convergent extension, while their subcellular localization at/around the nuclei and cell–cell contacts and is Ano10 dependent.** (A–C) Median confocal sections of 3 different genotypes: negative control, Ano10^CRISPR and rescue embryos undergoing notochord intercalation. The embryos express *Ciinte.Brac>mCherry::CaM* and *Ciinte.Brac>hCD4::GFP*; however, only the *mCherry::CaM* channel is shown for clarity. Images are pseudocolored with mpl-plasma. The fluorescence intensity calibration scale bar is shown in panel A. Dented white line illustrates an example

line used to quantify normalized intensities across cell–cell contacts shown in panels D–F. Dashed circle correspond to a typical ROI used for quantification in panel G. (D–F) Quantification of normalized fluorescence intensity across notochord cells of *Ciiinte.Brac>mCherry::CaM; Ciiinte.Brac>hCD4::GFP* double transgenic embryos. Green peaks highlighted with black arrowheads in panel D correspond to cell–cell membrane contacts. The Pearson's correlation coefficients (r) are shown. The lower (LL) and upper (UL) 95% confidence intervals were as follows: negative control (LL: 0.9524, UL: 0.9755), Ano10^CRISPR (LL: 0.3748, UL: 0.6195), and rescue embryos (LL: 0.9587, UL: 0.9788). We used 19≤n ≤21 animals and 99≤n≤107 cells per genotype. (G) Box plots quantifying the expression levels of *mcherry::CaM* relative to *3xnls::mNeonGreen* across the 3 different genotypes. For statistical analysis of the data, we performed a Kruskal–Wallis test followed by a Dunn's multiple comparisons test. We used 60≤n animals and 301≤n cells per genotype (see also S35 Table). (H–J) Median confocal sections of 3 different genotypes: negative control, Ano10^CRISPR and rescue embryos undergoing notochord intercalation. The embryos express *Ciiinte. Brac>mCherry::CaMK* and *Ciiinte.Brac>hCD4::GFP*; however, only the *mCherry::CaMK* channel is shown for clarity. Images are pseudocolored with mpl-plasma. The fluorescence intensity calibration scale bar is shown in panel H. Dented white line illustrates an example line used to quantify normalized intensities across cell–cell contacts shown in panels K–M. Solid circle lines enclose the region we defined as perinuclear. The region between the 2 circles corresponds to a typical ROI used for quantification in panel N. (K–M) Quantification of normalized fluorescence intensity across notochord cells of *Ciiinte. Brac>mCherry::CaMK; Ciiinte.Brac>hCD4::GFP* double transgenic embryos. Green peaks highlighted with black arrowheads in panel K correspond to cell–cell membrane contacts. The Pearson's correlation coefficients (r) values are shown. The LL and UL 95% confidence intervals were as follows: negative control (LL: 0.8922, UL: 0.9448), Ano10^CRISPR (LL: 0.3748, UL: 0.6195), and rescue embryos (LL: 0.9623, UL: 0.9797). (N) Perinuclear intensity levels of *mCherry::CaMK* across the 3 genotypes tested in our assay (see also S36 Table). (O–Q) Maximal projection confocal micrographs of (O) DMSO, (P) CaMK inhibitor, and (Q) CaM inhibitor-treated stage 23 embryos expressing *Ciiinte.Brac >NUP50::RFP; Ciiinte.Brac >hCD4GFP* marking the nuclei and plasma membranes of notochord cells, respectively. All embryos are oriented with anterior to the left. Lateral views shown. Scale bars = 30 μm. (R–U) Quantifications of notochord intercalation metrics for embryos treated with either DMSO, CaM, or CaMK inhibitors. (R) Quantification of notochord length. For statistical analysis, we performed a Kruskal–Wallis test followed by a Dunn's multiple comparisons test. We assayed 50≤n≤67 animals per condition (see also S37 Table). (S) Notochord cell medial protrusion angles. For statistical analysis, we performed a Kruskal–Wallis test followed by a Dunn's multiple comparisons test; 40≤n≤54 animals, 200≤n≤221 cells per condition (see also S38 Table). (T) Intercalation rates. Data presented as mean ± SEM, 24≤n≤40 animals per genotype. For statistical analysis, we used a mixed-effects model (REML), followed by a Tukey's multiple comparisons test (see also S39 Table). (U) Notochord cell misalignment along the A/P midline. Data presented as mean ± SEM, 15≤n animals per genotype. For statistical analysis, we used a mixed-effects model (REML), followed by a Tukey's multiple comparisons test (see also S40 Table). The numerical data underlying this figure can be found in 10.5281/zenodo.12506448.

5H, and 5K). In addition, CaM was enriched inside the nucleus of intercalating cells, while CaMK was enriched around the nucleus (Fig 5A, 5G, 5H, and 5N and S35 and S36 Tables). We wondered whether CaM and CaMK localization was dependent on Ano10. In Ano10-^CRISPR embryos undergoing intercalation, expression of mCherry::CaM and mCherry::CaMK along the cell–cell contacts was reduced and the correlation with the PM marker was markedly reduced relative to negative controls (Fig 5B, 5E, 5I, and 5L). Simultaneously, the nuclear localization of mCherry::CaM and the perinuclear localization of mCherry::CaMK were significantly down-regulated (Fig 5B, 5G, 5I, and 5N and S35 and S36 Tables). These localization phenotypes could be rescued by expressing a CRISPR/Cas9 resistant form of *Ano10* cDNA in the notochord, suggesting that Ano10 is able to cell-autonomously regulate CaM and CaMK localization (Fig 5C, 5F, 5G, 5J, 5M, and 5N and S35 and S36 Tables).

Subsequently, we wondered whether CaM and CaMK are required for notochord intercalation. Using pharmacological inhibitors from the onset of intercalation, we observed that inhibition of CaM and CaMK resulted in stubby embryos with highly irregular notochords as compared to DMSO controls (Fig 5O–5Q). At the end of convergent extension, the notochords of DMSO-treated embryos reached a wild-type mean length (372.1 μm) but CaM (180.3 μm) and CaMK (231.7 μm) inhibitor-treated embryos were significantly shorter (Fig 5R and S37 Table). In comparison to DMSO controls (33.30˚) intercalating cells, both CaM (46.30˚) and CaMK (51.46˚) inhibited cells were characterized by a larger mean value of the angle enclosed by the notochord cell protrusion (Fig 5S and S38 Table). CaM and CaMK inhibited embryos exhibited significantly slower medio-lateral intercalation rates compared to DMSO control embryos (Fig 5T and S39 Table). Finally, we quantified the defects in notochord morphology at the end of convergent extension due to erroneous intercalation. In DMSO control embryos where the notochord had successfully completed the intercalation

process the distance of each cell center relative to the midline of the skeleton was very short (Fig 5U and S40 Table). In contrast, most of the notochord cells in the CaM and CaMK inhibitor-treated embryos were positioned significantly further away from the midline skeleton (Fig 5U and S40 Table). Our analysis suggests that CaM and CaMK are required for robust convergent extension.

Following our analysis of CaM and CaMK during convergent extension, we explored the localization and role of these 2 molecules during tubulogenesis. Treatment of tailbud embryos that had just completed cell elongation with either a CaM or a CaMK inhibitor resulted in smaller or no lumen in contrast compared to the well-formed, continuous lumen of DMSO-treated controls (Fig 6A–6C). In CaM and CaMK inhibitor-treated embryos, the lumen curvature measured from transverse notochord sections was significantly higher compared to DMSO controls cells suggesting that they had a narrower lumen compared to controls (Fig 6D and S41 Table). We found that the lumen longitudinal radius at the end of the expansion phase was significantly shorter in CaM and CaMK inhibitor-treated embryos compared to DMSO controls (Fig 6E and S42 Table). Finally, the cross-sectional area of the lumen was significantly smaller in drug-treated animals compared to DMSO-treated controls (Fig 6F and S43 Table). Taken together, our data suggest that CaM and CaMK are required for lumen elongation and connection.

Having established the contributions of CaM and CaMK to tubulogenesis, we went on to characterize the subcellular localization of these 2 molecules in the same period, in control and Ano10$^{CRISPR}$ genetic backgrounds. We found that in control genetic background CaMK shows perinuclear localization as well as an enrichment along the apical domains of the notochord cells during lumen extension and bidirectional crawling (Fig 6G and 6J). We note that CaMK showed little to no enrichment at the leading edges in cells undergoing bidirectional crawling (Fig 6J). Both the apical domain and perinuclear enrichment were eliminated in *Ano10$^{CRISPR}$* notochord cells which were attempting to perform lumen extension (Fig 6H, 6K, and 6S and S44 Table). We were able to rescue in a notochord-specific manner the CaMK localization phenotypes by expressing a CRISPR/Cas9 resistant form of *Ano10* cDNA, suggesting that CaMK localization is regulated by Ano10 (Fig 6I, 6L, and 6S and S44 Table).

CaM, the second Ca$^{2+}$ sensor we characterized, showed strong enrichment in the nucleus and at the apical domains of expanding lumens (Fig 6M). The nuclear localization of CaM persisted throughout tubulogenesis, but the most striking observation we made during this phase of notochord development was the strong recruitment of CaM to the leading edges of the bidirectionally crawling cells (Fig 6P and 6U and S46 Table), which suggest that CaM may be involved in regulating the directional movement of notochord cells. In *Ano10$^{CRISPR}$* embryos, attempting lumen extension the nuclear and apical localization of CaM was reduced (Fig 6N). In *Ano10$^{CRISPR}$* notochord cells the enrichment of CaM at the leading edges was abrogated and the nuclear localization was significantly reduced (Fig 6Q, 6T, and 6U and S45 and S46 Tables). Importantly, the localization phenotypes of CaM throughout tubulogenesis could be rescued by the expression of a CRISPR/Cas9 resistant form of *Ano10* cDNA (Fig 6O, 6R, 6T, and 6U and S45 and S46 Tables).

## Ano10 and diverse "encoders" and "decoders" of Ca$^{2+}$ signals modulate Ca$^{2+}$ activity in the notochord during CE and tubulogenesis

Our data show that Ca$^{2+}$ as a second messenger and the underlying molecular machinery to both "encode" and "decode" Ca$^{2+}$ signals are required during notochord convergent extension and tubulogenesis. We also find that Ano10 cooperates with this machinery to ensure the robust development of *Ciona*'s notochord. Importantly, Anoctamins have been shown to

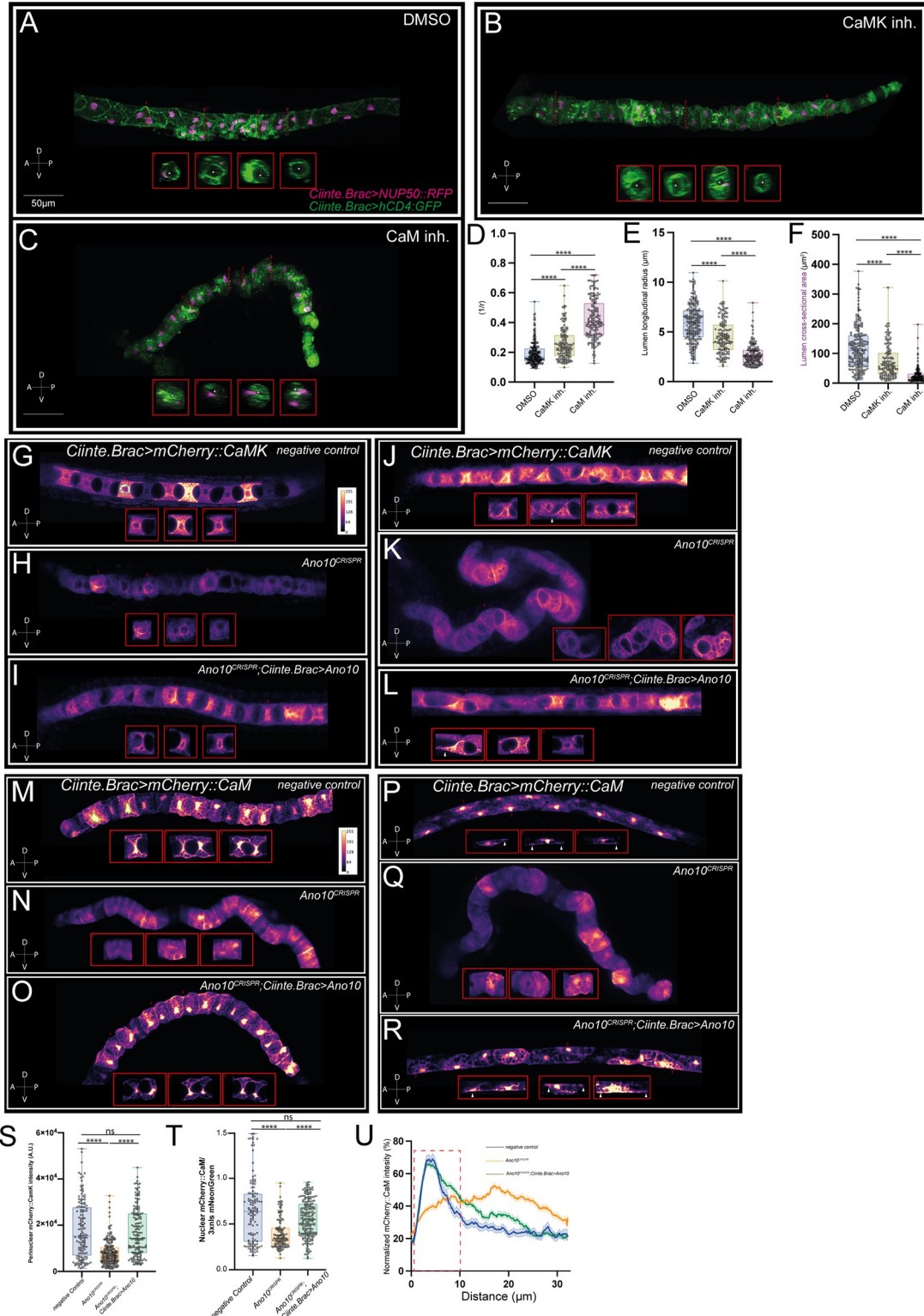

**Fig 6. The Ca²⁺ sensors CaM and CaMK modulate lumen formation, while their subcellular localization is modulated by Ano10.** (A–C) Maximal projections of stage 26 *Cionas* (notochord development stage VII) incubated with 5 μm DMSO, 50 μm CaMK, or 50 μm CaM inhibitors. Transgenic animals are labeled using *Ciinte.Brac> NUP50::RFP; Ciinte.Brac>hCD4::GFP* marking the nuclei and plasma membranes of notochord cells, respectively. Anterior is to the left of the image and lateral views are shown. The insets show transverse sections of the notochord along position marked with the dashed red lines and labeled 1 to

4. Asterisks indicate the notochord lumen. (D–F) Quantification of lumen metrics across negative controls, *Ano10*^CRISPR^ and rescue animals. (D) Lumen curvature; (E) lumen longitudinal radius; (F) lumen cross-sectional area. For statistical analysis of data presented in panels D–F, we performed a Kruskal–Wallis test followed by a Dunn's multiple comparisons test. We used $33 \leq n \leq 45$ animals, $142 \leq n \leq 214$ cells per condition (see also S41–S43 Tables). (G–L) Sum projections of confocal stacks from negative control, *Ano10*^CRISPR^ and rescue animals expressing *Ciinte.Brac> mCherry::CaMK*. In these examples, the notochord cells are undergoing either lumen expansion (G–I) or bidirectional crawling (white arrowheads point to examples of leading edges) (J–L). Images are pseudocolored with mpl-plasma. The fluorescence intensity calibration scale bar is shown in panel G. Insets correspond to median confocal sections correspond to the cells marked with numbers 1–3 in each panel. Lateral view of the embryos is shown. (M–R) Representative sum confocal projections of stacks from negative control, *Ano10*^CRISPR^ and rescue animals expressing *Ciinte.Brac> mCherry::CaM*. In these examples, the notochord cells are undergoing either lumen expansion (M–O) or bidirectional crawling (white arrowheads point to examples of leading edges) (P–R). Images are pseudocolored with mpl-plasma. The fluorescence intensity calibration scale bar is shown in panel M. Insets correspond to median confocal sections correspond to the cells marked with numbers 1–3 in each panel. (S) Quantification of perinuclear *mCherry::CaMK* during lumen extension across the 3 genotypes. For statistical analysis, we performed a Kruskal–Wallis test followed by a Dunn's multiple comparisons test. We used $28 \leq n \leq 45$ animals, $144 \leq n \leq 225$ cells per condition (see also S44 Table). (T) Quantification of the nuclear fraction of *mCherry::CaM* relative *3xnls::mNeonGreen* in cells undergoing bidirectional crawling. For statistical analysis, we performed a Kruskal–Wallis test followed by a Dunn's multiple comparisons test. We used $24 \leq n \leq 39$ animals, $122 \leq n \leq 196$ cells per condition (see also S45 Table). (U) Quantification of normalized *mCherry::CaM* line profiles along the basal domain of cells undergoing bidirectional crawling across the 3 genotypes in this assay. Data presented as mean ± SEM. Dashed red box marks the location of the leading edge. For statistical analysis, we performed a mixed-effects model (REML) test followed by a Tukey's multiple comparisons test. We used $10 \leq n \leq 26$ animals, $51 \leq n \leq 133$ cells per condition (see also S46 Table). The numerical data underlying this figure can be found in 10.5281/zenodo.12506448.

modulate intracellular $Ca^{2+}$ dynamics across a range of tissues [62–65]. Thus, we asked whether $Ca^{2+}$ activity in the notochord is modulated by Ano10.

To image long-term changes in $Ca^{2+}$ signaling, we used the photoconvertible Genetically Encoded Integrator CAMPARI2, which can integrate $Ca^{2+}$ levels over extended periods of time [86]. Photoconversion of green fluorescence to red fluorescence in the presence of UV light and $Ca^{2+}$ ions means that CAMPARI2 can act as a ratiometric integrator, since the Red/Green ratio values correlate with the levels of accumulated $Ca^{2+}$ activity in the cell/tissue. We imaged embryos expressing CAMPARI2 in the notochord either at the onset of convergent extension or tubulogenesis (pre data points), we then illuminated the embryos periodically with low levels of UV light and imaged again at end of convergent extension or tubulogenesis (post data points) (Figs 7A–7H and S7A–S7L). We quantified the Red/Green ratio (R/G) of negative control, *Ano10*^CRISPR^ and rescue embryos (Fig 7G and 7H and S47 and S48 Tables). As expected, before photoconversion we obtained a low R/G across all conditions (Fig 7G and 7H and S47 and S48 Tables). Following photoconversion at the end of convergent extension or tubulogenesis negative controls showed a large increase in R/G ratio suggesting that the notochord cells have exhibited high $Ca^{2+}$ activity during these developmental windows (Figs 7A, 7D, 7G, 7H, S7A, S7B, S7G, and S7H and S47 and S48 Tables). *Ano10*^CRISPR^ embryos exhibited an increase in R/G ratio; however, this was significantly smaller than that of negative controls or rescues indicating that loss of Ano10 results in reduced $Ca^{2+}$ activity over these 2 developmental windows (Figs 7B, 7E, 7G, 7H, S7C, S7D, S7I, S7J, and S47 and S48 Tables). Importantly, we were able to rescue the *Ano10*^CRISPR^ defects by expressing a Crispr/Cas9 resistant variant of *Ano10* cDNA (Figs 7C, 7F, 7G, 7H, S7E, S7F, S7K, and S7L and S47 and S48 Tables).

In addition, we performed volumetric $Ca^{2+}$ imaging to capture $Ca^{2+}$ dynamics in notochord cells by expressing the Genetically Encoded Calcium Indicator (GECI) GCaMP8f [87] under the Brachyury promoter in different genetic (negative control, *Ano10*^CRISPR^ and *Ano10*^CRISPR^ *Ciinte.Brac>Ano10*) and pharmacological backgrounds. While we were able to record some $Ca^{2+}$ activity during notochord intercalation the frequency of intercalating cells exhibiting $Ca^{2+}$ transients were very low under our imaging conditions which precluded further meaningful analysis between control, *Ano10*^CRISPR^ and drug-treated embryos. Instead, we focused on the last stage of tubulogenesis, namely the bidirectional crawling phase which was

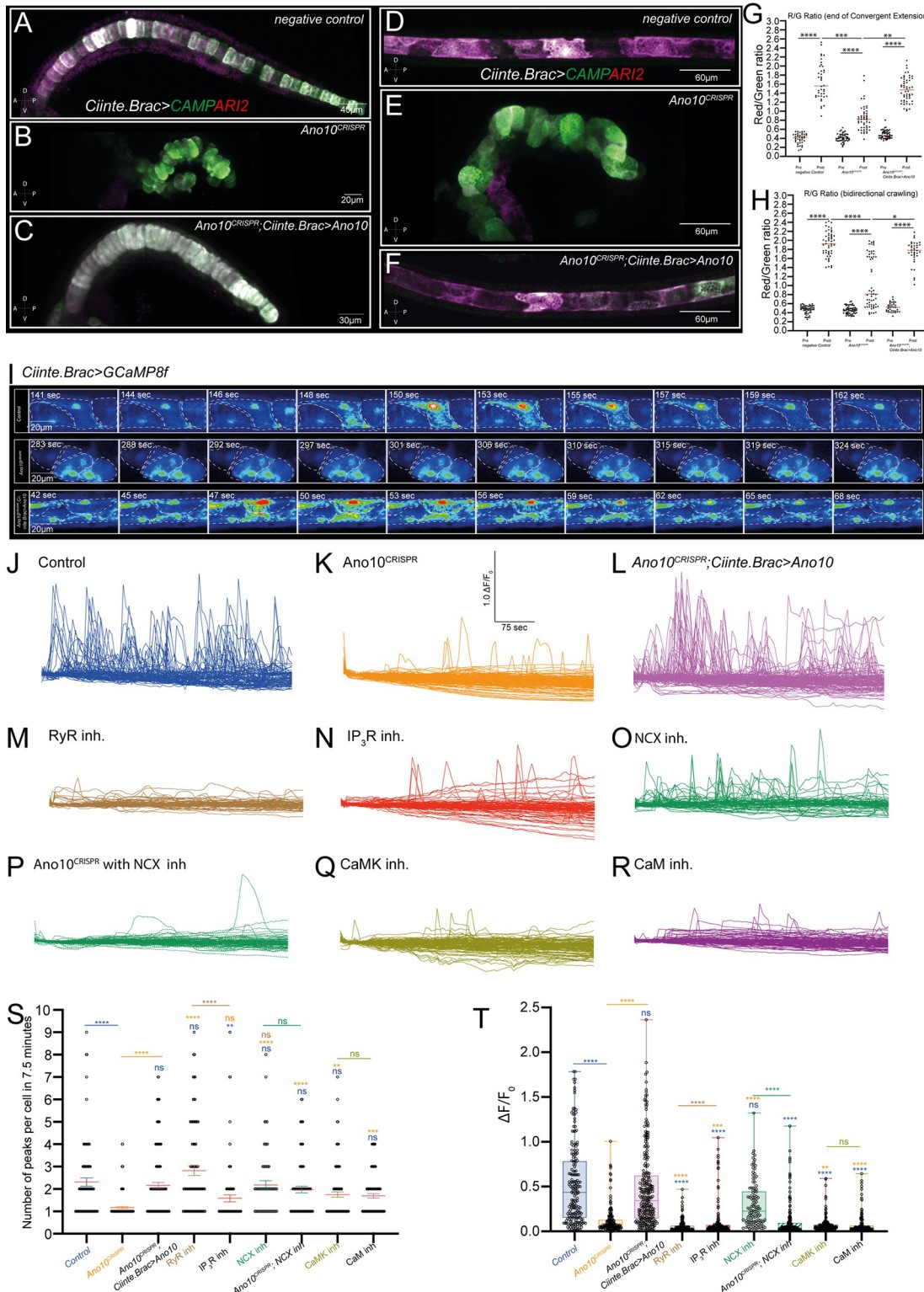

**Fig 7. Ano10 modulates Ca²⁺ dynamics during convergent extension, lumen formation, and extension.** (A–F) Composite sum projections embryos of representative negative control, *Ano10*$^{CRISPR}$ and rescue embryos expressing the Ca²⁺ integrator CAMPARI2 in the notochord, following photoconversion pulses during convergent extension (A–C) and during tubulogenesis (D–F) Anterior to the left, lateral views shown. (G, H) Quantification of Red/Green ratio of CAMPARI Ca²⁺ integrator before (pre) and after (post) photoconversion at the end of CE (G) and tubulogenesis (H). For statistical analysis of data in panels G and

H, we performed a Kruskal–Wallis test followed by Dunn's multiple comparisons test (see also S47 and S48 Tables). We assayed $40 \leq n \leq 52$ for data shown in panel G and $36 \leq n \leq 57$ for data shown in panel H. (I) Montage of 3 time-lapse movies from different embryos expressing *Ciinte.Brac>GCaMP8f* in 3 different genetic backgrounds top: negative control, middle: Ano10CRISPR, bottom: Ano10$^{CRISPR}$ *Ciinte.Brac>Ano10*. Movies were captured during the bidirectional crawling phase. Different cells are outlined with the dashed white lines (S4–S6 Movies). (J–R) Normalized Ca$^{2+}$ traces from (J) Control (5 µm DMSO), (K) Ano10$^{CRISPR}$, (L) Ano10$^{CRISPR}$ *Ciinte.Brac>Ano10*, (M) RyR inhibitor, (N) IP3R inhibitor, (O) NCX inhibitor, (P) Ano10$^{CRISPR}$; NCX inhibitor, (Q) CaMK inhibitor, and (R) CaM inhibitor notochord cells undergoing bidirectional crawling. (S) Quantification of the number of Ca$^{2+}$ peaks per cell analyzed. Each recording lasted 7.5 min. For statistical analysis, we performed a Kruskal–Wallis test followed by Dunn's multiple comparisons test (see also S49 Table). (T) Quantification of Ca$^{2+}$ peak amplitude. For statistical analysis, we performed a Kruskal–Wallis test followed by Dunn's multiple comparisons test (see also S50 Table). We imaged $6 \leq n \leq 13$ animals and $61 \leq n \leq 137$ cells per condition. The significance notation above each box plot is color-coded according to the experimental condition that it is compared to (e.g., when compared to control asterisks or ns are colored blue). The numerical data underlying this figure can be found in 10.5281/zenodo.12506448. NCX, Na+/Ca2+ exchanger.

characterized by robust Ca$^{2+}$ activity in control animals (Fig 7I, 7J, 7S, and 7T and S49 and S50 Tables and S4 Movie). An interesting observation we made is that negative control and rescue animals Ca$^{2+}$ signals can be transmitted across neighboring notochord cells via the leading and trailing edges, which can act as the site of origin for Ca$^{2+}$ waves within a cell (Figs 7I and S7M and S4 and S6 Movies). In *Ano10*$^{CRISPR}$ animals the frequency and amplitude of Ca$^{2+}$ transients were significantly reduced compared to controls (Fig 7I, 7K, 7S, and 7T and S49 and S50 Tables and S5 Movie). We were able to rescue the Ano10 loss of function defects by expressing a CRISPR/Cas9 resistant form of *Ano10* cDNA under the control of the *Brachyury* promoter (Fig 7I, 7L, 7S, and 7T and S49 and S50 Tables and S6 Movie).

Furthermore, we tested the impact of inhibiting the Ryanodine receptor, the IP$_3$ receptor, the NCX Na$^{+}$/Ca$^{2+}$ exchanger, as well as the CaMK and CaM molecules on the ability of the notochord cells to generate robust Ca$^{2+}$ activity during bidirectional crawling.

RyR inhibition led a significant decrease in the amplitude of Ca$^{2+}$ transients relative to controls and a small but not significant increase in the frequency Ca$^{2+}$ transients. The amplitude and frequency of Ca$^{2+}$ transients in RyR inhibitor-treated animals were significantly different to those obtained from Ano10$^{CRISPR}$ animals (Fig 7M, 7S, and 7T and S49 and S50 Tables). IP$_3$R inhibition resulted in a significant reduction in both amplitude and frequency of Ca$^{2+}$ transients compared to controls. We note that the Ca$^{2+}$ transient metrics of IP$_3$R inhibited notochord cells were statistically indistinguishable from those of *Ano10*$^{CRISPR}$ cells (Fig 7N, 7S, and 7T and S49 and S50 Tables).

We found that inhibition of NCX in a wild-type background did not result in a significant change in the frequency or amplitude of Ca$^{2+}$ transients (Fig 7O, 7S, and 7T and S49 and S50 Tables). In contrast, inhibition of NCX in an Ano10$^{CRISPR}$ background led to a significant reduction in the amplitude of Ca$^{2+}$ transients relative to control and Ano10$^{CRISPR}$ (Fig 7P, 7S, and 7T and S49 and S50 Tables). To our surprise, addition of NCX inhibitor in Ano10$^{CRISPR}$ animals "rescued" the frequency of Ca$^{2+}$ transients (Fig 7P, 7S, and 7T and S49 and S50 Tables). This observation suggests that Ano10 and NCX work in parallel to modulate the amplitude of Ca$^{2+}$ transients but they act antagonistically to control the frequency of Ca$^{2+}$ transients in bidirectionally crawling notochord cells.

Finally, inhibition of CaMK and CaM did lead to a small but not statistically significant reduction in Ca$^{2+}$ transient frequency (Fig 7Q, 7R, 7S, and 7T and S49 and S50 Tables). However, inhibition of these 2 important cellular Ca$^{2+}$ sensors resulted in a strong decrease in the amplitude of Ca$^{2+}$ transients that was significantly stronger than that observed in control and Ano10$^{CRISPR}$ animals (Fig 7Q, 7R, 7S, and 7T and S49 and S50 Tables). These data suggest that CaMK and CaM are key regulators of Ca$^{2+}$ transient amplitude but not Ca$^{2+}$ transient frequency in bidirectionally crawling notochord cells.

## Ano10 regulates cell behavior and cytoskeletal organization during tubulogenesis

We wondered whether the defect in lumen tilting and connection exhibited by *Ano10*<sup>CRISPR</sup> notochords could be attributed to perturbed morphology and locomotion of notochord cells. A characteristic morphological transition associated with the crawling behavior that notochord cells exhibit during lumen expansion and tilting is the development of protruding anterior or posterior leading edges (ALE, PLE) composed of lamellipodia (LA) and new extensions (NE) (Fig 8A and 8B) [19]. We observed that in confocal micrographs *Ano10*<sup>CRISPR</sup> notochord cells were rounder, characterized by truncated leading edges that were often missing lamellipodia and had smaller new extensions (NE) (Fig 8C and 8D). This phenotype could be rescued in a notochord-specific manner (Fig 8E and 8F). Quantification of the length of the ALE region across negative controls, *Ano10*<sup>CRISPR</sup> and rescue notochord cells showed that the ALE in *Ano10*<sup>CRISPR</sup> cells was significantly shorter than negative controls (Fig 8G and S51 Table). We also quantified the aspect ratio of these cells and found that *Ano10*<sup>CRISPR</sup> cells had a significantly lower aspect ratio (Fig 8H and S52 Table), indicating that they could not switch from the cylindrical shape they had at the onset of tubulogenesis, to the elongated endothelial-like flat cells. Interestingly, we found that these morphological defects were associated with the markedly slower movement of *Ano10*<sup>CRISPR</sup> cells during tubulogenesis (Fig 8I and S53 Table). The successful notochord-specific rescue of the *Ano10*<sup>CRISPR</sup> defects suggests that Ano10 is required in a cell autonomous manner to mediate several cell behaviors occurring during the lumen tilting and connection.

Previous work has shown that cortical actin is essential for lumen formation and that polarized microtubules contribute to lumen development by forming a network that is actin dependent. At later stages when notochord cells perform bidirectional crawling the microtubule network is organized towards the leading edges of the ALE and PLE, a process which is essential to organize the actin-based protrusions of the leading edge [17]. To assess whether actin and microtubule organization was perturbed in *Ano10*<sup>CRISPR</sup> notochord cells, we imaged and analyzed the organization the microtubule binding protein ensconsin-3xGFP, the microtubule plus-end marker EB3-mNeonGreen and the actin marker LifeAct-tdTomato throughout tubulogenesis.

We found that at the onset of lumen formation microtubules labeled with ensconsin-3xGFP showed a bias for radial distribution in negative controls (Fig 8J-8J" and 8M and S54 Table), arrayed mostly parallel to the A/P axis, with a higher concentration at the lateral surface (Fig 8J and 8J") where the lumen emerges, as previously shown [17]. Circumferential non-radial microtubules were also present across the different domains of the cell (Fig 8J' and 8J"). In *Ano10*<sup>CRISPR</sup> notochord cells microtubules were strongly disrupted (Fig 8K'-8K'"). They formed fewer but thicker bundles with an increase in the abundance of circumferential non-radial bundles (Fig 8K-8K"). Notochord-specific expression of a rescue construct in *Ano10*<sup>CRISPR</sup> notochord cells rescued the mutant phenotype (Fig 8L-8L"). Furthermore, we quantified the fraction of microtubules at the basal domains (Fig 8N and S55 Table). We found that this fraction is higher in *Ano10*<sup>CRISPR</sup> notochord cells compared to negative controls and rescue cells (Fig 8N and S55 Table). Subsequently, we monitored and quantified microtubule dynamics during bidirectional crawling using the microtubule plus end marker EB3-mNeonGreen (Fig 8O-8Q). *Ano10*<sup>CRISPR</sup> notochord cells showed a significantly lower enrichment for EB3 comets in the ALE and PLE regions as compared to negative control and rescue cells (Fig 8R and 8S and S56 Table).

Subsequently, we analyzed the subcellular organization of the actin marker tdTomato-LifeAct across the 3 genotypes used in our study (Fig 8T–8V). At the onset of tubulogenesis

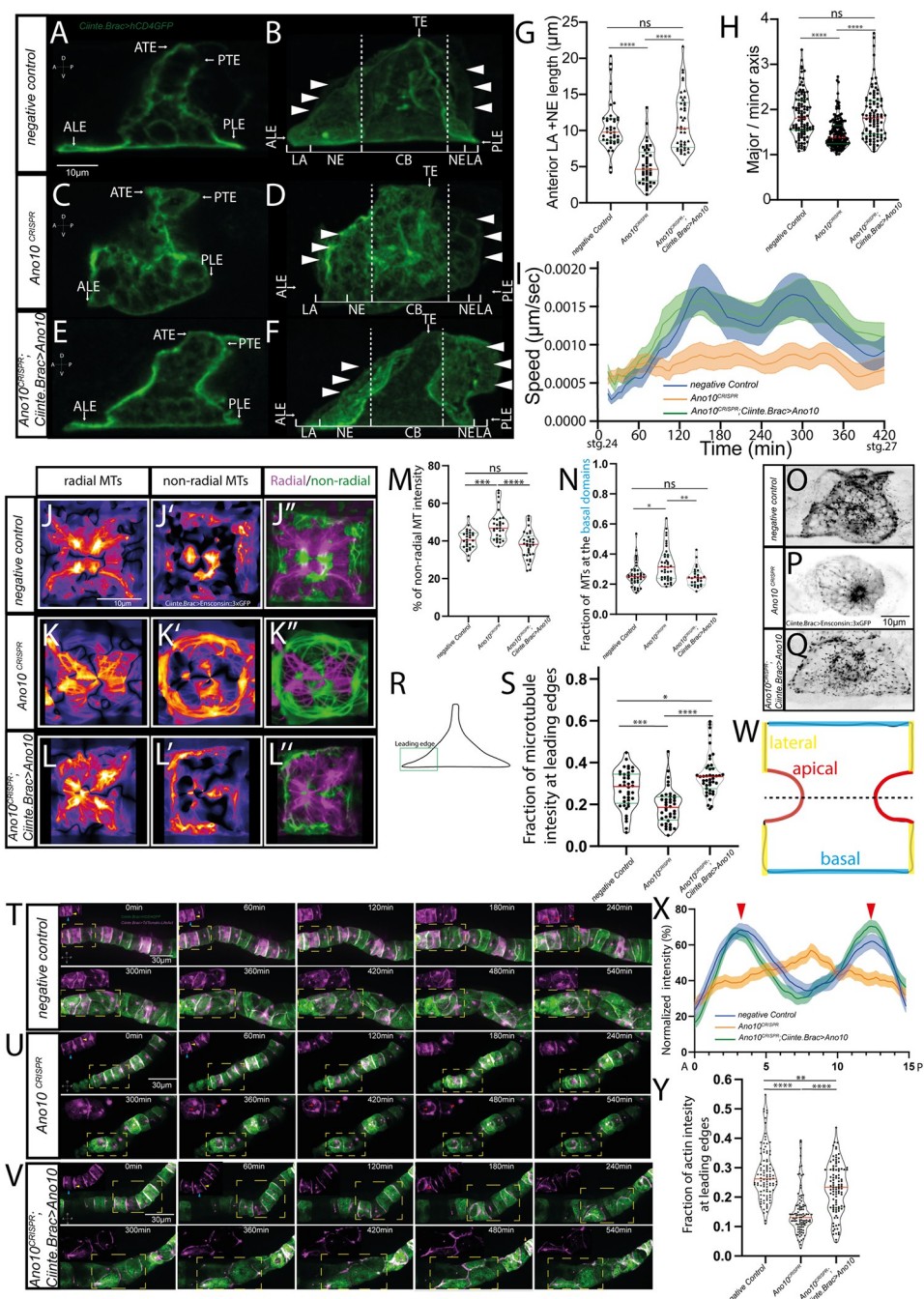

**Fig 8. Loss of Ano10 perturbs cell morphology, behavior, and the cytoskeletal dynamics during tubulogenesis.** (A–F) Confocal images of notochord cells expressing plasma membrane targeted GFP. A, C, E show medial sections while B, D, F show maximal projections of the same cells. In stage VI, the ATE and the PTE retreat along the A/P axis, in contrast to the ALE or the PLE which extend along the A/P axis. Landmarks in the cells were determined in accordance with Dong and colleagues [19]. The dashed lines in B, D, F contain the space occupied by the cells prior to the formation of protrusions. White arrowheads show the edges of the lamellipodia, which have wavy outlines. LA, lamellipodia; NE, new cell extension; TE, trailing edge; CB, main cell body. (G) Quantification of Anterior cell protrusion as defined by the combined length of the LA and NE regions. We assayed 10 animals and 42≤n cells per genotype. (H) Quantification of the aspect ratio of notochord cells at stage VI. We assayed 23≤n≤44 animals and 93≤n≤179 cells per genotype. (I) Evolution of apical domain speed from lumen initiation until the lumen connection is complete. Data presented as mean ± SEM. (*n* = cells used in analysis). For panel I, we used a mixed-effects model (REML) followed by Tukey's multiple comparisons test (see also S53 Table). We assayed 6≤n≤11 animals and 25≤n≤45 cells per genotype. (J–L) Lateral views of single notochord cells expressing *Ciinte.Brac> Ensconsin::3xGFP*

where radial (J, K, L), non-radial (J', K', L') MTs are highlighted using the FIJI plugin Radiality Map. A merge of non-radial and radial MTs is shown in (J", K", L"). (M) Quantification of non-radial MT intensity. We assayed $7 \leq n \leq 8$ animals and $28 \leq n \leq 34$ cells per genotype. (N) Quantification of the MTs localized at the basal domains of the notochord cells. We assayed $8 \leq n \leq 10$ animals and $28 \leq n \leq 40$ cells per genotype. (O–Q) Confocal maximal projection of a notochord cell during crawling from (O) negative control, (P) $Ano10^{CRISPR}$ and rescue (Q) embryos expressing *Ciinte.Brac>EB3::mNeonGreen*, which allows us to visualize the microtubule plus ends. (R) The schematic illustrates a crawling notochord cell. The leading-edge area is highlighted by a green box. (S) Quantification of the fraction of microtubule intensity at the leading edges. We assayed 10 animals per genotype and $40 \leq n \leq 43$ cells per genotype. (T–V) Montage of notochords from (T) negative control, (U) $Ano10^{CRISPR}$ and (V) rescue embryos expressing *Ciinte. Brac>hCD4GFP; Ciinte.Brac>LifeAct:: tdTomato* during tubulogenesis. Insets show the signal from LifeAct::tdTomato corresponding the region highlighted by the yellow dashed lines. The developing notochords are oriented with anterior to the left, lateral views shown. Yellow arrowheads point to lateral domains, blue arrowheads highlight the equatorial region of the basal cortex, red arrowheads point to the apical domains during lumen expansion, and brown arrowheads during lumen tilting and connection. (W) Schematic of a notochord cell during lumen extension. We highlight in red the apical domains, in yellow the lateral domains, and in blue the basal domains. (X) Normalized actin profile across notochord cells (corresponding to the dashed black line in W) imaged during lumen extension. Red arrowheads indicate the points that correspond to the apical domains. Data presented as mean ± SEM. For statistical analysis, we used a mixed-effects model (REML) followed by Tukey's multiple comparisons test (see also S57 Table). We assayed $9 \leq n \leq 14$ animals and $36 \leq n \leq 59$ cells per genotype. (Y) Quantification of the fraction of actin intensity at the leading edges, in notochord cells expressing *Ciinte.Brac>LifeAct::tdTomato* in different genetic backgrounds. In all violin plots in this figure the red line indicates the median, while the green lines indicate the quartiles. We assayed 25 animals and 100 cells per genotype. For statistical analysis of the data shown on all violin plots, we performed Kruskal–Wallis tests followed by Dunn's multiple comparisons tests (see also S51, S52, S54–S56, and S58 Tables). The numerical data underlying this figure can be found in 10.5281/zenodo.12506448. ALE, anterior leading edge; ATE, anterior trailing edge; PLE, posterior leading edge; PTE, posterior trailing edge.

negative control $Ano10^{CRISPR}$ and rescue notochord cells, showed similar localization of actin at the lateral surfaces of notochord cells. However, the actin levels at the equatorial region of the basal cortex were lower in $Ano10^{CRISPR}$ cells. During lumen formation, negative control and rescue cells showed extensive actin reorganization around the apical domain, in contrast to $Ano10^{CRISPR}$ cells which showed delayed recruitment of actin to the apical domain. We compared the actin distribution during lumen initiation (Fig 8W and 8X and S57 Table) by quantifying actin intensity in the median confocal sections (Fig 8W and 8X and S57 Table). We found that in negative controls actin is enriched in the apical/luminal domains as previously reported [17] (Fig 8X and S57 Table). In contrast, $Ano10^{CRISPR}$ notochord cells showed a more uniform distribution of actin with a small peak around the middle of the cells (Fig 8X and S57 Table). Ano10 rescue cells adopted a similar intensity profile to negative controls (Fig 8X and S57 Table). Finally, we measured the fraction of actin localized to the leading edges of cells that are bidirectionally crawling and we found that actin was enriched in the leading edges as previously observed [17] (Fig 8Y and S58 Table). The fraction of actin at the leading edges of the notochord cells was significantly reduced in $Ano10^{CRISPR}$ notochord cells (Fig 8Y and S58 Table). Expression of a rescue construct restored actin at the leading edges to negative control levels (Fig 8Y and S58 Table).

Given the importance of polarized actin and microtubules in the apical/lateral domains for lumen expansion our results raise the possibility that the $Ano10^{CRISPR}$ lumen expansion defect is in part due to deranged microtubule and actin localization. Similarly, the reduced presence of microtubules in the leading edge of $Ano10^{CRISPR}$ bidirectionally crawling cells suggests that the regulation exerted by Ano10 on the organization and dynamics microtubule and actin networks is important for notochord cell behaviors during tubulogenesis.

## Ano10 regulates RhoA and Cofilin localization during intercalation and tubulogenesis

To gain further insight on the putative mechanism by which Ano10 regulates the actin and microtubule dynamics of notochord cells, we turned our attention to 2 key regulators of

cytoskeletal dynamics the actin-binding protein cofilin and the signaling molecule RhoA GTPase [88–91] both of which have been implicated in notochord morphogenesis [18,21].

With the aim of determining the subcellular localization of RhoA during convergent extension and tubulogenesis we generated transient transgenic embryos expressing a translational fusion of RhoA (GFP::RhoA) in the developing notochord. During convergent extension the GFP::RhoA fusion strongly localized to 2 sites, the perinuclear region and the plasma membrane along the cell–cell contact domains of intercalating cells (Fig 9A, 9D, and 9G and S59 Table). At the cell–cell contact domains, GFP::RhoA showed strong positive correlation with an Lck::mScarlet PM marker (Fig 9D). In Ano10^CRISPR embryos undergoing intercalation, the localization of GFP::RhoA along the cell–cell contacts was weaker and the correlation with the PM marker was markedly reduced relative to negative controls (Fig 9B and 9E). In addition, the perinuclear localization of GFP::RhoA was significantly down-regulated (Fig 9B and 9G and S59 Table). We rescued these RhoA localization phenotypes by expressing a CRISPR/Cas9 resistant form of *Ano10* cDNA in the notochord, suggesting that Ano10 can cell-autonomously regulate RhoA localization in intercalating cells (Fig 9C, 9F, and 9G and S59 Table).

Following our analysis of RhoA localization during convergent extension, we explored its subcellular distribution during tubulogenesis. We found that in control animals GFP::RhoA was enriched at the lateral surfaces of notochord cells (Fig 9H, white arrowheads). This localization is similar to what has been previously reported in the case of actin filaments associated with adherens junctions in notochord cell interfaces [17]. We also noticed the presence of GFP:RhoA punctate clusters, which were closely associated with each expanding apical region during lumen expansion (Fig 9H, 9K, and 9N). However, when we repeated the experiment in a Ano10^CRISPR background, GFP::RhoA was mostly diffusely distributed throughout the cell cytoplasm with little to no plasma membrane enrichment even in instances where structures resembling lumens could be discerned (Fig 9I, 9L, and 9N). Using a CRISPR/Cas9 resistant form of *Ano10* cDNA, we were able to generate Ano10 rescue embryos, in which GFP::RhoA showed restored lateral surface localization and punctate cluster formation in association with the expanding luminal apical regions (Fig 9J, 9M, and 9N).

In the final step of tubulogenesis, bidirectionally crawling notochord cells form actin-rich lamellipodia at the anterior and posterior edges. We found that in control and Ano10 rescued notochord cells GFP::RhoA was enriched at these leading edge protrusions as well as around the nucleus (Fig 9O, 9Q, 9R, and 9S and S60 and S61 Tables). In *Ano10*^CRISPR embryos the GFP::RhoA enrichment at the leading edges and the perinuclear region was lost, with the GFP::RhoA signal showing a diffuse cytoplasmic distribution (Fig 9P, 9R, and 9S and S60 and S61 Tables).

Having established that RhoA localizes to subcellular locations that would be compatible with regulating the cytoskeleton of notochord cells during development, we turned our attention to the actin-binding protein cofilin. During convergent extension, the translational fusion of cofilin (mCherry::Cofilin) localized to the notochord cell–cell contacts, showing strong positive correlation with a hCD4::GFP plasma membrane marker similarly to RhoA (Fig 10A and 10D). In addition, Cofilin was very abundant within the nucleus of intercalating cells (Fig 10A and 10G and S62 Table). We wondered whether Cofilin localization was dependent on Ano10. In Ano10^CRISPR embryos undergoing intercalation, expression of mCherry::Cofilin along the cell–cell contacts was markedly reduced and the correlation with hCD4::GFP was significantly reduced relative to negative controls (Fig 10B and 10E). In Ano10^CRISPR embryos, the relative levels of cofilin in the nucleus were significantly down-regulated (Fig 10B and 10G and S62 Table). Both phenotypes were rescued with the expression of CRISPR/Cas9 resistant version of *Ano10 cDNA* under the Brachyury promoter (Fig 10C, 10F, and 10G and S62 Table).

During lumen initiation and extension Cofilin was localized along the apical and lateral domains of the notochord cells as well as in the equatorial region of the basal cortex (Fig 10H, 10K,

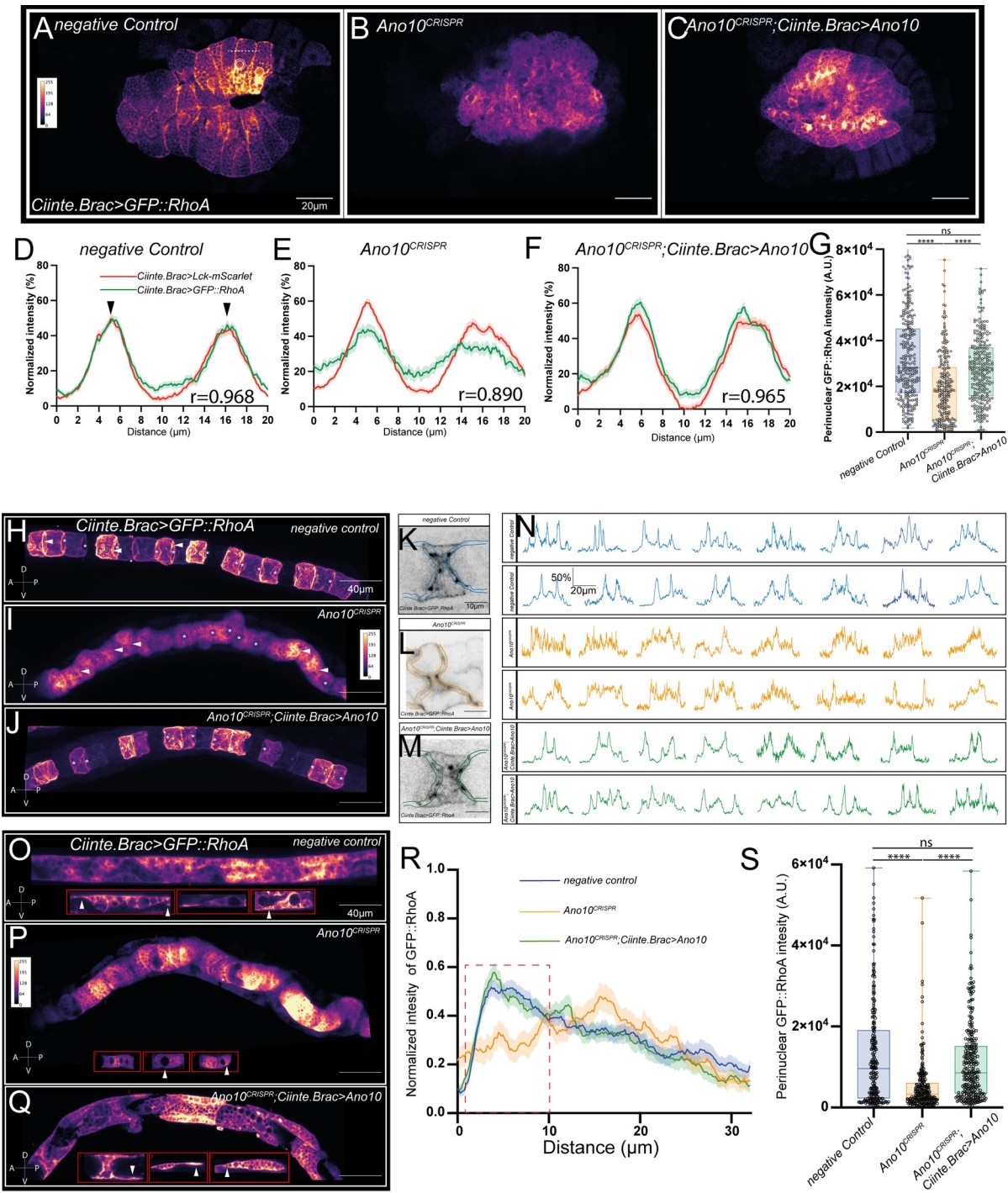

**Fig 9. The perinuclear and plasma membrane enrichment of RhoA is modulated by Ano10 during intercalation and tubulogenesis.** (A–C) Median confocal sections of 3 different genotypes: negative control, Ano10[CRISPR] and rescue embryos undergoing notochord intercalation. The embryos express *Ciinte.Brac>GFP::RhoA* and *Ciinte.Brac>Lck::mScarlet*; however, only the *GFP::RhoA* channel is shown for clarity. Images are pseudocolored with mpl-plasma LUT. The fluorescence intensity calibration scale bar is shown in panel A. Dented white line illustrates an example line used to quantify normalized intensities across cell–cell contacts shown in panels D–F. Solid white circle lines enclose the region we defined as perinuclear. The region between the 2 circles corresponds to a typical ROI used for quantification in panel G. Scale bar = 20 μm. Anterior to the left. (D–F) Quantification of normalized fluorescence intensity across notochord cells of *Ciiinte. Brac>GFP::RhoA; Ciinte.Brac>Lck::mScarlet*, double transgenic embryos. Red peaks highlighted with black arrowheads in panel D correspond to cell–cell membrane contacts. The Pearson's correlation coefficients (r) are shown. The lower (LL) and upper (UL) 95% confidence intervals were as follows: negative control (LL: 0.9560, UL: 0.9773), Ano10[CRISPR] (LL: 0.8508, UL: 0.9211), and rescue embryos

(LL: 0.9508, UL: 0.9752). We used 19≤n ≤25 animals and 87≤n≤126 cells per genotype. (G) Quantification of perinuclear GFP::RhoA in negative control, *Ano10^CRISPR* and rescue backgrounds. For statistical analysis of the data, we performed a Kruskal–Wallis test followed by a Dunn's multiple comparisons test. We assayed 49≤n animals and 249≤n cells per genotype (see also S59 Table). (H–J) Pseudocolored maximal projections of negative control, *Ano10^CRISPR* and rescue embryos expressing *Ciinte.Brac>GFP::RhoA* during lumen expansion. White arrowheads point to examples of lateral surfaces enriched in RhoA protein. Green arrowheads point to punctate clusters of GFP::RhoA found in close association with the expanding apical regions. Asterisks indicate examples of expanding lumens. Images are pseudocolored with mpl-plasma LUT. The fluorescence intensity calibration scale bar is shown in panel I. Scale bar = 40 μm. Anterior to the left. Lateral projections shown. (K–L) Representative example median confocal sections from negative control, Ano10^CRISPR and rescue notochord cells labeled with *Ciinte.Brac>GFP::RhoA*. The drawn lines indicate the thickness of the freehand lines drawn to obtain the line intensity profiles shown in panel N. Scale bar = 10 μm. (N) Normalized intensity line profiles of *Ciinte.Brac>GFP::RhoA* fluorescence across the apical region of notochord cells undergoing lumen expansion in negative control (blue), *Ano10^CRISPR* (orange), and rescue (green) embryos. (O–Q) Pseudocolored sum projections of *Ciinte.Brac>GFP::RhoA* confocal stacks in (O) negative control, (P) *Ano10^CRISPR*, (Q) and rescue genetic backgrounds. Insets 1–3 in each panel are median slices corresponding to the cells labeled 1 to 3. White arrowheads point to leading edges of notochord cells. Scale bar = 40 μm. Anterior to the left. Lateral projections shown. (R) Normalized intensity profiles of *Ciinte.Brac>GFP::RhoA* across the 3 genotypes along the basal domain of bidirectionally crawling cells. Dash red box marks the location of the leading edge along the x-axis. Medial slices were used for quantification. Data presented as mean ± SEM. We assayed 8≤n≤13 animals and 41≤n≤69 cells per genotype. For the data presented in panel R, we used a mixed-effects model (REML) followed by Tukey's multiple comparisons test (see also S60 Table). (S) Quantification of perinuclear GFP::RhoA across the 3 genotypes in notochord cells undergoing bidirectional crawling. We assayed 47≤n≤57 animals and 238≤n≤285 cells per genotype. For statistical analysis of data presented in panel S, we performed a Kruskal–Wallis test followed by Dunn's multiple comparisons test (see also S61 Table). The numerical data underlying this figure can be found in 10.5281/zenodo.12506448.

and 10L and S63 and S64 Tables). Interestingly, this region has been previously shown to be occupied by an actomyosin network [17]. We did not detect an obvious enrichment in the nucleus during that period in negative controls (Fig 10H). In *Ano10^CRISPR* embryos, we found that basal equatorial Cofilin and apical Cofilin levels were significantly reduced relative to negative controls (Fig 10I, 10K, and 10L and S63 and S64 Tables). These levels were restored in Ano10 rescued embryos (Fig 10J, 10K, and 10L and S63 and S64 Tables). During bidirectional crawling Cofilin was enriched at the leading edges and the nucleus of negative controls (Fig 10M, 10P, 10Q, and 10R and S65–S67 Tables). This enrichment in Cofilin was significantly reduced in *Ano10^CRISPR* animals (Fig 10N, 10P, 10Q, and 10R and S65–S67 Tables). By expressing a CRISPR/Cas9 resistant version of Ano10 *cDNA*, we were able to restore wild-type levels of Cofilin enrichment at both the leading edge and nucleus of *Ano10^CRISPR* rescued animals (Fig 10O, 10P, 10Q, and 10R and S65–S67 Tables).

## Ano10 functions as a channel and likely not as a scramblase

Next, we evaluated the ability of Ano10 to act as a phospholipid scramblase in cell culture. Phosphatidylserine (PS), a type of phospholipid, is normally constrained to the inner leaflet of the PM and gets exposed to the outer leaflet under various conditions and stimuli including large increases in $Ca^{2+}$ concentrations inside cells. Phospholipid scramblases including certain Anoctamin family members are responsible for the translocation of PS molecules between the inner and outer layers of the plasma membrane [41,49].

We expressed Ano10-GFP (Fig 11C, 11C', and 11C" and Table 1), or CAAX tag fused Ano10-GFP (targeting the protein to the PM) (Fig 11D, 11D' and 11D" and Table 1) in HEK293T cells and manipulated the intracellular $Ca^{2+}$ concentration with ionomycin. We then stained with AnnexinV-Alexa Fluor 568 antibody to detect phosphatidylserine activity. In both instances, we failed to observe phospholipid scramblase activity at higher levels compared to mock transfected controls (Fig 11A, 11A', and 11A" and Table 1). Our positive control mAno6 yielded robust scramblase activity (Fig 11B, 11B', and 11B" and Table 1). Importantly, during the process of screening *C. intestinalis* Anoctamins for scramblase we discovered that *Ciona* Ano6 displayed strong scramblase activity (Fig 11E, 11E', and 11E") indicating that *C. intestinalis* Anoctamins can be functional and show scramblase activity in a heterologous context.

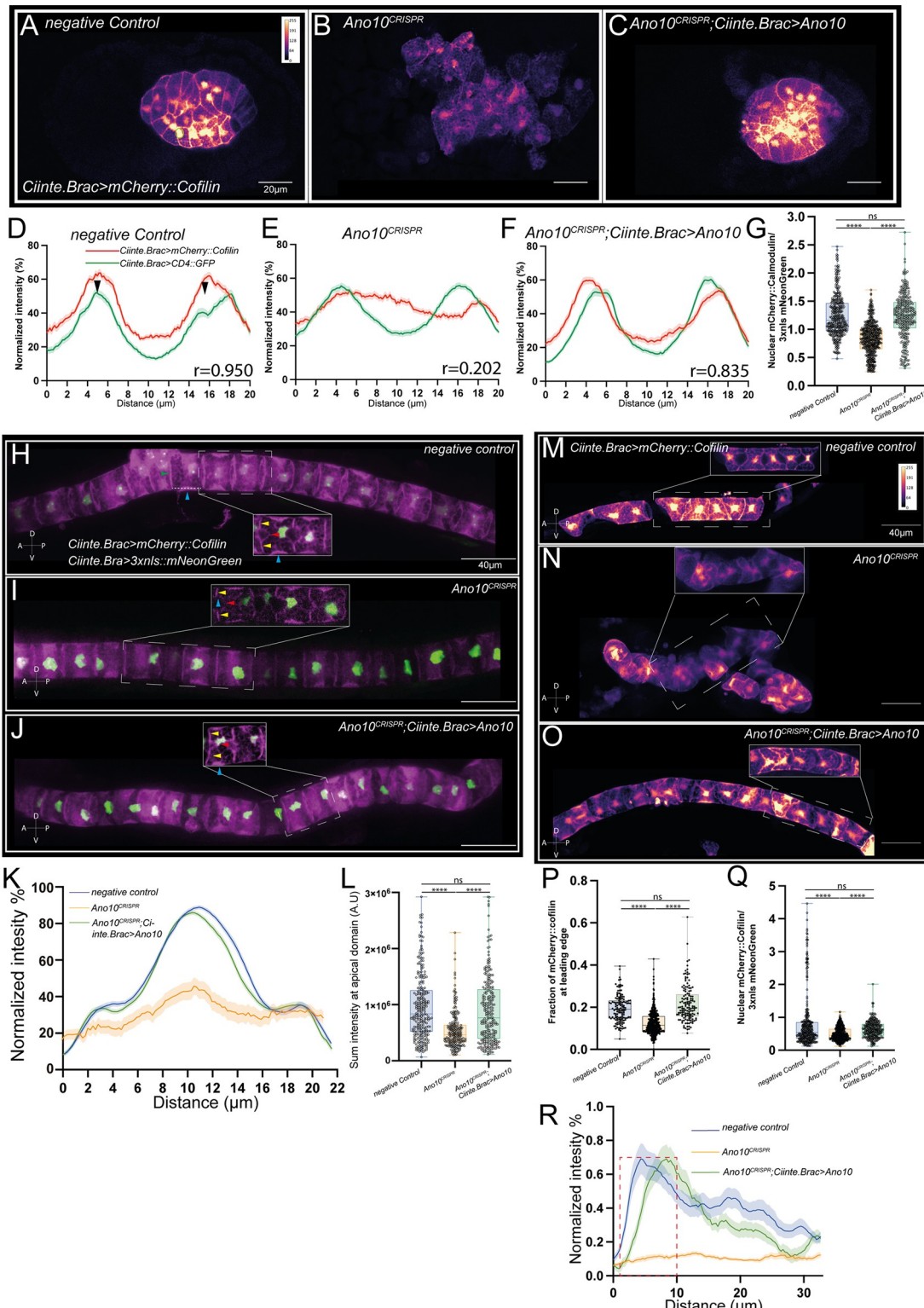

**Fig 10. Ano10 controls the subcellular localization of the actin-binding protein Cofilin during intercalation and tubulogenesis.** (A–C) Median confocal sections of 3 different genotypes: negative control, Ano10^CRISPR and rescue embryos undergoing notochord intercalation. The embryos express *Ciiinte.Brac> mCherry::Cofilin* and *Ciiinte.Brac>hCD4::GFP*; however, only the *mCherry::Cofilin* channel is shown for clarity. Images are pseudocolored with mpl-plasma LUT. The fluorescence intensity calibration scale bar is shown in panel A. Dented white line illustrates an example line used to quantify

normalized intensities across cell–cell contacts shown in panels D–F. Solid green circle encloses the region we defined as nuclear. The region between the 2 circles corresponds to a typical ROI used for quantification in panel G. Scale bar = 20 μm. Anterior to the left. (D–F) Quantification of normalized fluorescence intensity across notochord cells of *Ciiinte.Brac> mCherry::Cofilin; Ciiinte.Brac>hCD4::GFP*, double transgenic embryos. Black arrowheads in panel D mark cell–cell membrane contacts. The Pearson's correlation coefficients (r) are shown. The lower (LL) and upper (UL) 95% confidence intervals were as follows: negative control (LL: 0.9289, UL: 0.9638), Ano10$^{CRISPR}$ (LL: 0.03198, UL: 0.3602), and rescue embryos (LL: 0.7747, UL: 0.8803). We assayed 21≤n ≤32 animals and 105≤n≤161 cells per genotype. (G) Nuclear enrichment of Cofilin during intercalation, quantified as the fraction of *mCherry::Cofilin* over 3xnls::mNeonGreen in negative control, *Ano10$^{CRISPR}$* and rescue embryos expressing *Ciiinte.Brac> mCherry::Cofilin; Ciiinte.Brac>3xnls::mNeonGreen*. For statistical analysis of the data, we performed a Kruskal–Wallis test followed by a Dunn's multiple comparisons test. We assayed 30≤n animals and 302≤n cells per genotype (see also S62 Table). (H–J) Maximal projections of negative control, *Ano10$^{CRISPR}$* and rescue embryos at the onset of lumen elongation expressing *Ciiinte.Brac>mCherry::Cofilin; Ciiinte.Brac>3xnls::mNeonGreen*. The insets show median sections through the confocal stack. Yellow, green, and blue arrowheads mark the apical domain, lateral domains, and equatorial region of the basal cortex, respectively. Lateral views of the animals are shown, with anterior pointing to the left. Scale bars = 40 μm. (K) Mean normalized fluorescence intensity of *Brachyury>mCherry::Cofilin* along the equatorial region of the basal cortex of negative control, *Ano10$^{CRISPR}$* and rescue cells. Data presented as mean ± SEM. We assayed 21≤n≤25 animals and 108≤n≤128 cells per genotype. For the data presented in panel K, we used a mixed-effects model (REML) followed by Tukey's multiple comparisons test (see also S63 Table). (L) Quantification of sum intensity of *Ciiinte.Brac>mCherry::Cofilin* along the apical domain during the lumen expansion stage, in negative control, *Ano10$^{CRISPR}$* and rescue embryos. For statistical analysis of the data, we performed a Kruskal–Wallis test followed by a Dunn's multiple comparisons test. We assayed 44≤n animals and 220≤n cells per genotype (see also S64 Table). (M–O) Sum projections of *Ciiinte.Brac>mCherry::Cofilin* confocal stacks in negative control, *Ano10$^{CRISPR}$* and rescue genetic animals, at the onset of bidectional crawling. Insets correspond to median confocal slices of the regions enclosed in the white dashed boxes. Inset in panel N is rotated slightly relative to the maximal projection. Images are pseudocolored with mpl-plasma LUT. The fluorescence intensity calibration scale bar is shown in panel M. (P) Quantification of *mCherry::Cofilin* enrichment at the leading edge relative to entire cell across the 3 genotypes. For statistical analysis of the data, we performed a Kruskal–Wallis test followed by a Dunn's multiple comparisons test. We assayed 26≤n animals and 133≤n cells per genotype (see also S65 Table). (Q) Nuclear enrichment of *mCherry::Cofilin* in bidirectionally crawling notochord cells, quantified as the fraction of *mCherry::Cofilin* intensity divided by the intensity of a 3xnls::mNeonGreen nuclear marker. For statistical analysis of the data, we performed a Kruskal–Wallis test followed by a Dunn's multiple comparisons test. We assayed 27≤n animals and 276≤n cells per genotype (see also S66 Table). (R) Normalized intensity of *mCherry::Cofilin* along the basal domain of bidirectionally crawling notochord cells. Dashed red box marks the leading edge. Data presented as mean ± SEM. We assayed 21≤n≤24 animals and 87≤n≤99 cells per genotype. For the data presented in panel R, we used a mixed-effects model (REML) followed by Tukey's multiple comparisons test (see also S67 Table). The numerical data underlying this figure can be found in 10.5281/zenodo.12506448.

Finally, we employed electrophysiology to determine whether Ano10 possesses an ion channel functionality. We co-expressed Ano10-GFP and the purinergic receptor P2Y2R (which is coupled to Gq signaling and results in an increase in intracellular Ca$^{2+}$ upon stimulation by ATP) together in HEK293T cells and tested for current responses to ATP stimulation in a whole-cell configuration. Cells transfected with Ano10 exhibited a larger current than mock transfected, and the current–voltage relationship was nonlinear (Fig 11F and 11G and S68 Table). When we used the CAAX tag to tether Ano10 at the PM, we obtained larger currents at positive voltages (Fig 11F and 11G and S68 Table). Our experimental evidence argues in favor of Ano10 working as a channel rather than a phospholipid scramblase.

## Discussion

The present study has been designed to investigate the role of Ano10 in organ morphogenesis. Our study demonstrates that Ano10 coordinates notochord organ morphogenesis at the molecular, cellular, and supracellular levels. Specifically, our study reveals that ANO10 is required for both convergent extension and tubulogenesis steps during notochord formation (Fig 12A–12D). *Ano10$^{CRISPR}$* notochord cells exhibit defective convergent extension, which is manifested in defects in cell morphology, lower rates of successful intercalation events, irregular positioning of the cells along the A/P axis, and reduced A/P axis length (Fig 12A and 12B). To the best of our knowledge, ANO/TMEM16 proteins have not been previously implicated in the developmentally important process of cell intercalation [92]. Interestingly, members of the ANO/TMEM16 family are expressed in epithelial tissues that undergo cell intercalation during

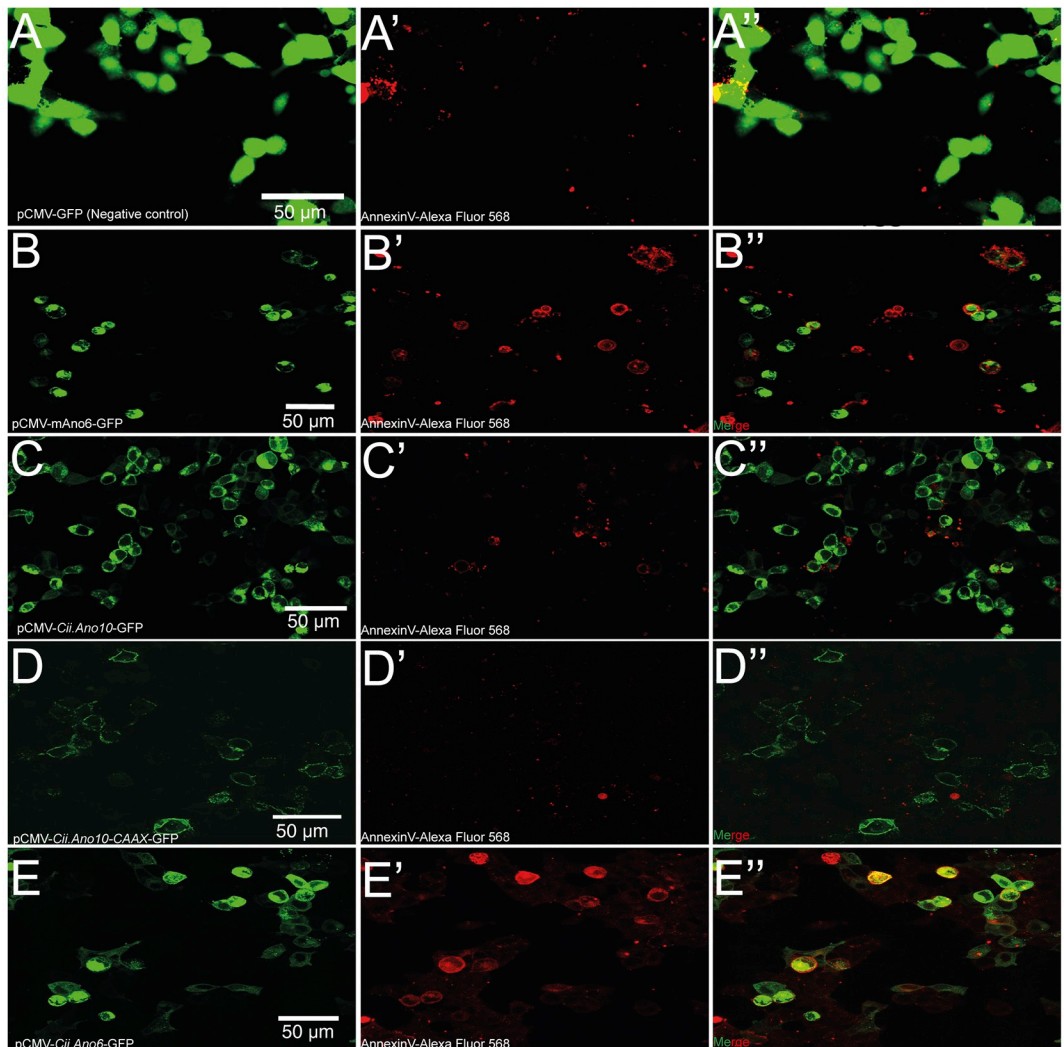

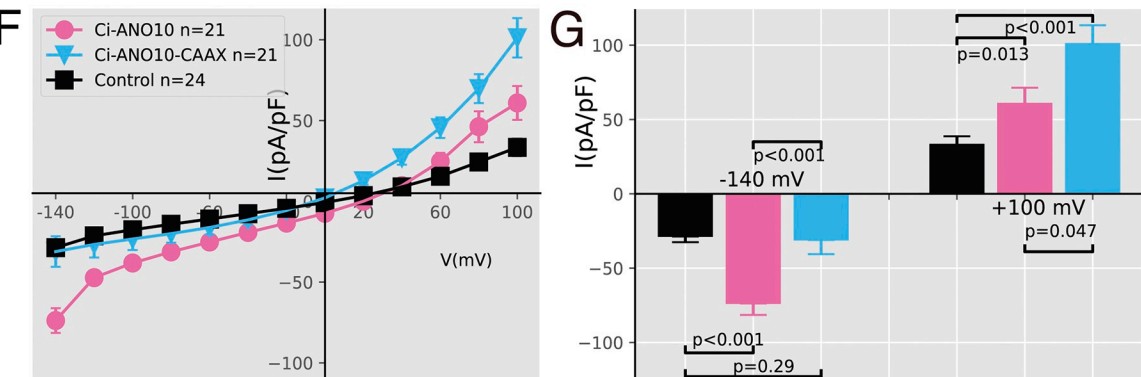

**Fig 11. Ano10 functions as a channel and likely not as a scramblase.** Representative confocal micrographs showing the results of the phospholipid scramblase assay in HEK293T cells. Overexpressing (A, A', A") pCMV-GFP (negative control), (B,B',B") pCMV-mAno6-GFP (positive control), (C,C',C") pCMV-Cii.Ano10-GFP; (D,D',D") pCMV-Cii.Ano10-CAAX-GFP, and (E,E',E") pCMV-Cii.Ano6-GFP. The left column shows GFP expression from the transfected constructs. The middle column shows Annexin V-Alexa Fluor 568 signal. The right column is the merge of the 2 channels. For quantification of the phospholipid scramblase assay result, please see

[Table 1](). (F) Stimulation with 100 μm ATP elicits currents in patch-clamp experiments for control (*n* = 10 cells), Ciinte.Ano10 (*n* = 13 cells), and Cii.Ano10-CAAX (*n* = 21 cells) expressed in HEK293T cells. Cells were held at 0 mV and pulsed to voltages from −140 mV to +100 mV (in steps of 20 mV). (G) Quantification of the current/voltage relationships between control, Cii.Ano10 and Cii.Ano10-CAAX expressing HEK293T cells. For statistical analysis, we used Mann–Whitney tests (see also S68 Table). The numerical data underlying this figure can be found in 10.5281/zenodo.12506448.

embryonic development [46,93]. For example, several murine Ano/Tmem16 gene family members including Ano10/Tmem16k are expressed in the neural tube [46]. This raises the possibility that some vertebrate Anoctamin proteins including ANO10/TMEM16K are involved in cell intercalation.

We additionally uncover a role for Ano10 in lumen expansion and lumen connection but not lumen initiation during tubulogenesis (Fig 12C and 12D). Our study adds Ano10 to the collection of membrane proteins responsible for tubulogenesis [25–28,94]. Previous work has shown that the anion transporter Slc26aα is specifically required for lumen expansion but not for lumen specification or lumen connection during *Ciona* notochord development [26]. Ano10 shows more pleiotropic phenotypes compared to Slc26aα, since Slc26aα is not involved in convergent extension and it is required for only one of the steps during lumen formation. However, neither of these transmembrane proteins seem to be involved in lumen initiation, thus the channels, transporters involved in this stage of lumen formation remain elusive. More generally, despite the fact that several Anoctamins have been shown to express in tubular structures such as kidney tubules, airways, intestine, pancreatic duct, and different types of glands [40–46,95] and even predicted via in silico simulations to contribute to pancreatic duct network development [96], evidence for the involvement of Anoctamins in biological tube development has been very limited, with the notable exception of 2 studies that implicated ANO1/TMEM16A in epithelial morphogenesis [97] and the development of the murine trachea [93]. Our work is complementary to these 2 studies since it implicates a new member of the ANO/TMEM16 protein family in tubulogenesis and provides a quantitative analysis of the role of Ano10 in regulating the cell behaviors and cytoskeletal organization of the notochord across different stages of development.

While it has been previously shown that notochord cells show sparse $Ca^{2+}$ activity during development [98] to date, it was unclear whether the second messenger $Ca^{2+}$ is required for convergent extension and/or tubulogenesis and to what extent the underlying cellular $Ca^{2+}$ machinery contributes to these processes. This study demonstrates that Ano10 works with

**Table 1. Scramblase activity of mAno6, *C. intestinalis* Ano10 and Ano6.**

| Condition | Total number of cells | AnnexinV (+) | Percentage (%) |
|---|---|---|---|
| Negative control (DMSO) | 1,049 | 16 | 1.53 |
| Negative control (Ionomycin) | 1,135 | 21 | 1.85 |
| mAno6 (DMSO) | 1,247 | 32 | 2.57 |
| mAno6 (Ionomycin) | 1,129 | 185 | 16.39 |
| Cii.Ano10 (DMSO) | 1,014 | 11 | 1.08 |
| Cii.Ano10 (Ionomycin) | 905 | 4 | 0.44 |
| Cii.Ano10-CAAX (DMSO) | 849 | 13 | 1.53 |
| Cii.Ano10-CAAX (Ionomycin) | 1,021 | 27 | 2.63 |
| Cii.Ano6 (DMSO) | 1,182 | 38 | 3.2 |
| Cii.Ano6 (Ionomycin) | 1,333 | 408 | 30.61 |

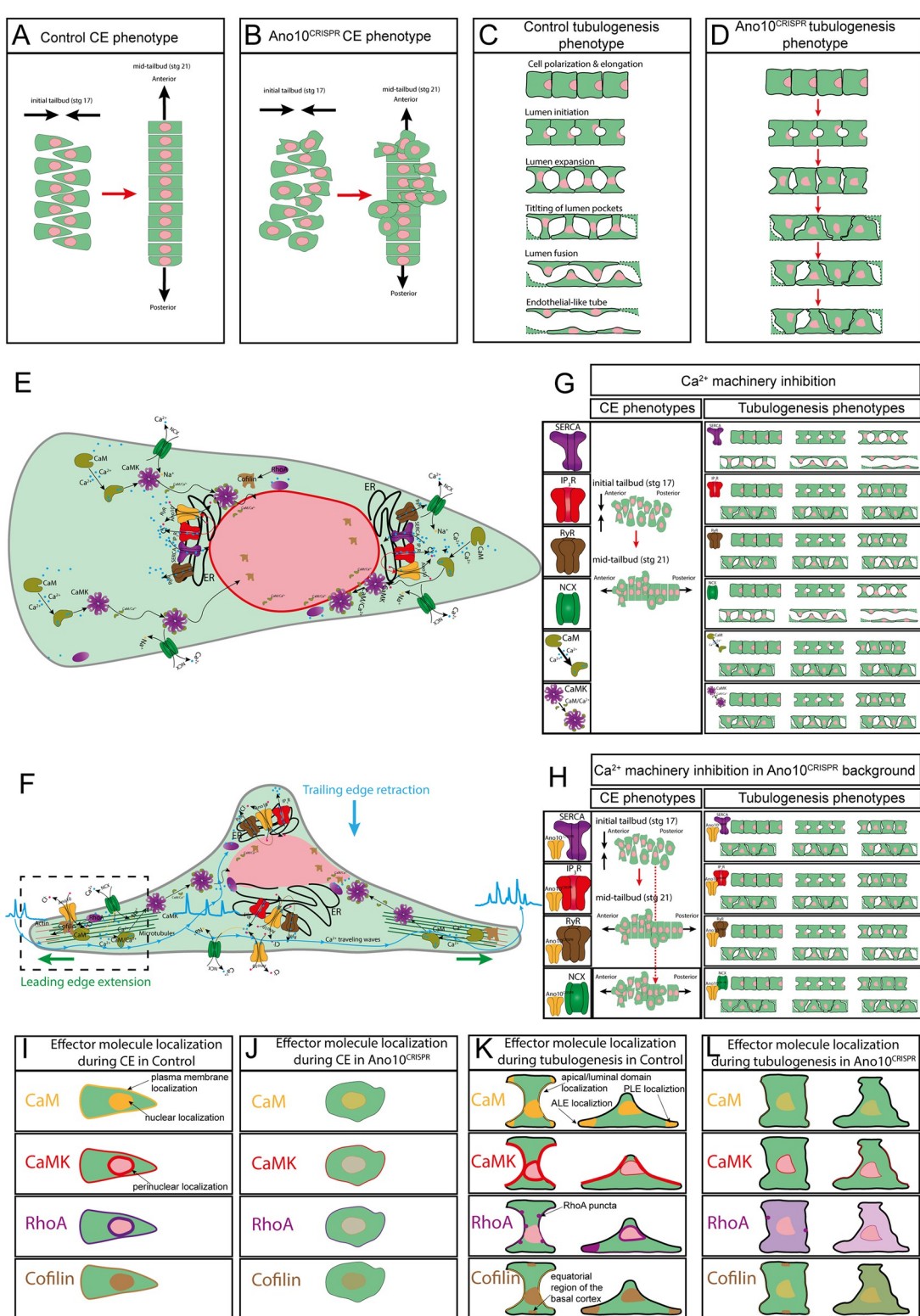

**Fig 12. Overview of the contribution of Ano10, cellular Ca²⁺ machinery and other effector molecules to notochord convergent extension and tubulogenesis.** (A, B) Ano10 coordinates notochord morphogenesis during convergent extension. Loss of Ano10 leads to erroneous cell intercalation, failing to form the regular stack of coins arrangement observed in controls. (C, D) Ano10 is dispensable for cell polarization, lumen initiation but it is required for lumen expansion, lumen tilting, and fusion. Ano10 mutants fail to form a notochord with the characteristic endothelial-like tube morphology. (E, F) Cartoons of

notochord cells during (E) CE and (F) lumen fusion (undergoing bidirectional crawling and trailing end retraction). The key components of the Ca$^{2+}$ signaling machinery and effector molecules are shown. Some of the proposed functional interactions between Ca$^{2+}$ signaling machinery and Ano10 are highlighted. Dashed square encloses the leading edge of the bidirectionally crawling notochord cell. The ALE and PLE are entry and exit sites of traveling Ca$^{2+}$ waves (blue lines with arrowheads and blue Ca$^{2+}$ traces). These Ca$^{2+}$ waves may coordinate cell movement between neighboring cells. (G) Summary of the convergent extension and tubulogenesis phenotypes observed following the pharmacological inhibition of key Ca$^{2+}$ signaling machinery (SERCA, IP3R, RyR, and NCX) and Ca$^{2+}$ sensors (CaM and CaMK). (H) Summary of the convergent extension and tubulogenesis phenotypes observed following the pharmacological inhibition of key Ca$^{2+}$ signaling machinery (SERCA, IP3R, RyR, and NCX) in an Ano10$^{CRISPR}$ background. Inhibition of the NCX transporter in Ano10 $^{CRISPR}$ background results in a much stronger CE phenotype. (I, J) During convergent extension, the Ca$^{2+}$ sensors calmodulin (CaM) and calmodulin kinase, as well as the cytoskeletal effectors RhoA (a small GTPase) and Cofilin show plasma membrane, nuclear, or perinuclear localization (I). These localization patterns are disrupted in Ano10 mutants (J). (K) Subcellular localization of CaM, CaMK, RhoA, and Cofilin molecules during lumen expansion and lumen fusion. They maintain the nuclear or perinuclear localization, but they also occupy new sites such as the equatorial region of the basal cortex, the apical/luminal domains, and the Anterior/Posterior leading edges (ALE/PLE). (L) In Ano10 mutants, the ALE/PLE and apical/luminal localization of these molecules is lost. The nuclear and perinuclear localization is also greatly diminished and a more diffuse (cytoplasmic) localization is often observed. NCX, Na+/Ca2+ exchanger.

endoplasmic reticulum and plasma membrane residing Ca$^{2+}$ machinery to modulate convergent extension and lumen expansion/connection, likely by controlling the generation and propagation of Ca$^{2+}$ signals that are required to orchestrate notochord morphogenesis. For most part Ano10 employs a common Ca$^{2+}$ signaling mechanism to regulate convergent extension and tubulogenesis (Fig 12E–12H). However, there are also certain differences that are worth highlighting.

First, during convergent extension Ano10 is acting downstream of the IP$_3$ Receptor (Fig 12E, red arrow). This is consistent with previous studies which have reported that Anoctamins are downstream effectors of the IP3R [64,99]. Conversely, during tubulogenesis it is RyR that is acting upstream of Ano10 (Fig 12F, brown arrow). This dichotomy may reflect a developmental stage-specific activation of Ano10 either by Ca$^{2+}$ mobilization from IP3-sensitive stores or by ryanodine-sensitive Ca$^{2+}$ stores.

Second, during convergent extension SERCA is part of the Ca$^{2+}$ machinery that mediates the Ano10-dependent intercalation defects (Fig 12E, 12G, and 12H). Inversely, during tubulogenesis SERCA is dispensable for the Ano10-dependent signaling mechanism (Fig 12F–12H).

The final difference we would like to highlight relates to the Na$^+$/Ca$^{2+}$ exchanger (NCX) which we have identified as a novel player in notochord morphogenesis. We have shown that NCX is essential for convergent extension, likely acting through an Ano10-independent pathway (Fig 12E, 12G, and 12H). However, during tubulogenesis we have found that NCX activity is subject to regulation by an Ano10-dependent inhibitory mechanism (Fig 12F–12H). Loss of Ano10 may lead to an unregulated activity of NCX, which results in Ca$^{2+}$ signaling and tubulogenesis defects. To the best of our knowledge, such an interaction has not been previously described for Ano10. However, it has been shown that in osteoblasts, ANO6 is required for NCX to fully operate in the context of bone mineralization [100]. We note that during tubulogenesis, *Ciona* Ano10 is strongly recruited to the plasma membrane in addition to the ER which is the main localization site during convergent extension (Fig 12E and 12F). To what extent this change in Ano10 localization contributes to its ability to interact with plasma membrane Ca$^{2+}$ machinery and regulate NCX remains unclear. While the mechanism is not fully understood, our findings provide an additional example of how Ano10 and more generally Anoctamins/TMEM16 proteins may control intracellular Ca$^{2+}$ signaling during development.

A surprising finding of our study is that 2 highly conserved Ca$^{2+}$ sensors, CaM and CaMK, [84,101] are required throughout notochord morphogenesis and that their subcellular localization in notochord cells is Ano10 dependent (Fig 12E–12G and 12I–12L). We note that a previous study in *Ciona* had reported the presence of a calmodulin-like gene in the notochord (BCamL) but its role in notochord morphogenesis was not explored [102]. We believe that the

enrichment of CaM and CaMK at important sites for notochord cell gene regulation and cell behavior (i.e., the nucleus, cell–cell contacts, the apical domains, and leading edges) suggest CaM and CamK act as "encoders and decoders" of compartmentalized $Ca^{2+}$ signals establishing a feedback loop between morphogenetic actions (e.g., cell shape changes), $Ca^{2+}$ dynamics and transcriptional activity. This hypothesis is supported by our $Ca^{2+}$ imaging, which revealed that inhibition of CaM and CaMK leads to deranged $Ca^2$ dynamics in the notochord.

While it has been previously shown that ANO1/TMEM16A activity can be regulated by CaM/CaMK-dependent mechanisms [103–105], to the best of our knowledge the inverse has not been shown. How exactly Ano10 interacts with CaM and CaMK to modulate their localization remains to be elucidated in future studies.

In *Ciona robusta* (formerly known as *C. intestinalis A)*, $Ca^{2+}$ activity has been reported in the presumptive notochord area during cell intercalation [98]. Here, we performed volumetric $Ca^{2+}$ imaging of the notochord during the bidirectional crawling phase. We recorded a surprising amount of robust bioelectrically activity in the notochord cells during this phase of tubulogenesis. $Ca^{2+}$ as a second messenger can mediate rapid, compartmentalized, and information-rich signaling, thus it is highly suited to orchestrate the highly complex cell behaviors carried out by bidirectionally crawling cells.

A novel observation we made is that $Ca^{2+}$ signals can be transmitted across neighboring notochord cells via the leading and trailing edges, which can act as the site of origin for $Ca^{2+}$ waves within a cell (Fig 12F). This is a novel function that we can attribute to the leading and trailing edges of notochord cells. Loss of An10 function strongly reduces the ability of leading edges to effectively initiate $Ca^{2+}$ waves within a cell and transmit $Ca^{2+}$ signals across notochord cells. The purpose of this bioelectrical activity warrants further investigation, but it is very likely involved in modulating actin and microtubule dynamics which are disrupted in Ano10-CRISPR notochord cells. Indeed, cytoskeletal reorganization is one of the proposed functions of $Ca^{2+}$ signaling at leading edges of migrating cell types both in physiological and pathological contexts [106].

Our study points to a potential link between Ano10 loss of function and deranged cytoskeletal organization during notochord development. How might this come about? We explored the localization of the small GTPase RhoA, a regulator of actin-based cytoskeletal dynamics and the essential actin regulatory protein Cofilin. Under control conditions RhoA and Cofilin localize in/around the nucleus throughout notochord development, as well as the cell–cell contacts in intercalating cells (Fig 12I–12L).

During tubulogenesis they localize at the apical domains and/or leading edges of notochord cells (Fig 12K). The localization of RhoA and cofilin is consistent with a role in regulating the actin filaments (and by association the microtubule bundles) that have been previously documented to be enriched in the apical domains and the leading edges during lumen formation and bidirectionally crawling, respectively [17] (Fig 12F). However, in Ano10-CRISPR notochord cells RhoA and Cofilin are diffusely localized (Fig 12L). This mislocalization may influence cytoskeletal organization in the notochord of Ano10-CRISPR embryos. Alternatively, other cellular processes may be affected by deranged RhoA localization such as vesicular trafficking and membrane dynamics [107–110]. It has been shown that vesicular trafficking is a mechanism which contributes to notochord lumen formation [11,19,111,112]. Thus, it is conceivable that Ano10 impacts vesicular trafficking and membrane dynamics during lumen formation via a RhoA-dependent mechanism. Future studies are required to ascertain the mechanism by which Ano10 impacts the RhoA and Cofilin signaling pathway.

Our work raises the possibility that robust $Ca^{2+}$ signaling which is at least in part mediated by Ano10 may act as a pleotropic signal integrator [113] during notochord development. More generally, our finding that *C. intestinalis* Ano10 contributes to $Ca^{2+}$ homeostasis in the

notochord is in agreement with previous studies in the field which have shown that mutations in Anoctamins, including ANO10 lead to deranged $Ca^{2+}$ activity, which in turn results in pathological conditions [62,63,80,114]. For example, the defects in ion transport, cell volume regulation and cell migration exhibited by Ano10 knockout mice have been attributed to deranged local $Ca^{2+}$ signaling [80]. Given the importance of $Ca^{2+}$ signaling during embryonic development in vertebrates [115], our findings raise the exciting possibility that at least some of the vertebrate ANO10/TMEM16K proteins may play a critical role in regulating $Ca^{2+}$ dynamics during development in a broad spectrum of biological tissues.

Different studies have indicated that vertebrate ANO10 can function either as a scramblase or an ion channel [56,59]. An in vitro study has suggested that ANO10 may act as a CaCC [59]; however, other studies have reported that ANO10 is an ER-residing phospholipid scramblase [55,56,116]. Our in vivo and in vitro work suggests that *C. intestinalis* Ano10 functions as a channel rather than as a phospholipid scramblase. Our finding is supported by a recent study which reported using a Naïve Bayes classifier that *C. intestinalis* Ano10 is indeed an ion channel [117].

In conclusion, cells must integrate a diversity of local signals to coordinate tissue and organ-level processes. Our study suggests that Ano10 is part of the machinery that integrates these signals and coordinates the generation of appropriate cellular behavioral output to build one of the defining characteristics of all chordates, the notochord. We hope that our study will prompt new studies on the role of TMEM16K/ANO10 and other Anoctamins in the development of tubular structures.

## Methods

### Animal collection

Adult *C. intestinalis* (formerly Type B) were collected from: Døsjevika, Bildøy Marina AS, postcode 5353, Bergen, Norway. The GPS coordinates of the site are the following: 60.344330, 5.110812.

### Rearing conditions for adult *C. intestinalis*

Adult *C. intestinalis* were housed in a purpose-made facility at the Sars Centre. Approximately, 50 to 100 adults were housed in large 50 L tanks with running sea water. The temperature of the sea water was maintained at 10˚C with constant illumination and food supply (various species of diatoms and brown algae) to enhance egg production and prevent spawning [118].

### Electroporations of zygotes and staging of embryos

Egg collection, fertilization as well as rearing were done following standard protocols [119]. Gravid adult *C. intestinalis* were dissected to obtain mature eggs and sperm to perform fertilization in vitro. Eggs were dechorionated using chemical dechorionation in a 1% sodium thioglycolate (Sigma Aldrich) and 0.05% pronase (from Streptomyces griseus, Sigma Aldrich) mix dissolved in filtered seawater. Eggs in the dechorionation solution were placed on a rocker for approximately 6 min until zygotes were completely dechorionated. Dechorionated eggs were washed several times using artificial seawater (ASW) and then fertilized with sperm for approximately 10 min. Electroporation was performed according to published protocols with minor modifications [120]. After thoroughly washing, the fertilized eggs were suspended in a solution containing 300 µl of ASW, 400 µl of D-mannitol solution, and a DNA solution which contain a mix of plasmids to be electroporated. Each plasmid was prepared as a midi-prep at a typical concentration range of 3 to 5 µg/µl. For each plasmid, we added typically a total of 40 to

50 μg. Importantly, the final volume of the DNA solution could not exceed 100 μl. We electroporated fertilized eggs in electroporation cuvettes with a 4 mm gap (MBP Catalog #5540) using a BIORAD GenePulserXcell equipped with a CE-module. The settings we used were as follows: Exponential Protocol: 50 V, Capacitance was set to: 800–13,00 μF, Resistance was set to ∞ and we aimed to achieve an electroporation time constant between 15 to 30 milliseconds. Embryos were cultured in ASW (Red Sea Salt) at 14˚C until they were used for experiments.

The pH of the ASW was set to 8.4 at 14˚C. The salinity of the ASW we prepared was set to 3.3% to 3.4%. Embryos were cultured during early development (zygote to stage 13 [77]) at 14˚C. For most live-imaging experiments, we tried to ensure that the temperature would stay stable around 18˚C using room and/or microscope cooling.

## Molecular biology procedures to obtain expression constructs for cell culture and *C. intestinalis* transient transgenesis

The primers for cloning *C. intestinalis Ano10* cDNA were designed according to the *C. intestinalis* genome databases (Aniseed Database gene model KH2012:KH.C3.109 and Ghost Database KY21.Chr3.1036 [121,122]). A FLAG-tag sequence as DYKDDDDK was directly added to the primers for *Ano10* cDNA amplification. Around 2 kb Ano10 cDNA was amplified by PCR using Norwegian *C. intestinalis* cDNA as the template. The PCR-amplified cDNA fragments were cloned into PEGF-N1 vector (Clontech, 6085–1) with XhoI/BamhI enzyme sites. To generate the CAAX tagged *C. intestinalis Ano10*, CAAX tag sequence KKKKSKTKCVIM was directly added to the primers for *Ano10* cDNA amplification, then followed with the same PCR and ligation process as mentioned above.

For experimental use in *C. intestinalis* transgenic embryos, the *Ano10* cDNA was amplified using cDNA at a concentration of 100 ng/μl, a dNTPs mix (Thermo Fisher, R0182) and Q5 High-Fidelity DNA Polymerase (M0491L, NEB) to perform the PCR reaction. Subsequently, we gel purified the PCR products using Zymogclean Gel DNA Recovery Kit (Zymo Research, D4002). The purified product was inserted into the pDONR221 vector using BP Clonase II (Invitrogen, P/N56480). We identified positive clones using restriction digest and we sequenced multiple clones with Sanger sequencing. Both the cDNA and the promoter sequence of *Ano10* were cloned into the Gateway system for experimental use in *C. intestinalis*. We inserted the PCR products of *Ano10* cDNA into PDONR221 vector and the promoter into pDONR P4-P1R vector with BP Clonase II (Invitrogen), respectively. Expression vectors were recombined from LR reaction with pDESTII vector and LR Clonase II (Invitrogen).

To amplify promoters for the following genes: *Brachyury*, Ano10, Carbonic anhydrase 2 (Aniseed Database gene model KH.C1.423; Ghost Database KY21.Chr1.1715), we extracted genomic DNA from local Norwegian animals using the Wizard Genomic DNA Purification Kit (A1120, Promega). We then used the purified gDNA at a concentration of 100 to 150 ng/μl, the primers shown in S69 Table, a dNTPs mix (Thermo Fisher, R0182), and Q5 High-Fidelity DNA Polymerase (M0491L, NEB) to perform PCR reactions. Subsequently, we gel purified the PCR products using Zymogclean Gel DNA Recovery Kit (Zymo Research, D4002). These were inserted into the P4-P1R vector using BP Clonase II (Invitrogen, P/N56480). We identified positive clones using restriction digest and we sequenced multiple clones with Sanger sequencing.

For the middle position, we used hCD4GFP, GCaMP6s, nls::Cas9::nls (subcloned from Eef1a-1955/-1>nls::Cas9::nls a gift from Lionel Christiaen, Addgene plasmid # 59987; http://n2t.net/addgene:59987; RRID:Addgene_59987) [61], EB3-mNeonGreen [123] a gift from Dorus Gadella (Addgene plasmid # 98881; http://n2t.net/addgene:98881; RRID:

Addgene_98881), tdTomato-Lifeact-7 was a gift from Michael Davidson (Addgene plasmid # 54528; http://n2t.net/addgene:54528; RRID:Addgene_54528). Primers for all subcloned constructs are shown in S69 Table.

We performed a four-way Gateway choosing one of the promoters at a time in the first position, with an appropriate middle position entry and unc-54 3′ UTR in the third position. These were recombined into a pDEST II backbone using LR Clonase II (Invitrogen, P/N56485). All expression constructs generated in this work were then midi-prepped using NucleoBond Xtra Midi kit.

The following plasmids were a gift from Dr. Di Jiang: (1) *Ciiinte.Brac>GFP::RhoA*; (2) *Ciiinte.Brac> mCherry::Cofilin*; (3) *Ciiinte.Brac>mCherry::CaMK*; (4) *Ciiinte.Brac>mCherry:: CaM*. They were sequenced using whole plasmid sequencing (Plasmid-EZ, Genewiz).

## Whole mount in situ hybridization

For in situ hybridization targeting *Ano10* mRNA, 700 bp *Ano10* cDNA was amplified and inserted into pCRII-TOPO vector (TOPO Cloning Kit, Thermo Fisher). Primers are shown in S8 Table. The antisense DNP-UTP (Roche) labeled probe for *Ano10* was generated by in vitro transcription with MEGAscript SP6 Transcription kit (Invitrogen). Whole-mount in situ hybridization was performed according to the procedure described by Christiaen and colleagues [120]. Embryos and probes were hybridized at 55˚C for 16 to 48 h with 200 ng/ml of probe. For imaging, embryos were suspended in 2% DABCO/50% glycerol in PBS and mounted on slides with zero thickness coverslips. We imaged our samples on a Nikon Eclipse E800 upright compound microscope. We performed 3 independent experiments, and we included negative controls (hybridization with a sense Ano10 probe).

## Phylogenetic analysis

The 91 orthologous Anoctamin protein sequences from 22 species (*Homo sapiens*, *Mus musculus*, *Xenopus tropicalis*, *Danio rerio*, *Petromyzon marinusurgerretus burgeri*, *Ciona intestinalis*, *Saccoglossus kowalevskii*, *Asterias rubens*, *Capitella teleta*, *Drosophila melanogaster*, *Oikopleura dioica*, *Branchiostoma floridae*, *Acyrthosiphon pisum*, *Strongylocentrotus purpuratus*, *Nematostella vectensis*, *Lottia gigantea*, *Caenorhabditis elegans*, *Aspergillus fumigatus*, *Nectria haematococca*, *Saccharomyces cerevisiae*, *Dictyostelium discoideum*), which were longer than 300 amino acids, were downloaded from Uniprot [124]. Four *C. intestinalis* Anoctamins were searched and downloaded from Ciona genome database (Ghost [122] and Aniseed [121,125]). Amino acid alignments were carried out by using the online version of MAFFT (https://www.ebi.ac.uk/Tools/msa/mafft/). Poorly aligned regions of the multiple protein alignment of Anoctamins were removed with Gblocks (http://phylogeny.lirmm.fr/phylo_cgi/one_task.cgi?task_type=gblocks) with the least stringent parameters. Maximum likelihood (ML) phylogenetic analyses were conducted with RAxML v8 (https://raxml-ng.vital-it.ch/#/) with the autoMRE option to calculate the bootstrap support values. The tree was manipulated and annotated by iTOL (https://itol.embl.de/). Through phylogenetic analysis, we named 4 Ciona Anoctamins respectively by KH.C3.109/KY21.Chr3.1036 as *C. intestinalis* Ano10, KH.C8.274/KY21.Chr8.1253 as *C. intestinalis* Ano5, KH.C10.524/KY21.Chr10.513 as *C. intestinalis* Ano6, and KH.C8.585/KY21.Chr8.1090as *C. intestinalis* Ano7.

## RNA isolation and cDNA preparation

Total embryos' RNA was extracted at different developmental stages using TRIZOL (Thermo Fisher) according to the manufacturer's instructions. Contaminating DNA was degraded by treating each sample with Dnase (Roche) then heating up to 95˚C to eliminate the enzyme

activity. The amount of total RNA was measured by Nanodrop (ND-1000 UV–Vis spectropho-tometer; NanoDrop Technologies) according to the absorbance at 260 nm and the purity by 260/280 and 260/230 nm ratios, and 500 ng of total RNA was retrotranscribed with the kit (SuperScript IV First-Strand Synthesis System, Invitrogen) for each sample according to the manufacturer's instructions.

## Generation of negative control and Ano10^CRISPR mutant embryos by CRISPR/Cas9

The guide RNA(gRNA) sequence targeting *C. intestinalis Ano10* was designed through online tools E-CRISP (http://www.e-crisp.org/E-CRISP/) [126] with relaxed selection and CRISPOR (http://crispor.tefor.net/) [127]. Six pairs of gRNAs were designed, and only 1 pair which target GTTTAAATGAGGTAACGCAG sequence on Exon 6 finally was used in subsequent experiments. The negative control gRNA was designed to not target any sequence in the *C. intestinalis* genome. The gRNA oligonucleotides were inserted into the U6>sgRNA(F+E) vector which was a gift from Lionel Christiaen (Addgene plasmid # 59986; http://n2t.net/addgene:59986; RRID:Addgene_59986) [61]. Primers used to build reagents in these experiments are shown in S8 Table.

The 30 μg *Cii.Brachyury>nls::Cas9::nls* or Carbonic anhydrase 2>nls::Cas9::nls and 70 μg *U6>Ano10gRNA* or *U6>ControlgRNA* plasmids were electroporated together into fertilized, dechorionated eggs. Embryos were fixed for genomic DNA extraction (JetFlex Genomic DNA Purification Kit, Thermo Fisher) according to the manufacturer's instructions. To detect the mutation, we used a melting curve assay, using single embryo genomic DNA as our template (CFX Connect Real-Time PCR Detection System of Bio-Rad). Positive samples from the melting curve assay were cloned into pCRII-TOPO vector and sent for Sanger sequencing. The cleavage detection assay was done as described by Stolfi and colleagues [61] (GeneArt Genomic Cleavage Detection Kit, Thermo Fisher). The efficiency of CRISPR/Cas9 editing was ≥39.0% across multiple independent experiments. For rescue experiments, we synthesized (GeneCust, France) an Ano10cDNA where the gRNA target site was replaced with the sequence GTTTAAACGAAGTAACACAA. We used synonymous codons thus we did not change the amino acid sequence encoded by the cDNA. Rescue constructs were electroporate in the same mix as *Cii.Brachyury>nls::Cas9::nls* or Carbonic anhydrase 2>nls::Cas9::nls and *U6>Ano10gRNA* at 60 to 80 μg.

## Embryo dissociation and MACS sorting

Transgenic embryos harboring relevant transgenes for negative control or Ano10^CRISPR conditions as well *Brac>CD4::GFP* were collected at Hotta stages 24~25 [77]. The embryos were incubated in ASW with 0.2% trypsin (Thermo Fisher) and cell dissociation was facilitated by pipetting up and down the mix. After approximately 4 to 6 min, we stopped the trypsin reaction by adding 0.05% bovine serum albumin (BSA) (Thermo Fisher). The solution was filtered with 35 μm cell strainer cap in 50-ml falcon tubes to remove non-dissociated cells. The supernatant was removed after centrifugation at 4°C with a speed of 3,000 RCF. Then, the remaining sample was washed twice with ASW containing 0.05% BSA. The cells were resuspended in 180 μl ASW containing 0.05% BSA and we added 20 μl of CD4 antibody (Miltenyi Biotec) conjugated with magnetic beads. The sample tubes were kept for 1 h at 4°C and then centrifuged at 1,000 RCF at 4°C for 3 min. After washing 3 times, the cells were resuspended with ASW containing 0.05% BSA. We performed the CD4 positive cell sorting using the OctoMACS starting kit (MIltenyi Biotec) following the manufacturer's protocol.

## Immunostainings

*Ciona* transgenic embryos used for Ano10 protein subcellular localization experiments were fixed in 4% paraformaldehyde (PFA) at room temperature for 15 min. We then blocked with 10% heat-inactivated goat serum in PBS for 2 h at room temperature. Commercial anti-FLAG antibody (Sigma-Aldrich) was used at 1:50 dilution and incubated for 2 h at room temperature. Secondary goat anti-mouse antibodies conjugated with Alexa Fluor 488 (Invitrogen) was used at 1:200 dilution. Cell nuclei were stained with DAPI (Invitrogen) at 1:1,000 dilution in PBT solution.

For determining the integrity of the notochord border, we performed immunostaining with an antibody against laminin (Merck L9393-100UL). Embryos were fixed in 4% PFA at room temperature for 15 to 20 min. We then blocked with 10% heat-inactivated goat serum in PBS for 2 h at room temperature. We then transferred the embryos to a PCR tube (with a max capacity of 250 μl) applied the anti-laminin antibody at 1:50 dilution and incubated overnight at room temperature with gentle agitation. A secondary antibody mouse anti-rabbit was applied. This was conjugated with Alexa Fluor 568 (Invitrogen) and was used at 1:200 dilution.

## Tissue culture and cell transfection

HEK293T cells and HeLa cells were cultured in DMEM and DMEM/F12 (Gibco), respectively, which were both supplemented with 10% fetal bovine serum, 2 mM L-glutamine, 100 U/ml$^{-1}$ penicillin, and 100 μg/ml$^{-1}$ streptomycin. All transfections were carried out by Lipofectamine Transfection Reagent (Invitrogen) following the manufacturer's protocol. All experiments were performed between 36 and 48 h after the transfection.

## Electrophysiology

Electrophysiological studies were carried out in HEK293T cells. The cells were seeded on the round coverslip coated by poly l-lysine. The fluorescence positive cells were selected under microscopy (Zeiss). Recordings were performed using the whole-cell patch-clamp configuration. Patch pipettes had resistances ranging from 3 to 6 MΩ when filled with the pipette solution. Data were acquired by a HEKA EPC10 amplifier and Patchmaster software (HEKA). Current measurements were performed at room temperature. Capacitance and access resistance were monitored continuously. Currents were filtered at 2.9 kHz with a low-pass Bessel filter. The Pipette (intracellular) solution contained (mmol/L): 146 CsCl, 5 EGTA, 2 MgCl$_2$, 10 sucrose, 10 HEPES, pH was adjusted to 7.3 with NMDG. The extracellular solution contains (mmol/L): 140 NaCl, 5 KCl, 2 CaCl$_2$, 1 MgCl2, 10 mM HEPES, 15 D-glucose, pH was adjusted to 7.4 with NaOH. Data were analyzed by python 3.8.

## Drugs used for targeting ER and PM Ca$^{2+}$ machinery

Control: Dimethyl sulfoxide (D8418, Sigma), SERCA: Thapsigargin (T9033, Sigma), RyR: JTV-519 (SML0549, Sigma), IP$_3$R: 2-Aminoethyl diphenylborinate (D9754, Aldrich), NCX: Benzamil hydrochloride (B2417,Sigma), CaMK: Autocamtide-2 (189485, EMD Millipore), CaM: Calmidazolium (C3930, Sigma).

## Calcium imaging using GCaMP8f

An FV3000 inverted confocal microscope with 30 or 40× silicon immersion objectives was used. Illumination was provided using a 488-nm laser. Embryos with similar expression levels were selected under a dissecting fluorescent microscope. They were transferred to a 6-cm cell culture petri dish and embedded in 0.5% low melting point agarose (Fisher BioReagents,

BP1360-100). Imaging was done at 18˚C. We collected 8 to 9 confocal slices per notochord volume. The number of stacks was adjusted to achieve a total recording time of 7.5 min. Data was collected using a HSD detector. Analysis was performed using the $Ca^{2+}$ imaging analysis platform Mesmerize [128]. On average 10 cells per animal were analyzed.

## Phospholipid scramblase assay

For the cell culture phospholipid scramblase assay, we generated and transfected into HEK293T the following constructs pCMV-GFP (negative control), pCMV-mAno6-GFP (positive control), pCMV-*Cii.Ano10*-GFP; pCMV-*Cii.Ano10*-CAAX-GFP and pCMV-*Cii.Ano6*-GFP. Primers for reagents used in these experiments are shown in S69 Table. Following transfection, we waited for 24 h and we then seed the transfected HEK293T cells in a confocal imaging chamber (0 thickness coversleep, 22 mm diameter). The cells were washed twice with PBS. Then, they were incubated with 1:100 dilution of AnnexinV-Alexa Fluor 568 antibody in binding buffer (Thermo Fisher) for 10 min, which also contained 1× RedDot1 Far-Red Nuclear Stain for dead cell detection and with 10 µm ionomycin or DMSO. We then fixed the cells for 15 min with 4% PFA. The cells were washed 3 times with PBS and imaged using a Leica TSP5 confocal. We used machine vision to segment and extract the contours in the green (transfected construct tagged with GFP), red (Annexin v-AlexaFluor568), and magenta (RedDot1 Far-Red) channels. We curated manually the in ImageJ. Cell numbers of Green and/or Red positive cells were counted using Scikit-image in python 3.8. RedDot1 positive cells were excluded from the analysis.

## Quantification of notochord cell intercalation fraction, notochord length analysis, and cell shape parameters

We electroporated the reporter plasmids *Ciinte.Brac>hCD4::GFP* and *Ciinte.Brac>NUP50::RFP* in combination with our negative control, Ano10[CRISPR] or rescue electroporation mixes. If the animals were to be used in combination with drugs only at Hotta stage 14 [77] animals were were transferred to Mattek chambers and imaged using an OLYMPUS spinning disk confocal using a 30× silicon oil objective, illuminated with 488 nm laser, with an exposure time of 400 ms. Stacks approximately 80 µm wide with a z-step size of ca 1.5 µm were collected. Animals were imaged every 15 min for a maximum of 300 min. To score the fraction of notochord cells intercalated, we followed the approach of Veeman and Smith [9]. In brief, we scored each cell manually as being intercalated if the contacts it made with non-notochord cells formed a closed ring in contrast to non-intercalated if these contacts formed only a segment of the ring. For notochord length quantification over time, we fixed animals grown at 18˚C every 30 min (stage 14 until stage 25 according to Hotta [77] and imaged those using an OLYMPUS FV3000 confocal using a 30× objective. For the quantification of cell shape parameters, we analyzed embryos exhibiting mediolateral elongation and alignment. In Fiji, we selected a medial slice from each stack and measured across 4 to 5 randomly selected per animal the angle enclosed by the notochord cell protrusion and the aspect ratio of the cells. Cell volumes were quantified in IMARIS. The person performing the quantifications was blind to the genotype/ pharmacological treatment.

## Quantification of notochord cell center to skeleton midline distance metric

*Brac>hCD4::GFP* expressing embryos were allowed to develop until the end of stage 24 and then we fixed them. We labeled cellular F-actin with a conjugated Alexa Fluor 647 phalloidin antibody at a dilution of 1:250 and imaged under a confocal microscope. Using custom made software, we labeled the notochord cell centers from position 0–39 (or 1–40) in increasing

order along the Anterior-Posterior axis using the hCD4::GFP signal to identify individual cells and segmented the embryo outline using the phalloidin signal. For all labeled cells, we calculated the distances to the central skeleton line. Comparisons of the distance to the central skeleton line for each cell number are an indication of intercalation quality.

### Notochord lumen curvature and area measurements

*Brac>hCD4::GFP* and *Brac>Cav1::mCherry* plasmids were co-electroporated at 40 μg each with or without the CRISPR/Cas9 constructs to generate transgenic embryos. When the embryos reached notochord development stage IV, we transferred them to a Mattek chamber and they were embedded in low melting point agarose (0.5% in ASW). For the pharmacological inhibition experiments, we incubated control animals in DMSO or anoctamin/ER/PM $Ca^{2+}$ machinery inhibitors diluted to a final concentration of 100 μm. We imaged tubulogenesis using an OLYMPUS spinning disk confocal. Each animal was revisited every 20 to 30 min. To measure average lumen curvature at the end of our recordings, we opened the acquired stacks in ImageJ and generated orthogonal views of the last acquired time point. We then used the Fiji plugin Kappa (https://github.com/fiji/Kappa) to obtain the average curvature values. We traced the shape of the lumen with a B-spline curve (type: closed, stroke thickness 1) using on average 10 points and then used the data fitting algorithm: Point Distance Minimization.

### Measurement of morphology features in bidirectionally crawling notochord cells

*Brac>hCD4GFP* expressing embryos were allowed to reach bidirectional crawling stage, at which point they were fixed and imaged using a FV3000 point scanning confocal. We quantified the leading-edge length by loading the confocal stacks in Fiji and measured in Fiji with the segmented line tool the sum of the distance corresponding to the Anterior LA plus NE length. We assayed 4 to 5 cells per condition. Scoring was done blind with respect to the treatment/genotype of the animals.

For live tracking and speed imaging of the notochord cells, we electroporated *Brac>hCD4GFP* and *Brac>NUP50-mCherry* in combination with the relevant CRISPR/Cas9 genome editing plasmids. Late stage 23 embryos were mounted in low melting point agarose (0.5% in ASW) and imaged every 5 min for up to 7 h (stage 24 till stage 27) under the spinning disk confocal. Notochord cell movement was tracked using the Imaris Software Spots module.

### Measurement of radial/non-radial organization of microtubules

We electroporated 60 μg of *Brac>Ensconsin::GFP* [17] (a kind gift from Dr. Di Jiang) in combination with the respective negative control, *Ano10^CRISPR* and rescue plasmids. We fixed animals at the onset of stage V [19] at the onset of lumen initiation. We acquired confocal stacks which were used to generate maximal projections. To separate the images into radial and non-radial components, we used a customized ImageJ macro (https://github.com/ekatrukha/radialitymap). We selected the "Cubic Spline Gradient" method and Tensor Sigma parameter of 8 pixels to separate the radial and non-radial image that correspond to the separated radial and non-radial components of the original projection. Subsequently, we used both the original image and the 2 images showing the separated radial/non-radial components to generate radial intensity profiles. These were generated using the Radial Profile Angle plugin with the center located at the middle of the notochord cell and the edge located at the most distant portion of the cell periphery. A 180° integration angle was used. To quantify the non-radiality of the microtubule organization for each cell, the areas under the curve (AUC) of the radial intensity profile of the original image (total intensity) and the AUC of the non-radial map image were calculated using GraphPad Prism. Using these values, we calculated the non-radial intensity as a percentage of the total intensity: $(AUC_{non-radial}/AUC_{Total\ intensity}) *100$.

### Live imaging of microtubules using EB3-mNeonGreen

To image microtubule dynamics, we used as a marker the *Cii.Brachyury>EB-3-mNeonGreen* reporter, which was electroporated at 40 µg together with CRISPR/Cas9 plasmids. Electroporated embryos were allowed to grow to the end of Stage IV and imaged using the spinning disk confocal every 5 min. Comet count analysis was performed using the ComDet plugin (https://github.com/ekatrukha/ComDet). We measured the number of comets across the entire cell and the number of comets at the leading edges. We then calculated the ratio of leading-edge comet intensity/overall comet intensity across the entire cell.

### Actin distribution quantifications

To image actin distribution in the notochord, we electroporated *Brac>hCD4GFP* (40 µg) and *Brac>TdTomato-Lifeact* (40 µg). Following the completion of convergent extension embryos were transferred them to a Mattek chamber and imaged every 5 min on a spinning disk confocal. First, we analyzed actin distribution during lumen initiation by measuring actin intensity in the median confocal sections by drawing a straight line across the Anterior-Posterior axis of each cell, crossing the apical domains. To quantify the fraction of actin intensity at leading edge in bidirectionally crawling cells, we traced the cell outlines using a freehand selection line. We measured the raw integrated density of the entire cell. Then, we drew outlines around the leading edges and measured the raw integrated density. We then calculated the ratio of: raw integrated density $_{\text{(leading edges)}}$/raw integrated density $_{\text{(entire cell)}}$.

### Analysis of CaMK, CaM, RhoA, and Cofillin enrichment at different subcellular location

All analysis was performed in FIJI [129] using median slices from confocal stacks collected using an FV3000 confocal. Key parameters such as laser intensity and detector gain were maintained constant across replicates and conditions. For perinuclear intensity analysis, a freehand line (thickness 20 to 30) was used to obtain the integrated (fluorescence) density. For nuclear intensity, we used Ciinte.Bra>3xnls::mNeonGreen as a reference of integrated (fluorescence) density that should not vary between different genotypes/treatments. The freehand selection tool was used. Multiple ROIs were used and Integrated (fluorescence) density was measured across both channels (red for the translation fusions of interest and green for 3xnls::mNeon-Green). The fraction of Red channel/Green channel was then used as a metric of relative enrichment of a given molecule in the nucleus. For cell–cell contact enrichment quantifications, we drew straight lines of thickness 30, used the plot profile function and measured the gray value in increments of ca 0.15 µm across both channels (plasma membrane channel which acted as control across conditions and channel of the translation fusion of interest). We analyzed the data in GraphPad Prism. For leading edge enrichment, we drew straight lines of length ca 33 µm. The 0 µm point was just outside the start of the leading edge. Scoring was done blind with respect to the treatment/genotype of the animals. We quantified 4 to 5 cells per animal for most experiments except for the nuclear enrichment experiments where we measured 10 cells per animal.

### CAMPARI imaging and analysis

*Ciinte.Brac>CAMPARI2* DNA was electroporated at 40 to 50 µg mixed with CRISPR/Cas9 plasmids. At stage 14, animals were transferred to Mattek chambers and imaged using an OLYMPUS spinning disk confocal. We took a pre-photoconversion stack for each animal assayed using two-color acquisition mode, illuminating our samples with 488 nm and 561 nm

laser lines. We then used a 405 nm laser at 40% for 5 s periods every 5 min. After 300 min, we took another stack for each animal using the 488 nm and 561 nm laser as the post-photoconversion stack. Using Fiji, we generated single-cell ROIs around multiple cells in each notochord that we imaged. The same ROIs were used to measure mean intensity in the Green (488nm) and Red (561nm) channel. We then calculated the R/G ratios for pre- and post-photoconversion cells.

## Supporting information

**S1 Fig. *C. intestinalis* Ano10 clusters together with vertebrate ANO10/TMEM16K.** (A) Phylogenetic tree for the ANO/TMEM16 family using 89 sequences from a diversity of eukaryotic species. The different ANO/TMEM16 branches are labeled with the different colors. ANO/TMEM16 proteins from *Ciona intestinalis* are highlighted in red boxes. (B) Multiple sequence alignment for a subset of ANO/TMEM16 proteins which have been reported to exhibit phospholipid scramblase or ion channel activity, using Clustal W and Clustal X. The colors correspond to amino acid identity. The blue lines mark the putative transmembrane domains. The phospholipid scramblase domain of mANO6 is enclosed in the black box. Residues that are important for $Ca^{2+}$ sensitivity are marked with red asterisks.
(TIFF)

**S2 Fig. A*no10* is expressed in the notochord during embryonic development; it is localized throughout development primarily in the ER and shows PM localization during tubulogenesis.** (A–K) Representative colorimetric whole-mount in situ with *Ano10* probe showing that initially *Ano10* is expressed broadly during neurula stages including expression in the endoderm, mesenchyme, and presumptive notochord. During early, mid, and late tailbud stages *Ano10* expression is restricted mostly to mesenchyme and the notochord. (L, M) A 2 kb *Ano10* regulatory element drives GFP expression in the notochord during late tailbud stages. In the representative pictures, we show maximal projection of late tailbud embryos. (N) At the end of notochord cell elongation ANO10 is localized primarily in the endoplasmic reticulum. Top panel shows expression of the ER marker *Ciintel.Brac>KDEL::BFP*; the middle panel shows *Brac>*Ano10::GFP in the same notochord cells. Bottom panel shows the merge between the 2 channels. (O) During lumen connection and cell flattening Ano10 is expressed in the ER but also at the plasma membrane (marked by white arrowhead). (P, Q) Representative examples of bidirectionally crawling cells from Stage VII notochords expressing the plasma membrane marker *Ciintel.Brac >hCD4::GFP* and a translational fusion of *Brac>*Ano10::mKate2. White arrowheads point to Ano10::mKate2 fusion localized to the Leading Edge of bidirectionally crawling cells. Red dash lines correspond to the lines used to generate the normalized intensity profiles shown in panels T and U. (R, S) Representative examples of notochord cells from Stages III and IV expressing *Ciintel.Brac >hCD4::GFP* and *Brac>*Ano10::mKate2. Red dash lines correspond to the lines used to generate the normalized intensity profiles shown in panels V and W. (T–W) Examples of normalized fluorescence intensity profiles of *Ciintel.Brac >hCD4::GFP* (green) and *Brac>*Ano10::mKate2 (magenta). The Pearson correlation coefficient between the 2 curves is show in each plot. (X) Quantification of Pearson correlation coefficients of fluorescence intensity line profiles (*Ciintel.Brac >hCD4::GFP* vs *Brac>*Ano10::mKate2) akin to those shown in panels (T–W). Two groups of cells were used. One group contained Stage VII cells and the other Stage III or IV; 40 cells per group were assayed. For statistical analysis, we performed a Mann–Whitney test (**** = $p < 0.0001$). Please see also S1 Table. The numerical data underlying this figure can be found in 10.5281/zenodo.12506448.
(TIFF)

**S3 Fig. Ano10 localizes primarily to the endoplasmic reticulum but also to the plasma membrane of mammalian cells.** (A) Representative confocal image of *C. intestinalis* Ano10::GFP expressed in HEK293T cells. (B) Representative confocal image showing the localization of the ER marker mCherry::Sec61 beta in HEK293T cells. (C) *C. intestinalis* Ano10::GFP and mCherry::Sec61 beta colocalize in the ER. (D) Additional representative confocal image of *C. intestinalis* Ano10::GFP expression in HEK293T cells. (E) Confocal image showing the localization of VE-cadherin::mApple in the same HEK293T cells. (F) A merge of the images in D and E. The white line corresponds to the site used for generating the intensity profiles shown in panel G. (G) Intensity profiles measured across the white line shown in panel F corresponding to *C. intestinalis* Ano10::GFP (green curve) and VE-cadherin::mApple (red curve). The numerical data underlying this figure can be found in 10.5281/zenodo.12506448.
(TIFF)

**S4 Fig. CRISPR/Cas9 efficiency for Ano10 locus and *Ano10$^{CRISPR}$* defects in maintenance of notochord boundary straightness and cell volume during convergent extension.** (A) Cleavage assay of *Ano10* Exon6 amplicon from pooled embryos electroporated with *Cii.Brachyury>nls::Cas9::nls*; *Cii.Brachyury>hCD4GFP* and *U6>ControlgRNA* or *U6>Ano10gRNA*. In embryos where *U6>Ano10gRNA* was used the cleavage efficiency for hCD4(+) enriched cell eluate was calculated at 39.0%. For the hCD4(-) eluate from the same pool of embryos the cleavage efficiency was 5.2% suggesting that Cas9 may have been functional outside the notochord. For example, the mesenchyme is a common site of ectopic expression of *Ciona* transgenes. The cleavage of *Ano10* Exon6 amplicon from control (*U6>ControlgRNA*) embryos was not detected. Cleavage of (B) control (top row marked with "C") and mutant (rows 1–4) *Ano10* alleles cloned from MACS-sorted hCD4(+) cells dissociated from embryos electroporated with *Cii.Brachyury>nls::Cas9::nls*; *Cii.Brachyury>hCD4GFP* and *U6>ControlgRNA* or *U6>Ano10gRNA*, respectively. Four out of 10 clones had a mutation. Target sequence indicated in blue. Deletions are indicated in red. (C–J) Illustration of the notochord boundary regularity measurement approach. (C, D) Maximal confocal projection examples of negative control and *Ano10$^{CRISPR}$* embryos expressing *Brac>GFP*. (E, F) Segmentation of the notochord using a binary mask operation (Watershed). (G, H) Notochord boundary outlines from the same embryos. (I, J) A straight line along the A–P axis is drawn to divide the notochord boundary to 2 sides. Both the net and total distance of the border are measured. (K) Quantification of notochord boundary straightness in negative control, *Ano10$^{CRISPR}$* and *Ano10$^{CRISPR}$*; *Brac>Ano10* embryos as the ratio of Total to net length. We assayed 33≤n≤40 animals per genotype. For statistical analysis, we performed a Kruskal–Wallis test, followed by Dunn's multiple comparisons test (**** $p < 0.0001$, ns $p > 0.05$, i.e., not significant). Please see also S7 Table. (L–N) Confocal projections showing immunostaining against laminin in negative control and *Ano10$^{CRISPR}$* embryos. Red arrowheads show the laminin staining along the notochord boundary. (O) Box plots quantifying the number of notochord cells in negative control and *Ano10$^{CRISPR}$* embryos. Statistical analysis was performed using Mann–Whitney test (ns = $p > 0.05$). Please see also S8 Table. We assayed $n \leq 47$ per genotype. (P) Box plot quantifications of notochord cell volumes. For statistical analysis, we performed Kruskal–Wallis test, followed by Dunn's multiple comparisons test (** $p < 0.005$; *** $p < 0.0005$, ns $p > 0.05$). We assayed 37≤n≤59 animals and 187≤n≤295 per genotype. Please see also S9 Table. The numerical data underlying this figure and the original gel picture corresponding to panel S4A can be found in 10.5281/zenodo.12506448.
(TIFF)

**S5 Fig. Expression pattern of carbonic anhydrase 2 KH.C1.423; KY21.Chr1.1715.** (A–C) Maximal projections of confocal stacks of embryos electroporated with a plasmid harboring a

3 kb fragment upstream of the carbonic anhydrase gene KH.C1.423; KY21.Chr1.1715. The promoter drives the expression of GFP primarily in the notochord, in some cases in the mesenchyme and in larvae; we also observed expression in a few neurons in the trunk. (A) Early tailbud embryos, (B) late tailbud embryo, (C) larva. See S18 Table for breakdown of % of electroporated animals expressing GFP under either the *Ciinte.Brac* or *Ciinte.Cah* promoter. (TIFF)

**S6 Fig. Drugs blocking Ano10 activity perturb notochord tubulogenesis.** (A–D) Montage of time-lapse confocal movies from (A) DMSO control, (B) 100 µm T16A(inh)-treated, (C) 100 µm NPBB-treated, and (D) 100 µm NS3728-treated embryos. Insets correspond to the regions marked by the yellow boxes. They show the developing lumen as demarcated by *Ciinte.Brac> Cav1::mCherry* (see S7–S10 Movies). (E) Quantification of lumen longitudinal radius during lumen initiation in DMSO and drug-treated embryos. For statistical analysis, we performed a Kruskal–Wallis test followed by a Dunn's multiple comparisons test; $61 \leq n$ cells and $12 \leq n$ of animals per condition (see also S23 Table). (F) Quantification of lumen longitudinal radius during lumen expansion in DMSO and drug-treated embryos in DMSO and drug-treated embryos. For statistical analysis, we performed a Kruskal–Wallis test followed by a Dunn's multiple comparisons test; $54 \leq n$ cells and $10 \leq n$ of animals per condition (see also S24 Table). (G) Quantification of lumen curvature after lumen connection. For statistical analysis, we performed a Kruskal–Wallis test followed by a Dunn's multiple comparisons test; $36 \leq n$ cells and $7 \leq n$ of animals per condition (see also S25 Table). The red and green lines correspond to the median and quartiles, respectively. (H) Evolution of lumen cross-sectional area during tubulogenesis. Schematic shows in purple the lumen cross-section and in green the notochord cell membrane. For statistical analysis, we performed a mixed-effects model followed by Tukey's multiple comparison test; $24 \leq n$ of animals per condition (see also S26 Table). (I) Quantification of the distance between adjacent lumens in negative control, Ano10$^{CRISPR}$, and rescue embryos. Inset shows a transmitted light view of adjacent expanding lumens. Red line corresponds to the distance between adjacent expanding lumens. For statistical analysis, we performed a Kruskal–Wallis test followed by a Dunn's multiple comparisons test; $150 \leq n$ cells and $30 \leq n$ of animals per condition (see also S27 Table). (I) Quantification of the lumen tilting angle in negative control, Ano10$^{CRISPR}$ and rescue embryos. Inset shows a transmitted light view of adjacent tilting lumens. Red lines correspond to the axes of titling of the cells. The tilting angle ($\theta$) is measured between each of the red lines and the perpendicular white line along the dors-ventral axis. For statistical analysis, we performed a Kruskal–Wallis test followed by a Dunn's multiple comparisons test; $168 \leq n$ cells and $33 \leq n$ of animals per condition (see also S28 Table). The numerical data underlying this figure can be found in 10.5281/zenodo.12506448. (TIFF)

**S7 Fig. *Ciinte.Brachyury>CAMPARI2* reports differences in long-term notochord Ca$^{2+}$ activity between negative control, *Ano10$^{CRISPR}$* and Ano10 rescue animals.** (A–F) Split views of sum projections corresponding to the composite sum projections from representative negative control, *Ano10$^{CRISPR}$* and rescue embryos expressing the Ca$^{2+}$ integrator CAMPARI2 in the notochord, following photoconversion pulses during convergent extension (panels A–C in Fig 7). The Red (561 nm) and Green (488 nm) channels are shown separately. Anterior to the left. Lateral animal views are shown. (G–L) Split views of sum projections corresponding to the composite sum projections from representative negative control, *Ano10$^{CRISPR}$* and rescue embryos expressing the Ca$^{2+}$ integrator CAMPARI2 in the notochord, following photoconversion pulses during tubulogenesis (panels D–F in Fig 7). The Red (561 nm) and Green (488 nm) channels are shown separately. Anterior to the left. Lateral animal views are shown.

(M) Montage of selected from S4 Movie, highlighting a $Ca^{2+}$ wave that is traveling from one bidirectionally cell (Cell1) via neighboring leading and trailing edges (edges shown with white arrowheads) to an adjacent cell (Cell2) and from there again to a third cell (Cell3). After the $Ca^{2+}$ transient enters Cell2 it elicits a $Ca^{2+}$ response in its nucleus (white asterisk). The white arrow indicates the direction of travel of the $Ca^{2+}$ transient.
(TIFF)

**S1 Movie. Confocal time lapse imaging of *Ciiinte.Brac>hCD4-GFP (green); Ciiinte.Caveolin 1-mCherry* (magenda) covering the period of tubulogenesis in a negative control embryo.**
(AVI)

**S2 Movie. Confocal time lapse imaging of *Ciiinte.Brac>hCD4-GFP (green); Ciiinte.Caveolin 1-mCherry* (magenda) covering the period of tubulogenesis in an Ano10<sup>CRISPR</sup> embryo.**
(AVI)

**S3 Movie. Confocal time lapse imaging of *Ciiinte.Brac>hCD4-GFP (green); Ciiinte.Caveolin 1-mCherry* (magenda) covering the period of tubulogenesis in an Ano10<sup>CRISPR;</sup> *Ciiinte. Brac>Ano10cDNA rescue* embryo.**
(AVI)

**S4 Movie. Representative example movie of *Ciiinte.Brac>GCaMP8f* expressing notochord cells showing $Ca^{2+}$ activity during bidirectional crawling in negative control animals. Each frame is a sum projection of a confocal volume composed of 8 to 10 slices.**
(AVI)

**S5 Movie. Representative example movie of *Ciiinte.Brac>GCaMP8f* showing $Ca^{2+}$ activity in Ano10<sup>CRISPR</sup> notochord cells during the period that they should be performing bidirectional crawling. Each frame is a sum projection of a confocal volume composed of 8 to 10 slices.**
(AVI)

**S6 Movie. Representative example movie of *Ciiinte.Brac>GCaMP8f* showing $Ca^{2+}$ activity in *Ano10<sup>CRISPR</sup>; Ciinte.Brac>Ano10* rescued notochord cells during the period that they should be performing bidirectional crawling. Each frame is a sum projection of a confocal volume composed of 8 to 10 slices.**
(AVI)

**S7 Movie. Confocal time lapse imaging of *Ciiinte.Brac>hCD4-GFP (green); Ciiinte.Caveolin 1-mCherry* (magenda) covering the period of tubulogenesis in DMSO incubated control embryo.**
(AVI)

**S8 Movie. Confocal time lapse imaging of *Ciiinte.Brac>hCD4-GFP (green); Ciiinte.Caveolin 1-mCherry* (magenda) covering the period of tubulogenesis in a T16A inhibitor (100 μm) incubated embryo.**
(AVI)

**S9 Movie. Confocal time lapse imaging of *Ciiinte.Brac>hCD4-GFP (green); Ciiinte.Caveolin 1-mCherry* (magenda) covering the period of tubulogenesis in a NPPB (100 μm) inhibitor incubated embryo.**
(AVI)

**S10 Movie. Confocal time lapse imaging of *Ciiinte.Brac>hCD4-GFP (green); Ciiinte. Caveolin 1-mCherry* (magenda) covering the period of tubulogenesis in a NS3728 (100 μm)**

**inhibitor incubated embryo.**
(AVI)

**S1 Table. Statistical analysis of data corresponding to panel S2X Fig.**
(XLSX)

**S2 Table. Statistical analysis of data corresponding to panel Fig 1G.**
(XLSX)

**S3 Table. Statistical analysis of data corresponding to panel Fig 1H.**
(XLSX)

**S4 Table. Statistical analysis of data corresponding to panel Fig 1I.**
(XLSX)

**S5 Table. Statistical analysis of data corresponding to panel Fig 1P.**
(XLSX)

**S6 Table. Statistical analysis of data corresponding to panel Fig 1Q.**
(XLSX)

**S7 Table. Statistical analysis of data corresponding to panel S4K Fig.**
(XLSX)

**S8 Table. Statistical analysis of data corresponding to panel S4O Fig.**
(XLSX)

**S9 Table. Statistical analysis of data corresponding to panel S4P Fig.**
(XLSX)

**S10 Table. Statistical analysis of data corresponding to panel Fig 2D.**
(XLSX)

**S11 Table. Statistical analysis of data corresponding to panel Fig 2E.**
(XLSX)

**S12 Table. Statistical analysis of data corresponding to panel Fig 2F.**
(XLSX)

**S13 Table. Statistical analysis of data corresponding to panel Fig 2G.**
(XLSX)

**S14 Table. Statistical analysis of data corresponding to panel Fig 2R.**
(XLSX)

**S15 Table. Statistical analysis of data corresponding to panel Fig 2S.**
(XLSX)

**S16 Table. Statistical analysis of data corresponding to panel Fig 2T.**
(XLSX)

**S17 Table. Statistical analysis of data corresponding to panel Fig 2U.**
(XLSX)

**S18 Table. Breakdown of % of electroporated animals expressing GFP under the *Ciinte. Brac* or *Ciinte.Cah* promoter corresponding to S5 Fig.**
(XLSX)

**S19 Table. Statistical analysis of data corresponding to panel Fig 3E.**
(XLSX)

**S20 Table. Statistical analysis of data corresponding to panel Fig 3F.**
(XLSX)

**S21 Table. Statistical analysis of data corresponding to panel Fig 3G.**
(XLSX)

**S22 Table. Statistical analysis of data corresponding to panel Fig 3H.**
(XLSX)

**S23 Table. Statistical analysis of data corresponding to panel S6E Fig.**
(XLSX)

**S24 Table. Statistical analysis of data corresponding to panel S6F Fig.**
(XLSX)

**S25 Table. Statistical analysis of data corresponding to panel S6G Fig.**
(XLSX)

**S26 Table. Statistical analysis of data corresponding to panel S6H Fig.**
(XLSX)

**S27 Table. Statistical analysis of data corresponding to panel S6I Fig.**
(XLSX)

**S28 Table. Statistical analysis of data corresponding to panel S6J Fig.**
(XLSX)

**S29 Table. Statistical analysis of data corresponding to panel Fig 4D.**
(XLSX)

**S30 Table. Statistical analysis of data corresponding to panel Fig 4E.**
(XLSX)

**S31 Table. Statistical analysis of data corresponding to panel Fig 4F.**
(XLSX)

**S32 Table. Statistical analysis of data corresponding to panel Fig 4Q.**
(XLSX)

**S33 Table. Statistical analysis of data corresponding to panel Fig 4R.**
(XLSX)

**S34 Table. Statistical analysis of data corresponding to panel Fig 4S.**
(XLSX)

**S35 Table. Statistical analysis of data corresponding to panel Fig 5G.**
(XLSX)

**S36 Table. Statistical analysis of data corresponding to panel Fig 5N.**
(XLSX)

**S37 Table. Statistical analysis of data corresponding to panel Fig 5R.**
(XLSX)

**S38 Table. Statistical analysis of data corresponding to panel Fig 5S.**
(XLSX)

**S39 Table. Statistical analysis of data corresponding to panel Fig 5T.** (XLSX)

**S40 Table. Statistical analysis of data corresponding to panel Fig 5U.** (XLSX)

**S41 Table. Statistical analysis of data corresponding to panel Fig 6D.** (XLSX)

**S42 Table. Statistical analysis of data corresponding to panel Fig 6E.** (XLSX)

**S43 Table. Statistical analysis of data corresponding to panel Fig 6F.** (XLSX)

**S44 Table. Statistical analysis of data corresponding to panel Fig 6S.** (XLSX)

**S45 Table. Statistical analysis of data corresponding to panel Fig 6T.** (XLSX)

**S46 Table. Statistical analysis of data corresponding to panel Fig 6U.** (XLSX)

**S47 Table. Statistical analysis of data corresponding to panel Fig 7G.** (XLSX)

**S48 Table. Statistical analysis of data corresponding to panel Fig 7H.** (XLSX)

**S49 Table. Statistical analysis of data corresponding to panel Fig 7S.** (XLSX)

**S50 Table. Statistical analysis of data corresponding to panel Fig 7T.** (XLSX)

**S51 Table. Statistical analysis of data corresponding to panel Fig 8G.** (XLSX)

**S52 Table. Statistical analysis of data corresponding to panel Fig 8H.** (XLSX)

**S53 Table. Statistical analysis of data corresponding to panel Fig 8I.** (XLSX)

**S54 Table. Statistical analysis of data corresponding to panel Fig 8M.** (XLSX)

**S55 Table. Statistical analysis of data corresponding to panel Fig 8N.** (XLSX)

**S56 Table. Statistical analysis of data corresponding to panel Fig 8S.** (XLSX)

**S57 Table. Statistical analysis of data corresponding to panel Fig 8X.** (XLSX)

**S58 Table. Statistical analysis of data corresponding to panel Fig 8Y.**
(XLSX)

**S59 Table. Statistical analysis of data corresponding to panel Fig 9G.**
(XLSX)

**S60 Table. Statistical analysis of data corresponding to panel Fig 9R.**
(XLSX)

**S61 Table. Statistical analysis of data corresponding to panel Fig 9S.**
(XLSX)

**S62 Table. Statistical analysis of data corresponding to panel Fig 10G.**
(XLSX)

**S63 Table. Statistical analysis of data corresponding to panel Fig 10K.**
(XLSX)

**S64 Table. Statistical analysis of data corresponding to panel Fig 10L.**
(XLSX)

**S65 Table. Statistical analysis of data corresponding to panel Fig 10P.**
(XLSX)

**S66 Table. Statistical analysis of data corresponding to panel Fig 10Q.**
(XLSX)

**S67 Table. Statistical analysis of data corresponding to panel Fig 10R.**
(XLSX)

**S68 Table. Statistical analysis of data corresponding to panel Fig 11G.**
(XLSX)

**S69 Table. Primers list.**
(XLSX)

**S1 Raw Images Raw blots.**
(PDF)

## Acknowledgments

We would like to thank Dr. Mie Wong for her feedback on the manuscript. We are grateful to Dr. Di Jiang and his laboratory members (Sars Centre 2006–2014) for sharing various constructs used in this study.

## Author Contributions

**Conceptualization:** Zonglai Liang, Marios Chatzigeorgiou.

**Data curation:** Zonglai Liang, Daniel Christiaan Dondorp, Marios Chatzigeorgiou.

**Formal analysis:** Zonglai Liang, Daniel Christiaan Dondorp, Marios Chatzigeorgiou.

**Funding acquisition:** Marios Chatzigeorgiou.

**Investigation:** Zonglai Liang.

**Methodology:** Zonglai Liang, Daniel Christiaan Dondorp.

**Project administration:** Marios Chatzigeorgiou.

**Software:** Daniel Christiaan Dondorp.

**Supervision:** Marios Chatzigeorgiou.

**Validation:** Zonglai Liang, Daniel Christiaan Dondorp, Marios Chatzigeorgiou.

**Visualization:** Zonglai Liang, Daniel Christiaan Dondorp, Marios Chatzigeorgiou.

**Writing – original draft:** Zonglai Liang, Marios Chatzigeorgiou.

**Writing – review & editing:** Zonglai Liang, Daniel Christiaan Dondorp, Marios Chatzigeorgiou.

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
