## [Editor Report · Decision Letter 0]

15 Feb 2023

Dear Dr Chatzigeorgiou, 

Thank you for submitting your manuscript entitled "Anoctamin 10/TMEM16K coordinates organ morphogenesis across scales in the urochordate notochord." for consideration as a Research Article by PLOS Biology.

Your manuscript has now been evaluated by the PLOS Biology editorial staff as well as by an academic editor with relevant expertise and I am writing to let you know that we would like to send your submission out for external peer review.

Once your full submission is complete, your paper will undergo a series of checks in preparation for peer review. After your manuscript has passed the checks it will be sent out for review. To provide the metadata for your submission, please Login to Editorial Manager (https://www.editorialmanager.com/pbiology) within two working days, i.e. by Feb 17 2023 11:59PM.

Kind regards,

Ines

--

Ines Alvarez-Garcia, PhD

Senior Editor

PLOS Biology

---

## [Decision Letter · Decision Letter 1]

28 Apr 2023

Dear Dr Chatzigeorgiou,

Thank you for your patience while your manuscript entitled "Anoctamin 10/TMEM16K coordinates organ morphogenesis across scales in the urochordate notochord" was peer-reviewed at PLOS Biology. Also please accept my sincere apologies for the delay in providing you with our decision. Your manuscript has been evaluated by the PLOS Biology editors, an Academic Editor with relevant expertise, and by three independent reviewers.

The reviews are attached below. As you will see, the reviewers find your work potentially interesting, however they also raise several concerns that would need to be addressed in order for us to consider the manuscript further for publication. The reviewers think that the mechanistic details should be better explored and that stronger evidence is needed to support the conclusion that Ano10 functions as a calcium channel during convergence extension and lumen formation.

Based on their specific comments and following discussion with the Academic Editor, it is clear that a substantial amount of work would be required to meet the criteria for publication in PLOS Biology. However, given our and the reviewer interest in your study, we would be open to inviting a comprehensive revision of the study that thoroughly addresses all the reviewers' comments. Due to the extent of revision that would be needed, we cannot make a decision about publication until we have seen the revised manuscript and your response to the reviewers' comments. Your revised manuscript would need to be seen by the reviewers again, but please note that we would not engage them unless their main concerns have been addressed. 

We appreciate that these requests represent a great deal of extra work, and we are willing to relax our standard revision time to allow you 6 months to revise your study. Please email us (plosbiology@plos.org) if you have any questions or concerns, or envision needing a (short) extension.

**IMPORTANT - SUBMITTING YOUR REVISION**

3. Resubmission Checklist

a) *PLOS Data Policy*

b) *Published Peer Review*

d) *Blurb*

Please also provide a blurb which (if accepted) will be included in our weekly and monthly Electronic Table of Contents, sent out to readers of PLOS Biology, and may be used to promote your article in social media. The blurb should be about 30-40 words long and is subject to editorial changes. It should, without exaggeration, entice people to read your manuscript. It should not be redundant with the title and should not contain acronyms or abbreviations. For examples, view our author guidelines: https://journals.plos.org/plosbiology/s/revising-your-manuscript#loc-blurb

Sincerely,

Ines

--

Ines Alvarez-Garcia, PhD

Senior Editor

PLOS Biology

Reviewers' comments

Rev. 1:

This manuscript describes the results of tissue-specific knockout with the aim to analyze the role of anoctamin 10 during notochord morphogenesis in the ascidian embryo. Anoctamin family members have been known as calcium-activated chloride channels, and some of the members show phospholipid scramblases activity in vertebrates. ANO 10 gene is expressed in developing notochord in ascidians. The manuscript reports that knockout of ANO 10 in developing notochord cells of ascidians resulted in defects in larval notochord morphogenesis, convergent extension and tubulogenesis steps. During convergent extension, notochord cells failed to intercalate each other. The authors showed that outer boundary of notochord tissue was not smooth probably because of decrease of laminin. During tubulogenesis, the lumen tilting and connection was abrogated with deranged microtubule and actin localization. In both steps, ANO 10 was shown to modulate intracellular Ca2+ dynamics, but the effect was rather minor or small. The authors also showed ascidian ANO 10 acts as an ion channel but not as a phospholipid scramblase using mammalian culture cells.

Phenotypes of ANO 10 knockout in notochord cells were evaluated across scales, namely tissue morphology, cell shape, cytoskeletal organization, and Ca transient, however, causal relationship between those abnormalities at each scale was not shown. Especially, effect on Ca transient was rather small. Although the findings are of interest, their appeal to the readership of this journal is too limited to warrant publication. As presented, the paper is better suited for a more specialized journal. The manuscript is not well written and should be reorganized to remove redundant sentences and phrases. Introduction section is not enough to provide readers with previous knowledge on ANO 10.

1. I am curious why ANO 10 affect intracellular calcium dynamics because ANO 10 is likely calcium-activated chloride channels. Ascidian ANO 10 has conserved calcium binding amino acid residues as shown in Figure S1.

2. Methods section is too long. It spans 9 pages and is longer than Result section. There are a lot of redundant description and reiterated sentences. The detailed descriptions of protocol should be transferred to Supplementary materials.

3. The sentence in Introduction section "The notochord is an essential tubular organ present in the embryonic midline region of all members of the chordate phylum" would be also included in Abstract.

4. Introduction. "While some Anoctamins, such as ANO1/TMEM16A, act as a CaCC, and ANO6/TMEM16F, as a PS; for other members of the family, including ANO10/TMEM16K, their functions remain under fruitful debate". Please explain more about ANO 10. What is "fruitful debate"?

Minor comments:

5. Page 5, "while the nuclei exhibited a medial localization". I do not understand where a medial localization means.

6. Page 6, "at later developmental stages". Please explicitly describe when the carbonic anhydrase promotor drives the expression. Theis is important information to interpret the experimental results.

7. Page 6, "applying three different drugs that are well established blockers of Anoctamins". What kind of activity of Anoctamins is blocked by the drugs, calcium-activated chloride channels or scramblases activities?

8. Page 6, "Caveolin 1-mCherry 24 to simultaneously visualize the plasma membrane and the apical/luminal region of the notochord cells." Please explain why Caveolin 1 is localized to luminal membrane.

9. Page 7. Fig. 3J-L and O-Q. Are these lateral views of notochord or transverse sections of a notochord cells?

10. Page 9. the acting- based protrusions would be the actin-based protrusions.

11. Page 9, "at the basal domains". I did not understand where is the basal domain until I looked at Fig. 3W. The authors should transfer Fig. 3W to Fig. 1 to help readers easily recognize which region corresponds to where because notochord cells are cylindrical and it is not ordinary epithelial cells.

12. Fig. 3. I do not recognize differences between Fig. 3T and U.

13. Table 1. Why minor differences observed in GCaMP6 is expanded in observations using CAMPARI2? I understand the duration of observation is different between these two. Please explain why longer observation is better to recognize the differences.

14. Fig. 4 A-B. Is it possible to draw the outline of each notochord cells?

15. Fig. 4 legends. (K,L) should be replaced with (N, O).

16. Page 11, "to detect PS activity". Does PS stand for phospholipid scramblases or Phosphatidylserine (see 7th line above)? This is confusing.

17. Page 12, "When we used the CAAX tag to tether Ano10 at the PM, we obtained larger currents at positive voltages". It seems strange that addition of the membrane tethering motif, CAAX, to ANO 10, which already has ten transmembrane domains, further enhances membrane localization of ANO 10.

18. Methods. Page 14, "100 to 200 µg of DNA" would be "100 to 200 µg/ml of DNA".

19. Page15, "the Ano10 cDNA was amplified We then using cDNA". Remove "We then".

20. Page 16, "TdTomato-LifeAct tdTomato-Lifeact-7 was a gift". Remove "-LifeAct tdTomato".

21. Page 16, "We imaged our samples were on a Nikon Eclipse E800". Remove "were".

22. Page 17, "Primers used to build reagents used in these experiments". Remove the second "used".

23. Page 18, "A secondary antibody mouse anto-rabbit was applied." Anto would be aniti.

24. Page 20, "We then stained me with a conjugated Alexa Fluor 647 phalloidin antibody". This sentence is strange.

25. Page 23, "the paper with input from D". D would be D.D.

26. Page 31, "Ca2+ activity during notochord cell bi-directional cloning". Cloning would be crawling.

Rev. 2:

Chatzigeorgiou's manuscript revealed the functions of an evolutionary conserved transmembrane protein, anoctamin 10/TMEM16K, in notochord tissue morphology in a marine ascidian. They found that Ano10 was expressed and required for the whole processes of notochord development including convergent extension, elongation, and lumen formation and coalescence. The evidences that they showed demonstrated that Ano10/Tmem16k seemly acted as an ion channel to regulate the elctrophysioloigical signaling, and therefore affected the cytoskeleton distribution and organization that drive notochord morphogenesis. This work is an interesting story, which reveals the new role of bioelectrical signaling in organ morphogenesis. I support the publication of this work given the follow concerns could be well addressed.

Main concerns:

1. The authors stated that Ano10 was required for notochord convergent extension, lumen expansion, and tubulogenesis, three main processes of notochord morphogenesis. One of my main concerns is that whether Ano10 use the same pathway and machinery on these three processes? The authors need to clarify on this point and show the related supporting data.

2. The authors showed that Ano10 was important for regulating Ca2+ dynamics during notochord morphogenesis. The current data suggested Ano10 acted as a channel not scamblase. The question remains to the audience is that how a channel protein regulates Ca2+ signaling, what is the behind connection. The current data seems not to be straightforward and need further addressed.

3. The authors observed that the polarity and organization of cytoskeleton were defected during convergent extension and tube formation in Ano 10 knockout notochord cells. Further detailed need to elucidate how actin and/or microtubules are controlled by Ano10 during these processes.

Additional comments

4. The gene expression in ascidian shows mosaic pattern. When authors showed the phenotypes in Ano 10 knockouted embryos, the crispr-expressed cell should be marked out (Fig. 1 L, M). So that the audience could know which cell is knockout one and which one is wild type?

5. The knockout efficiency of Ano 10 needs to be showed out. Either by genomic digestion approach or by the comparison of the expression level at RNA or protein level.

6. I didn't find J, K, L in Figure 3. It might be mislabeled.

7. The name of Ciona species should be consistent. Both Ciona robusta (Page 10) and Ciona intestinalis (many places in ms.) were used in this manuscript. The authors could should let the audience know that Ciona intestinalis type A= Ciona robusta, Otherwise, might cause some confusion.

Rev. 3:

The work by Liang and collaborators uses genome editing and quantitative imaging to characterise the role of Ano10/TMEM16K during Ciona organogenesis to find that Anotacmin/TMEM16 is key for convergent extension, lumen expansion and connection of the notochord during development. They show lack of Ano10/TMEM16K results in a lack of bioelectrical signalling from the notochord, using electrophysiology approaches, and that Ano10/TMEM16K acts as a ion/channel but not as a phospholipid scramblase, as it is the case for the mouse Ano6. The authors conclude that Ano10/TMEM16K might be a conserved pathway controlling organ morphogenesis, as their orthologous genes have been also seen during the formation of tubular structures in vertebrates.

I think in general the data is very compelling. The reporter expression and the imaging is really beautiful and clear. Their phenotypes on convergence extension (CE) and lumen formation are very clear too. I think the conditional knock down using the carbonic anhydrase promotor to dissect the CE role from the role in lumen formation is a very clever strategy. I am not so sure why they worked with the hypothesis that Ano10 might be a phospholipid scramblase, as the Ano6 belongs to another subfamily, and it is one of the few members of the family having that role, but the experiments are good enough to discard that possibility.

The calcium imaging is also very informative, but I do not think they have uploaded the images or movies of the Calcium reporter and the Campari embryos in an ANO10-CRISPR background. If the data is there, could it be that it has not been properly labelled?

In Figure 4, for example, which are the ANO10 crispr embryos? There is also no information on how the experiment was done here: is it a Ciinte.Brac>GCaMP6s + Ano10CRISPR? Why there are no images of these embryos? There is no rescue embryos here either?

I would encourage the authors to make this part clearer. The quantifications of the calcium amplitude, duration and rising and falling slopes do not show big differences, especially during convergence extension (FIG 4D-G), so making the movies they have used for the quantification available might help to reinforce the argument in favour of ANO10 acting as a calcium channel during CE and lumen formation.

The only other thing I miss is a connection between the different parts of the study. If ANO10 is a calcium channel, key for CE and lumen formation:

1. What is the mechanism?

2. How is ANO10 guiding the formation of the lamellipodia at the right site of the cell and the tilting at the right angle to connect the lumen?

3. Do the authors think is the same mechanism acting during CE and lumen formation?

Co-localisation of ANO10 to the membrane and in relation to the site of formation of the lamellipodia would be a useful piece of data to show. The authors have generated a lot of useful lines that I think would answer these questions. Could the authors use, for example, their Cinte.ANO10>Unc76::GFP or the Brac>Ano10::GFP to see where ANO10 localises? Supplementary figure 2 shows ANO10 in the RE and the plasma membrane, but the images are not very clear. Do the authors have magnified views of that localisation? And when in the plasma membrane, does this correlates with a specific phase of cell shape change? Or with a burst of Calcium?

Calcium could be acting as a directional queue or as a regulator of actin cytoskeleton during the formation of lamellipodia, what the authors think is happening? Cofilin for example is an actin-depolymerization factor that promotes lamellipodia formation. This is inhibited by RAC, which is activated by Calcium. Do the authors think ANO10 might be a mediator in this process or is it a completely different mechanism?

Other comments that might improve the manuscript:

1. Some cells do not appear to have staining by the look of the pictures - probably due to mosaic expression - (e.g. Fig1), so if this is right, were the authors able to quantify all cells? And if not? Which proportion of the cells were quantified? And was this proportion equal across embryos and conditions?

2. In Figure 1 ANO10-CRISPR embryos seem to have a marked staining because they do not extend. Is that taken into account in the quantifications?

3. Just out of curiosity, is the volume of the cells changing in ANO10-CRISPR embryos or only the aspect ratio?

4. The orientation of the embryos is not indicated anywhere. I guess it is lateral in most cases, but it would be good to be specific here. It gives a better context when comparing with previous morphometric work done by the Veeman lab and others.

5. If in the ANO10-CRISPR embryos the lumen does not connect is it because it is expanding in the wrong direction, or it simply does not reach?

I think if the authors could address these concerns the paper would be suitable for publication in Plos Biology.

---

## [Decision Letter · Decision Letter 2]

11 Jun 2024

Dear Dr Chatzigeorgiou,

Thank you for your patience while we considered your revised manuscript entitled "Anoctamin 10/TMEM16K coordinates organ morphogenesis across scales in the urochordate notochord." for publication as a Research Article at PLOS Biology. This revised version of your manuscript has been evaluated by the PLOS Biology editors, the Academic Editor and two of the original reviewers.

Based on the reviews, we are likely to accept this manuscript for publication, provided you satisfactorily address the remaining points raised by Reviewer 2. Please also make sure to address the data and other policy-related requests stated below.

In addition, we would like you to consider a suggestion to improve the title:

"The ion channel Anoctamin 10/TMEM16K coordinates organ morphogenesis across scales in the urochordate notochord"

We expect to receive your revised manuscript within two weeks. 

*Published Peer Review History*

*Press*

Sincerely,

Ines

--

Ines Alvarez-Garcia, PhD

Senior Editor

PLOS Biology

Fig. 1G, H, I, P, Q; Fig. 2D-G, R-U; Fig. 3E-H; Fig. 4D-F, Q-S; Fig. 5D-G, K-N; R-U; Fig. 6D-F, S-U; Fig. 7G, H, J-T; Fig. 8G-I, M, N, S, X, Y; Fig. 9D-G, R, S; Fig. 10D-Q, K-R; Fig. 11F, G; Fig. S2T-X; Fig. S3G; Fig. S4K, O, P and Fig. S6E-J

CODE POLICY

Per journal policy, if you have generated any custom code during the curse of this investigation, please make it available without restrictions upon publication. Please ensure that the code is sufficiently well documented and reusable, and that your Data Statement in the Editorial Manager submission system accurately describes where your code can be found. 

Please note that we cannot accept sole deposition of code in GitHub, as this could be changed after publication. however, you can archive this version of your publicly available GitHub code to Zenodo. Once you do this, it will generate a DOI number, which you will need to provide in the Data Accessibility Statement (you are welcome to also provide the GitHub access information). See the process for doing this here: https://docs.github.com/en/repositories/archiving-a-github-repository/referencing-and-citing-content

We require the original, uncropped and minimally adjusted images supporting all blot and gel results reported in an article's figures or Supporting Information files - in this case in Fig. S4A. We will require these files before a manuscript can be accepted so please prepare and upload them now. Please carefully read our guidelines for how to prepare and upload this data: https://journals.plos.org/plosbiology/s/figures#loc-blot-and-gel-reporting-requirements

Reviewers comments:

Rev. 2:

The authors did a very good job to address most of my concerns in the revision. The only left unclear thing is an exact molecular mechanism on how the Ca2+ signaling regulates the localization of RhoA/cofilin, and if the RhoA/cofilin-cytoskeleton is the only target for notochord morphogenesis regulation. Due to the extensive influence of Ca2+ signaling, it requires a lot of additional work to clarify above questions. It might not be reasonable to ask the authors to provide the exact mechanism in the current already detailed report. However, consideration on the other potential cellular processes such as the vesicle trafficking and membrane transport that might be also influenced by RhoA, I suggest the authors tone down their statement that the RhoA/cofilin-cytoskeleton is a potential pathway and add additional discussion on the other possibility.

In addition, it will be friendly for the audience to understand the multiple potential mechanisms on the roles of Ca2+ signaling at the different stages of notochord morphogenesis. I suggest the authors make a model illustration at the end of paper.

Minor comments for the Figures

1. Figure 9P, the figure supposed to be labeled as Ano10CRISPR. However, the labeling is hided by the picture.

2. Figure 9I, there exists an extra asterisk near the labeling of "CRISPR" that need to be removed.

3. Figure 10N, it seems that the magnified picture does not match the regions of projection.

Please check the detailed for all the panels of the figures.

Rev. 3: Elia Benito Gutierrez - note that this reviewer has signed her review

The authors have done an excellent job at revising their manuscript. I find this version much clearer and more interpretable than the previous one. Their results are now better supported and their argument and conclusion much more robust. The mechanism of action of ANO10 is clearer and it is very interesting that slightly different interactions might be at play during convergent extension and tubulogenesis.

---

## [Editor Report · Decision Letter 3]

20 Jul 2024

Dear Dr Chatzigeorgiou,

Thank you for the submission of your revised Research Article entitled "The ion channel Anoctamin 10/TMEM16K coordinates organ morphogenesis across scales in the urochordate notochord" for publication in PLOS Biology. On behalf of my colleagues and the Academic Editor, Yi-Hsien Su, I am delighted to let you know that we can in principle accept your manuscript for publication, provided you address any remaining formatting and reporting issues. These will be detailed in an email you should receive within 2-3 business days from our colleagues in the journal operations team; no action is required from you until then. Please note that we will not be able to formally accept your manuscript and schedule it for publication until you have completed any requested changes.

PRESS

Sincerely, 

Ines

--

Ines Alvarez-Garcia, PhD

Senior Editor

PLOS Biology
